# Divergence of seafloor elevation and sea level rise in coral reef ecosystems

Kimberly K. Yates[1], David G. Zawada[1], Nathan A. Smiley[1], Ginger Tiling-Range[2]

[1]U.S. Geological Survey, Coastal and Marine Science Center, 600 Fourth St. South, St. Petersburg, Florida 33701, U.S.A.

*Correspondence to*: Kimberly K. Yates (kyates@usgs.gov)

**Abstract.** Coral reefs serve as natural barriers that protect adjacent shorelines from coastal hazards such as storms, waves and erosion. Projections indicate global degradation of coral reefs due to anthropogenic impacts and climate change will cause a transition to net erosion by mid-century. Here, we provide a comprehensive assessment of the combined effect of all
10 of the processes affecting seafloor accretion and erosion by measuring changes in seafloor elevation and volume for 5 coral reef ecosystems in the Atlantic, Pacific and Caribbean over the last several decades. Regional-scale, mean-elevation and volume losses were observed at all 5 study sites and in 77% of the 60 individual habitats that we examined across all study sites. Mean seafloor elevation losses for whole coral reef ecosystems in our study ranged from -0.09 m to -0.8 m, corresponding to net volume losses ranging from 3.4 to 80.5 $Mm^3$ for all study sites. Erosion of both coral-dominated
substrate and non-coral substrate suggests that the current rate of carbonate production is no longer sufficient to support net accretion of coral reefs or adjacent habitats. We show that regional-scale loss of seafloor elevation and volume has accelerated the rate of relative sea level rise in these regions. Current water depths have increased to levels not predicted until near the year 2100, placing these ecosystems and nearby communities at elevated and accelerating risk to coastal hazards. Our results set a new baseline for projecting future impacts to coastal communities resulting from degradation of
coral reef systems and associated losses of natural and socio-economic resources.

## 1 Introduction

Coral reef ecosystems develop over thousands of years as organisms build skeletons of calcium carbonate minerals that form complex 3-dimensional structures, and keep pace with rising sea level through continued growth and accretion of carbonate sediments. These ecosystems support up to 25% of fisheries in tropical regions and developing nations (Garcia and Moreno,
2003), and economic and recreational services for more than 100 countries (Burke et al., 2011). Reef framework and shallow, non-coral-dominated habitats serve as natural barriers that protect shoreline ecosystems and coastal communities by reducing hazards from waves, storm surges and tsunamis for more than 200 million people around the world (Sheppard et

al., 2005; Ferrario et al., 2014). Local and global, natural and human-induced stressors have caused the loss of reef-building organisms and reef structure (Alvarez-Filip et al., 2009), a decrease in biodiversity, a transition to algal-dominated communities (Pandolfi et al., 2003), and an increase in bioerosion by some species of sponges and other boring and endolithic bioeroders (Tribollet et al., 2009; Wisshak et al., 2012; Reyes-Nivia et al., 2013; Wisshak et al., 2013; Decarlo et

al., 2015; Enochs et al., 2015) placing coral reefs around the world in a state of rapid decline (Madin and Madin, 2015).

Coral reef degradation and the causes have been well documented since the 1970's (Gardner et al., 2003) when regional-scale species compositions that remained stable throughout the Pleistocene and Holocene began changing (Aronson et al., 2002). Most reef degradation has been attributed primarily to impacts of coastal development, overfishing, pollution, nutrient

enrichment, coral bleaching and disease (Glynn, 1984; Greenstein et al., 1998). Abundance of reef-building corals has decreased as much as 72% since 1968 on Pacific reefs and as much as 50% since 1970 on many Caribbean reefs (Gardner et al., 2003; Bruno and Selig, 2007; Jackson et al., 2014). Carbonate production rates have decreased to below historical values on Caribbean reefs and carbonate-budget models indicate that some are already experiencing net erosion (Perry et al., 2013; Enochs et al., 2015). Projections indicate up to 66% of coral reefs worldwide will continue to degrade in the next few

decades due to ocean warming and acidification from unprecedented rates of global climate change (Frieler et al., 2013) that may cause reef erosion to exceed accretion (Hoegh-Guldberg et al., 2007). Ancient reef crises linked to global climate change during the past 500 million years are marked in the geologic record by major decreases in carbonate accretion and reef volume (Kiessling and Simpson, 2011).

Numerous studies have modelled reef- to regional-scale accretion and erosion on coral reefs based solely on carbonate budgets some of which account for rates of bioerosion (eg., Stearn et al., 1977; Brock et al., 2006b; Moses et al., 2009; Kennedy et al., 2013; Leon and Woodroffe, 2013; Perry et al., 2013; Perry et al., 2014; Enochs et al., 2015; Perry et al., 2015). Recent studies have also measured regional-scale chemical erosion of carbonates due to dissolution caused by ocean acidification (Brock et al., 2006b; Moses et al., 2009; Muehllehner et al., 2016). Very few studies have quantified sediment

transport and export on reef systems (Hubbard, 1986, 1992; Hubbard et al. 1981, 1990; Kench and McLean, 2004; Morgan and Kench, 2014). Studies such as these are essential for attributing ecosystem change to cause, setting target levels for restoring and maintaining healthy reefs (Kennedy et al., 2013) and for measuring and predicting reef degradation due to these specific processes. However, no prior studies provide a comprehensive assessment of the combined effect of all of the processes affecting seafloor accretion and erosion in coral reef ecosystems, including, for example, physical erosion;

redistribution, import or export of seafloor sediments; compaction; direct human alterations to the seafloor; carbonate production; bioerosion; and chemical erosion. Such comprehensive assessments are essential for accurately quantifying and monitoring whole system accretion and erosion, identifying resilient and vulnerable seafloor habitats, and for identifying gaps in accounting for the individual processes causing ecosystem change. Additionally, development of hydrodynamic and other numerical models used to assess and predict the impact of reef degradation on the vulnerability of coastal communities

to hazards caused by storms, waves, sea level rise and erosion have been limited by lack of comprehensive assessments of change in the complex seafloor structure of coral reef ecosystems.

Recent advances in high-resolution bathymetric mapping capabilities such as Light Detection and Ranging (LiDAR) systems (Brock et al., 2002; Crane et al., 2004) provide the ability to map coral reef ecosystems with the spatial resolution needed to accurately measure elevation over their highly variable and complex topography. Combined analysis of high-resolution digital elevation models derived from LiDAR data and appropriate historical bathymetric data can provide comprehensive assessments of seafloor elevation and volume change to determine regional-scale and habitat-scale accretion and erosion in coral reef ecosystems. These types of regional-scale, elevation-change analyses can be used by numerical modellers to improve our ability to predict potential impacts to coastal communities due to degradation of coral reef ecosystems.

The aim of our study was to quantify regional-scale changes in seafloor elevation in coral reef ecosystems in the Atlantic (Upper and Lower Florida Keys), Caribbean (St. Thomas and Buck Island, U.S. Virgin Islands), and Pacific (Maui, Hawaii) including both coral-dominated habitats and adjacent, non-coral dominated habitats. Our study is based on the premise that elevation-change analysis measures the net result of all of the constructive processes that cause accretion (or increases in seafloor elevation) and the destructive processes that cause erosion (or decreases in seafloor elevation) on whole reef systems. This includes processes that have been broadly studied (e.g., carbonate production, biological erosion, chemical erosion, etc.), as well as processes that have not been well characterized (for example physical erosion and export). While this method does not attribute change to cause, it does provide a measure of the net result of all impacts (natural or anthropogenic) to seafloor structure in these ecosystems over the time periods studied. Furthermore, combining results from this method with habitat data helps identify potentially vulnerable and resilient habitats and provides some insight into relative contributions of broad scale processes that cause accretion, erosion, sediment redistribution and export. We used historical bathymetric data sets from the 1930's to 1980's and contemporary LiDAR-derived digital elevation models (DEMs) from the late 1990's to 2000's to calculate changes in seafloor elevation for each study site over the past few decades. We then created elevation-difference models from which we calculated corresponding changes in seafloor volume for whole study sites and for habitat types within each site. This type of elevation change analysis (using comparisons between historical and modern elevation data sets) has traditionally been used by coastal engineers to monitor seafloor changes such as sediment accretion and erosion, and migration of shipping channels, sand bars, mud banks and other seafloor features in coastal environments (Byrnes et al., 2002; Taylor and Purkis, 2012; Byrnes et al., 2013). To our knowledge, this study is the first to apply these methods to coral reef ecosystems. Here, we introduce the application of these seafloor elevation-change methods to whole coral reef ecosystems. We discuss elevation- and volume-change results in general context with known processes occurring in these systems to demonstrate the magnitude of change we observed and the value of this approach for regional-scale research and monitoring of impacts to reef ecosystems. Additionally, we report

substantial regional-scale net change in seafloor elevation that has occurred over the past several decades and its implications for sea level rise and risk from coastal hazards.

## 2 Methods and error analyses

Our study examined changes in seafloor elevation within 5 regions characterized by extensive coral reef ecosystems that
included reef flats as well as reef crest and slope habitats, adjacent (non-coral dominated) habitats such as seagrass beds, sand bottom, and hard bottom communities, and (in some cases) deeper water habitats. We chose to perform our analyses at the coral reef ecosystem scale because accretion in non-coral dominated habitats within these ecosystems, as well as off shore habitats (and beaches which are excluded from our study), are supported by sand/sediment production from the breakdown of carbonate produced by corals and other calcifying reef organisms. Few studies have quantified the effect of
reef degradation on accretion and erosion of non-coral dominated habitats at the coral reef ecosystem scale.

Our Florida Keys study site included outer reef track and surrounding habitats of the Upper Florida Keys (UFK) and Lower Florida Keys (LFK), and examined changes in elevation from 1934/1935 to 2002. Our St. Thomas, U.S. Virgin Islands study site (STT) included the entire southern coastline of the island including habitat out to 5 km offshore and examined changes
in elevation from 1966/1973 to 2014. Our Buck Island study site (BI) is an uninhabited location that included an extensive area east of the island and to a depth of 37 m, and examined changes in elevation from 1981/1982 to 2014. Our Maui, Hawaii study site included the coastal region surrounding the entire island to a depth of 20 m and examined changes in elevation from 1961/1965 to 1999. The size of each study area is provided in Table 1.

Long term monitoring data show that each of these coral reef ecosystems have experienced losses in live coral cover as well as changes in abundance of macroalgae and grazing reef bioeroders over the past couple of decades. Jackson et al. (2014) synthesized results from 19 different studies in the Florida Keys spanning the years 1973 to 2011, and 32 different studies in the U.S. Virgin Islands spanning the years 1973 to 2011, that collected benthic cover and herbivore data. These data show that, in general, live coral cover has decreased and macroalgal cover has increased in the UFK, LFK, STT, and St. Croix
(including Buck Island). *Diadema sp.* sea urchin densities have either decreased and/or remained absent at these study sites; and the biomass of parrotfish increased at in the UFK, LFK, and STT and decreased in St. Croix over these time periods (Table 1). Jackson et al. (2014) provide a detailed discussion of the major drivers of coral reef decline in the Caribbean including human population density, fishing, coastal pollution, ocean warming, invasive species, coral disease and the role of hurricanes. Sparks et al. (2015) performed benthic surveys of 10 areas along the western coast of Maui including 20 reef
sites as well as shallow (1 to 4 m) and deep (6 to 13 m) sites from 1999 to 2015. Results from these surveys indicate significant decreases in coral cover at 8 coral reef sites (average decrease ~12%), significant increases at 4 sites (average increase ~4%), no significant change at 8 sites, and an average decrease of ~4% for all study sites combined during this time

period. Macroalgal cover was monitored at 5 locations and showed substantial decreases at most sites. Parrotfish were monitored at 6 sites, showed no significant change at most sites, and significant increases at two sites located in protected areas (Table 1). Sparks et al. (2015) noted that declines in coral cover are likely due to sedimentation, land based pollution and overfishing, and that monitoring sites with stable coral cover are remote and less affected by urban and other anthropogenic factors.

The general concept of calculating the difference between corresponding modern and historical elevation data points is simple. However, the methodology used to validate and prepare data for analysis, determine error and quantify the effect of that error on results is complex. We, therefore, provide both a general discussion of the methodology (depicted in the flow-diagram in Fig. 1) for the broader scientific audience and a more technical discussion in Supplemental Methods Section 1 including step-by-step data-processing methods for those interested in pursuing similar analyses.

## 2.1 Selection and description of data sets

Our selection of study sites was based on the availability of historic hydrographic data that had sufficient point density and geospatial information to support meaningful comparisons with corresponding, more contemporary LiDAR data. Each of our five study sites was analysed independently over the specific time period indicated, using the oldest appropriate data available and the most recent data available. Our analysis was limited by the areal extent of available data sets. We note that there are gaps in the Florida Keys analysis where data was unavailable, including the Middle Florida Keys and near shore areas. There are also gaps in the Buck Island analysis in shallow areas surrounding the island and other areas where data was not collected. These gaps are identified in mapped results as white masked areas (Section 3.1). Full analysis of anthropogenic impact factors (e.g., urban extent, developed land, terrestrial water run-off, water quality, etc.) was beyond the scope of this study. However, our selected data sets spanned time periods during which human population approximately doubled and local anthropogenic impacts increased at each study site as well as at one uninhabited site (Table 2).

We caution that not all historical and modern data sets are appropriate for elevation change analyses. Careful consideration must be given to the original purpose of the surveys, the horizontal and vertical resolution of paired historical and modern data sets for comparison, and the horizontal resolution of data relative to the total size of the study area. We specifically chose data sets that were collected for the purpose of uniform regional-scale mapping of seafloor depth/elevation without bias toward specific seafloor features (e.g., only shallow areas, man-made features, etc.) to minimize skewing of the data point distribution. We performed multiple analyses to quantify and validate the horizontal and vertical error of the data sets (See Sections 2.11, 2.12, 2.13), as well as the effect of horizontal resolution and alignment error on our results (See Section 2.13). Coarse horizontal resolution of many historical data sets limits their use for elevation-change analyses to only large, regional-scale areas such as those addressed in this study (Section 2.13).

We used historical hydrographic survey (H-sheet) data collected by the United States Coast and Geodetic Survey (USC&GS) or the National Ocean Service (NOS) between 1934 and 1982. These historical data included digital XYZ sounding data for water depth as well the accompanying descriptive reports and smooth sheets (final map renderings), and were downloaded from the National Oceanic and Atmospheric Administration's (NOAA) Office of Coast Survey (NOAA Office of Coast Survey). The number of H-sheets used for each study site varied from 1 to 24 depending on geographic area. All data sources and descriptions are listed in Table 3. During these surveys several methods were used to collect depth soundings by USC&GS/NOS including lead line, graduated sounding-pole and fathometer, and methodology was noted in each H-sheet descriptive report along with measurement resolution information

In general, for the 1930's surveys, sounding poles were often used in water depths approximately less than 3 m and replaced with lead lines at greater depths. Sounding positions for 1930's data were determined by sextant triangulations using both permanent and temporary topographic reference points or triangulation stations, theodolite fixes, navigational beacons, and ship time and speed fixes (Shalowitz, 1964). Most of the surveys after 1960 were performed using a fathometer, and augmented with lead line soundings. The historical sounding data for Buck Island and Maui, collected using a fathometer, were reported to the nearest tenth of a fathom (0.18 m) for water depths less than 20 fathoms (37 m) and 11 fathoms (20 m), respectively, and to the nearest 1 fathom (1.8 m) for greater water depths. Therefore, we only used data from shallower depths than these thresholds (0.18 m resolution) at these study sites. Following field surveys, hydrographic data went through quality control protocols including adjustment for tides, removal of erroneous points, validation of track line cross-soundings, and determination of agreement with adjacent surveys to create a published smooth sheet (Shalowitz, 1964). The descriptive reports include notes from the USC&GS/NOS survey teams and details on any adjustments or changes made by reviewers. Hydrographic data used in our study that was downloaded from NOAA Office of Coast Survey had already been digitized by personnel at NOAA.

The U.S. Office of Coast Survey imparted strict standards on collection of historical hydrographic data from the 1800's to the 1950's that allowed for less error than later in the 1900's when electronic means of collecting sounding data replaced lead lining and poling (Shalowitz, 1964). For data collected by the Coast Survey from the late 1800's to 1950's, maximum allowable vertical differences at crossing data points was 0.06 m for depths less than 4.6 m, and 0.46 m for depths between 22 and 29 m (Byrnes et al., 2002). Lead line and poling vertical resolution reported for the historical data sets that we used was 0.15 m (similar to that of LiDAR), while fathometer resolution was 0.3 m. Furthermore, sounding poles were used in depths less than 3 m (where many shallow patch reefs are located). Many H-sheet descriptive reports indicated that the water was clear enough during surveys to see the bottom (which improves accuracy of the measurement). The most common error likely to occur during use of lead lines or sounding poles was overestimation of water depth due to angling of the line or pole as currents move the boat past the point of measurement. Overestimation of historical water depth would erroneously

decrease elevation losses calculated using our methods. Therefore, it is more likely that our erosion estimates are underestimated rather than overestimated due to lead line and poling techniques. We estimated horizontal error for historical data sets of approximately 5.0 m (See Section 2.13).

LiDAR data sets from the USGS Coastal and Marine Geology LiDAR Program and the U.S. Army Corps of Engineers (USACE) Joint Airborne LiDAR Bathymetry Technical Center of Expertise (JALBTCX) were downloaded for each study site (Table 3, U.S. Army Corps of Engineers-JALBTCX, 1999, 2004; Brock et al., 2006a; Brock et al., 2007; Fredericks et al., 2015a, b). LiDAR-derived digital elevation models (DEMs) for Atlantic/Caribbean and Pacific sites were at horizontal spatial resolutions of 1 m and 4 m, respectively, with horizontal datums of UTM NAD83, and vertical datums of NAVD88

GEOID03 for the Atlantic, VIVD09 for the Caribbean, and MLLW for the Pacific. Vertical accuracy of LiDAR data is typically reported as the 95% RMSE (Hodgson and Bresnahan, 2004). Vertical RMSEs for the LiDAR data used in this study ranged from 0.135 m to 0.15 m as reported in the metadata accompanying the LiDAR data for each site and based on independent validation by in-water acoustic-based depth measurements at the time of collection.

## 2.2 Preliminary inspection of historical data sets

We performed visual inspection of all of the historical (H-sheet) data relative to 2016 aerial and satellite imagery with a resolution of 1 m (2016 World Imagery from ArcGIS online via ArcMap 10.2.2, source: Esri, DigitalGlobe, GeoEye, Earthstar Geographics, CNES/Airbus DS, USDA, USGS, AEX, Getmapping, Aerogrid, IGN, IGP, swisstopo, and the GIS User Community) to measure differences between stable coastal and geographic features following the methods of Lukas (2014) for use of historical elevation data sets. Visual inspection of map features that have not changed over the study period

allows for confirmation of general horizontal alignment of the compared data sets. For example, in the Upper Florida Keys a historic bridge was surveyed in the 1930s and still existed at the time this study was performed. We overlaid the georectified 1934 H-sheet on 2016 georectified Worldview imagery in ArcMap 10.2.2 and used software-measuring tools to examine the offsets in this structure. The maximum offset of bridge boundaries between historical and modern locations was 4.8 m (Fig. 2). This offset is similar to the sum of errors reported in our horizontal error analysis in Section 2.13. We were able to

identify other stable, rocky coastlines and/or man made features (such as shoreline berms) for these types of comparisons at all study sites. In offshore areas, we examined locations in the historical data where the boundaries of large patch reefs were outlined by soundings when depths were too shallow for a boat to pass over the reef causing deviations from linear transect lines. There were no significant misalignments between historical and modern boundaries for reefs that experienced minimal erosion. We performed these visual inspections for all historical (H-sheet) data from all study sites and found similar

alignment agreement. Additionally, we visited select areas at all study sites to visually inspect the seafloor for erosion or accretion features that were consistent with trends identified by our analysis.

**2.3 Sea level rise correction of historical soundings**

The historical sounding data was corrected for sea level rise that occurred from the time of its collection to the time of collection of the corresponding LiDAR data. We adjusted the historical soundings using NOAA relative sea level rise (RSLR) estimates to account for any differences at local tide stations caused by sea level rise, platform movement, and/or subsidence (Parker, 1992; Byrnes et al., 2013). Long-term sea level rise data (in mm per year) recorded by NOAA sea level trend stations near our study sites were used to calculate the total sea level rise at each study site by multiplying the mean rate of annual sea level rise over the study time period (mm yr$^{-1}$) by the number of years between historical and modern data sets. These correction values ranged from approximately 7 cm to 16 cm and were added to the historical sounding value. NOAA reports 95% confidence intervals for these data ranging from approximately +/- 0.15 to +/- 0.81 mm/yr. The potential error from these corrections was insignificant relative to other sources of error, and we, therefore, excluded it from our RMSE calculations. Sea level trend stations, rates and correction factors are presented in Table 4.

**2.4 Conversion of historical soundings to elevations**

Vertical units for the original historical sounding measurements were converted from feet and fathoms to meters by the NOAA Office of Coast Survey, but remained in their original vertical datum (reference surface of zero elevation). We determined vertical datums for the digitized soundings by reviewing the H-sheet descriptive reports and smooth sheet notes. Historical sounding data were reported relative to tidal datums of mean low water (MLW) or mean lower low water (MLLW). Comparison between historical and modern data sets required transferring sounding data to the same vertical datum as the corresponding LiDAR data so elevations could be compared relative to a common zero elevation system. NOAA also transferred horizontal datums (coordinate systems for identifying physical locations) of the digitized XYZ historical hydrographic data to North American Datum 1983 (NAD83) during the digitization process. Similarly, historical data required transformation to the same horizontal datum as corresponding LiDAR data. We performed these data transformations using publicly available software from NOAA called VDatum v 3.6 that is available along with user guides and tutorials at https://vdatum.noaa.gov/. Geospatial and transformation information for these conversions are provided in Table 4, and a more detailed description of the transformation procedure is provided in Supplemental Methods Section 1.1.

**2.5 Extraction of modern seafloor elevation data at historical point locations**

The horizontal spacing of the historical sounding points was much coarser than the resolution of the LiDAR digital elevation models (1 to 4 m). Average historical sounding point densities for each whole study site were 109 pts km$^{-2}$, 89 pts km$^{-2}$, 422 pts km$^{-2}$, 268 pts km$^{-2}$, 326 pts km$^{-2}$ for the UFK, LFK, STT, BI, and Maui, respectively. Therefore, we used the locations of the historical soundings to extract modern elevation values from the LiDAR digital elevation models. This approach is more accurate than creating an interpolated model surface (or DEM) from the historical data and then determining elevation

change from two interpolated surfaces. Digital data files containing the XYZ coordinates for historical elevations were used with Spatial Analyst Tools in ArcMap 10.2.2 to extract LiDAR elevation values at the XY locations of historical soundings to create a single data file that contained the X and Y coordinates and the corresponding historical and LiDAR elevation values (see Supplemental Methods Section 1.2 for processing steps).

**2.6 Calculation of elevation change**

The data files created during the extraction process in Section 2.5 were used to calculate elevation changes ($Z_{change}$) by subtracting the historical elevation data from the corresponding modern elevation data via Eq. (1).

$$Z_{change} = \text{modern LiDAR elevation data point} - \text{historical elevation data point} \qquad (1)$$

Using this equation, negative values indicate loss of seafloor elevation and positive values indicate gain of seafloor elevation. The resulting data files for each study site are located in Tables S1, S2, S3, S4, and S5. These files contain data columns for the historical H-sheet source, year of historical data value, latitude and longitude in geographic and UTM coordinates, historical elevation values (m), modern LiDAR elevation values (m), and the difference in meters ($Z_{change}$) between historical and modern elevations, as well as vertical and horizontal datum information (see Supplemental Methods Section 1.3 for processing steps). These data sets were used for elevation change analysis of each study site including calculation of mean elevation change and analysis of elevation change by habitat type. We note that much reef degradation contributing to elevation change likely occurred after 1970 (Gardner et al., 2003; Bruno and Selig, 2007; Jackson et al, 2014). Therefore, data sets containing pre-1970's data (Table 3) could be biased toward lower annual elevation and volume-change rates.

**2.7 Calculation of elevation change statistics for study sites and habitat types**

We used the $XYZ_{change}$ (northing, easting and $Z_{change}$) data sets created in Section 2.5 for elevation analysis of each study site and of each habitat type within each study site. Mean elevation change and standard deviation were calculated for the total area of each study site using all $Z_{change}$ data points. We used benthic habitat maps for the Upper and Lower Florida Keys from the Florida Fish and Wildlife Conservation Commission (2015), and for the USVI and Maui from NOAA (Kendall, 2001a, b; Battista and Christensen, 2007) to delineate the boundaries of all habitat classes within each site. We calculated elevation-change statistics from the data points included within or on the boundary of each habitat including maximum elevation losses and gains, total number of elevation-loss data points, mean elevation loss and standard deviation, total number of elevation-gain data points, mean elevation gain and standard deviation. We also calculated elevation-change statistics for combined habitat classes within each study site that included only coral dominated substrate (e.g., scattered coral rock in unconsolidated sediment, aggregate reef, reef rubble, individual or aggregate patch reefs, spur and groove, etc.),

as well as for combined habitat classes that included only adjacent habitats (e.g., non-coral dominated substrate types such as sand, seagrass, macroalgae, pavement unconsolidated sediment, etc.). Coral-dominated substrate types within each study site that were combined for these calculations were denoted with an asterisk and adjacent habitats were indicated by the absence of an asterisk in data tables. We created an ArcMap 'model' to automate the $XYZ_{change}$ data-point processing for calculation of elevation-change statistics to ensure consistency for each habitat class (see Supplemental Methods Section 1.4 for data processing steps).

## 2.8 Elevation-change surface model

We created a single elevation-change surface model for each study site to estimate seafloor volume change, using the $XYZ_{change}$ data points calculated in Section 2.5. We used the Delaunay triangulation algorithm (in ArcGIS 10.2.2) to create a triangulated irregular network (TIN) from these data points. The TIN consists of a comprehensive, three-dimensional network of triangles spanning the entire data set. The three dimensional location of the vertices of each triangle was determined by using the XY data to establish its horizontal position and the $Z_{change}$ data to establish its vertical position in the network of triangles. TIN modelling allows for placement of nodes over irregular surfaces allowing for higher resolution over surface areas that are highly variable (such as coral reefs). Delaunay triangulation creates the triangle network such that no vertex lies within the interior of any of the circumcircles of the triangles in the network to avoid creation of long, thin triangles that can degrade surface resolution (http://pro.arcgis.com/en/pro-app/help/data/tin/tin-in-arcgis-pro.htm). Figure 3 shows examples of (a) the $XYZ_{change}$ data plot, (b) the three dimensional TIN model developed from the $XYZ_{change}$ data, (c) a sub-section of the TIN model, and (d) a horizontal cross-section view of the TIN model sub-section for the Upper Florida Keys study site. In Figs. 3b, c and d, the zero elevation-change plane lies between positive elevation-change values (denoted in blue tones) and negative elevation-change values (denoted in red tones). The gray areas in Fig. 3b and c and the dashed lines in Fig. 3d indicate all elevation-change values between +0.5 m and -0.5 m (90% RMSE). See Supplemental Methods Section 1.5 for data processing steps.

## 2.9 Calculation of volume change for study sites

We calculated volume changes (in units of millions of cubic meters, $Mm^3$) for each study site using the surface models created in Section 2.8. Maximum volume-change calculations included all elevation-change data. Minimum volume-change calculations excluded all elevation-change data within or equal to plus or minus 0.5 m (see Fig. 3 for Upper Florida Keys example). The 0.5 m threshold was derived from our vertical error analysis (see Section 2.11). We calculated volume-change for 4 cases based on placement of a reference plane within the surface model as follows:

1) gross accretion (maximum volume): plane height set to 0.0 m, volume calculated above the plane,

2) gross erosion (maximum volume): plane height set to 0.0 m, volume calculated below the plane,

3) gross accretion (minimum volume): plane height set to +0.5 m, volume calculated above the plane,

4) gross erosion (minimum volume): plane height set to -0.5 m, volume calculated below the plane.

For example, case 1 for the Upper Florida Keys included all of the elevation-change data above a 0 m plane that bisects the dashed lines in Fig. 3d, while case 2 included all of the data below that 0 m plane. Case 3 included all of the blue-toned data above the upper dashed line denoting the +0.5 m plane in Fig. 3d, while case 4 included all of the red-toned data below the lower dashed line denoting the -0.5 m plane. Maximum net volume changes ($Mm^3$) were calculated by summing the results of cases 1 and 2 for each study site. Minimum net volume changes ($Mm^3$) were calculated by summing the results of cases 3 and 4 for each study site. Area-normalized volume changes ($Mm^3/km^2$) were calculated by dividing net volume changes for each study site by its total area. See Supplemental Methods Section 1.6 for data processing steps. Seafloor accretion and erosion is a function of total mass balance in coral reef ecosystems (Schlager, 1981). However, we did not convert volume to mass of calcium carbonate because volume-change includes sediment compaction, loss due to mortality and degradation of framework-building coral colonies, and sediments of various porosities. Therefore, our volume changes are limited to approximations of mass changes.

**2.10 Calculation of volume change for habitat types**

We used the benthic habitat maps referenced in Section 2.7 (also used for elevation change calculations) to delineate habitat boundaries for calculation of volume changes for each habitat class within each study site. Habitat classes within each map were represented by individual polygons. We created new TIN models for each habitat polygon that only covered the areal extent of the specific habitat class polygon where it intersected the whole study site TIN. We calculated minimum and maximum volume changes for gross erosion and gross accretion in each habitat class using the 4 reference plane cases described in Section 2.9. We calculated volume changes for combined habitat classes within each study site including combined coral-dominated substrate and combined adjacent habitat using the same habitat combinations and procedures similar to those used for combined habitat elevation-change statistics described in Section 2.7 and using processing steps in Supplemental Section 1.7.

**2.11 Vertical error analysis**

U.S. Federal Standards were applied for all vertical and horizontal data transformations and error analyses. Our vertical error analysis included error terms for:

1) $RMSE_{Sounding}$ = uncertainty for each historical data set as determined from our analysis of repeat measurements that were performed by the original surveyors at the time of data collection,

2) RMSE$_{LiDAR}$ = uncertainty for modern LiDAR data sets that was determined by independent validation of airborne LiDAR measurements with in-water acoustic sounding measurements performed at the time that the LiDAR data was collected and reported in the metadata for these data sets, and

3) RMSE$_{VDatum}$ = uncertainty from transformation of data to a common vertical datum as calculated using VDatum version 3.6 (NOAA National Ocean Service, 2016) for each individual data set.

These uncertainty values (RMSE values) are specific to each data set and are reported in Table 5.

We determined RMSE$_{Sounding}$ for each historical data set from repeat sounding measurements collected by the original surveyors. When historical sounding data were collected, repeat surveys were performed along select transect lines and/or additional soundings were collected along intersecting transects for the purpose of determining vertical accuracy. Only maximum difference of repeat data points was reported as accuracy information in the historical data set descriptive reports. We, therefore, re-analysed all of the historical repeat and cross-track line survey data from the original historical data sets to perform a more rigorous, systematic vertical-error analysis (rather than simply using the reported maximum difference). We calculated the mean difference in depth between all pairs of repeat measurements within each historical data set, and used these data to calculate an RMSE$_{Sounding}$ for each data set using Eq. (2) whereby $d_{i,1}$ and $d_{i,2}$ were paired repeat depth measurements and $n$ = the number of sounding-point pairs used for each location (see Supplemental Methods Section 1.8 for procedures to identify repeat sounding measurements).

$$RMSE_{Sounding} = \sqrt{\frac{\sum_{i=1}^{n}\left(d_{i,1} - d_{i,2}\right)^2}{n}} \tag{2}$$

The number of sounding-point pairs used for each location was 40, 253, 40, and 51 for the UFK, STT, STC and Maui, respectively. The LFK was excluded from this analysis because no adjacent points were separated by 5 m or less. Implicit in this approach is the assumption that the points in each pairing represent two independent depth measurements at approximately the same location. Therefore, the mean difference in depth between all pairs should not be significantly different from 0 m. For each study site, we performed a 2-tailed $t$-test at the $\alpha = 0.05$ level on the set of depth differences between paired points to test the null hypothesis of a zero mean difference. $t$-tests were performed using Matlab v8.4.0 (R2014b). All of the RMSE$_{Sounding}$ values ranged from 13 to 32 cm, and are reported in Table 5 along with $t$-tests results.

The maximum cumulative uncertainty (MCU) reported by NOAA National Ocean Service (2016) for operational VDatum

regions of South Florida and the Virgin Islands are 9.6 cm and 11.8 cm, respectively. Our calculated $RMSE_{VDatum}$ for the UFK, LFK, STT and BI study sites fell within reported MCU values (Table 5). No vertical transformations were performed for the Maui data because both historical and modern elevation data were collected and reported relative to MLLW; therefore, $RMSE_{VDatum}$ was excluded from vertical error estimates for this study site.

We followed the general approach used by the U.S. Army Corps of Engineers (Byrnes et al., 2002) to calculate composite root-mean-square errors ($RMSE_{Total}$) for each study site  (Table 5) according to Eq. (3) to provide conservative estimates of elevation changes based on combined sources of uncertainty.

$$RMSE_{Total} = \sqrt{RMSE_{Sounding}^2 + RMSE_{LiDAR}^2 + RMSE_{VDatum}^2} \qquad (3)$$

Our average $RMSE_{Total}$ (Table 5) for all study sites was 0.29 m, and ranged from 0.20 m to 0.37 m. Our average $RMSE_{Total}$
(0.29 m) was similar to that reported for lidar and acoustic sounding data and half the reported value for previous studies that used nautical chart data to create a digital elevation model for examining the response of a reef to sea level rise (Leon et al., 2013; Hamylton et al., 2014). We then calculated a more conservative $RMSE_{Total}$ for each study site. We considered the $RMSE_{Total}$ values from each study as proxies for the standard deviations. Data within plus or minus one standard deviation of the mean encompasses approximately 68% of the variability in a normal distribution; and data within plus or minus 2
standard deviations of the mean encompasses approximately 95% of the variability. We used the Normal Inverse Cumulative Distribution Function to calculate that 90% of our elevation change values would occur within plus or minus 1.65 standard deviation of the mean. We then multiplied the $RMSE_{Total}$ values by 1.65 (denoted 1.65 x $RMSE_{Total}$) to capture approximately 90% of the variability in the depth differences. The average 1.65 x $RMSE_{Total}$ for all study sites was 0.48 m, which we rounded to 0.5 m and applied to all of our study sites for calculation of minimum volume changes.

**2.12 Independent vertical error validation through pavement analysis**

Within our study sites, the most stable areas were likely those classified as pavement or bedrock. We performed an analysis of these habitat types to independently estimate the potential vertical error associated with comparing historical and modern sounding data. We caution that this type of analysis does not constitute a true control because no subaqueous surface areas (including stable pavement or bedrock) are exempt from elevation change. For example, our data indicated areas where
pavement had been exposed likely due to erosion of overlying sediments, as well as areas where pavement had been buried likely by deposition of sediments or growth of corals, etc. However, areas were identified in each data set where little to no elevation change were also observed. We examined habitats labelled 'pavement' in the UFK and LFK, 'uncolonized

pavement' in STT, 'colonized pavement' in BI, and 'coral pavement' in Maui. We filtered the elevation-change data points for these habitats to include only elevation changes of less than +/- 0.5 m (1.65 x $RMSE_{Total}$). We then calculated the average difference and standard deviation between these points for each data set.

More than 50% of elevation-change points for pavement in the UFK, STT, and BI; 17% in the LFK; and 23% in Maui showed elevation changes within our 1.65 x $RMSE_{Total}$ of 0.5 m. Standard deviations for these locations were all within our $RMSE_{Total}$ of 0.29 m. Mean differences between historical and modern elevation data in these locations were very small (ranging from 3 to 6 cm, Table 6), indicating that vertical resolution and accuracy of historical and modern elevation data are comparable. Large standard deviations relative to the means and histograms showing skewed distribution of data points from

this analysis (Fig. 4) demonstrate that all areas of pavement are not stable and, therefore, should not be considered a true control.

**2.13 Horizontal error analysis**

No horizontal error was reported for any historical data sets. Our historical sounding data sets ranged from the 1930's to 1980's. Therefore, we determined the potential horizontal error associated with our oldest data sets from the 1930's in the

Florida Keys as an estimate of maximum horizontal error for all study sites.  The Florida Keys historical soundings were measured using the least accurate horizontal positioning methods and had average point densities that were more than 50% lower than the other study sites.

The original horizontal datum for data sets from the Florida Keys was North American Datum 1927 (NAD27).

Transformation of data from NAD27 to NAD83 using NOAA National Ocean Service program NADCON introduces no more that 15 cm of uncertainty in the continental United States (Dewhurst, 1990). The Annual Report of the U.S. Coast and Geodetic Survey 192 (1880) indicates that it was possible under normal controls to measure distances (using triangulation) with an accuracy of 1 meter, and the position of the plane table could be determined within 2 to 3 meters of its true position (Shalowitz, 1964). The horizontal resolution of LiDAR data is reported as better than 1m (Brock et al., 2006a and b; Brock

et al., 2007; Fredericks et al., 2015a, b). A sum of these estimated sources of error is approximately 5 m. We chose to sum these estimated sources of error to provide a more conservative error estimate rather than to calculate a horizontal RMSE. This error estimate is similar to the maximum offset of 4.8 m between the historical and modern measurements of a bridge structure located south of Lower Matecumbe Key in the UFK as depicted on a 1934 H-sheet and on the 2016 World Imagery identified during visual inspection of H-sheets described in Section 2.2.

Horizontal error of approximately 10 m is typically reported for data plotted at a 1:20,000 scale (Anders and Byrnes, 1991; Fletcher et al., 2003; Morton et al., 2004). Therefore, we performed a horizontal shift analysis to test the potential impact for

systematic plus random horizontal error of up to 10 m that represents an approximate doubling of our sum of estimated sources of error (5.0 m). Historical XYZ elevation data from the Upper and Lower Florida Keys were shifted by 10 meters to the north, south, east and west by adding 10 m to, and subtracting 10 m from, UTM northings and eastings for each data point (26,341 and 1,688 points for the Upper and Lower Florida Keys, respectively). This exercise generated 4 new, experimental historical XYZ elevation point-data sets that simulate up to 10 meters of horizontal error in each of 4 directions. The area of the un-shifted LiDAR DEMs for each site were clipped to each of the four shifted historical data sets generated for each study site, resulting in 4 new pairs of historical and modern elevation data sets for each study site Elevations from the LiDAR DEMs were then extracted at each historical point location for each paired historical and modern data set, and used to calculate elevation differences between the shifted historical soundings and un-shifted LiDAR data sets as described in Section 2.6. We created new TIN models from the shifted $XYZ_{change}$ point-data sets and calculated volume changes for each of these data sets using the TIN modelling process and surface volume analysis described in Sections 2.8 and 2.9.

Results from the horizontal shift analysis indicate that 10 m of horizontal error produces no more than 10% difference in net and area-normalized volume change calculations in the Upper Florida Keys, and up to a 21% difference in the Lower Florida Keys (Table 7) likely due to the lower density of data points over a smaller geographic area. The LFK had the lowest historical sounding point density of all data sets at 89 pts km$^{-2}$ and the smallest study site area of 19 km$^2$. Imparting 10 m of horizontal error in these experimental data sets does not change the general conclusion of our study indicating high magnitude of net erosion at these study sites. Our results are consistent with reports that over large areas (such as in our study), random errors largely cancel-out relative to change calculations derived from two surfaces (Byrnes et al., 2002).

## 3 Results and observations

### 3.1 Regional-scale loss of mean seafloor elevation and volume

Loss of seafloor elevation and volume, and transition from net accretion to net erosion, has occurred at all study sites (Table 8). Mean seafloor elevation losses for whole coral reef ecosystems in our study ranged from -0.09 m to -0.8 m. Note that mean elevation changes ('Mean elevation-change' columns in Tables 8 and 9) represent the net change encompassing all negative (elevation losses) and positive (elevation gains) values. Figures 5, 6 and 7 show the complex spatial distribution and large range of elevation gains and losses across the study sites. Individual measurements of elevation-change were often very large (greater than 1 meter, see minimum and maximum elevation loss and gain data by habitat in Table 9). Maximum elevation losses exceeded 4.0 m and maximum elevation gains exceeded 6.5 m at all study sites. Therefore, standard deviations for mean elevation changes were high and ranged from 0.7 m to 1.5 m for all sites.

Net volume losses were also observed in all regions indicating export of sediments from these systems (Table 1). Minimum volume changes (calculated by excluding elevation change data within a vertical range of ±0.5 m, see 1.65 x RMSE calculations in Table 5) provide conservative estimates of net volume losses ranging from 0.2 to 52.8 Mm³ for all study sites. Maximum volume changes (including all elevation-change data) indicate net volume losses ranging from 3.4 to 80.5 Mm³

for all study sites.

## 3.2 Changes in mean seafloor elevation and volume by habitat type

We analysed 118,710 elevation change data points from 60 habitats across all five study sites. Fifty-nine of these habitats showed net elevation change, 46 (77%) showed net elevation losses, and 13 (22%) showed net elevation gains (Table 9). The 'Mean loss' column in Table 9 (that includes only elevation loss values within habitats) indicates that only two habitats
showed mean losses less than our $RMSE_{Total}$ of 0.29 m (STT reef rubble and BI seagrass). Only 14 habitats showed mean losses less than our 1.65 x $RMSE_{Total}$ of 0. 5 m (that encompasses 90% of the variability). Therefore, mean losses in 97% of the habitats we analysed were greater than our vertical $RMSE_{Total}$ of 0.29 m, and 77% of the habitats we analysed showed mean losses greater than 1.65 x $RMSE_{Total}$ or 0.5 m. In several cases, and particularly associated with coral-dominated substrate, mean losses exceeded one meter. Conversely, if one examines only elevation gain values within habitats ('Mean
gain' column in Table 9), 92% of the habitats show showed mean gains greater than our $RMSE_{Total}$ of 0.29 m and 60% showed mean gains greater than our 1.65 x $RMSE_{Total}$ of 0.5 m.

Greatest mean elevation losses were generally associated with shallow, coral-dominated habitats (Tables 8 and 9) at all study sites consistent with observations of general flattening of reef topography (Alvarez-Filip et al., 2009) and observations of
decreasing abundance of reef-building corals (Gardner et al., 2003; Bruno and Selig, 2007; Jackson et al., 2014; Sparks et al., 2015). However, greatest net volume losses at most study sites were generally associated with adjacent (non-coral dominated) habitats despite smaller mean elevation losses. This apparent contradiction is due to the greater areal extent of these habitats (Tables 8 and 10). Maximum values of area-normalized volume change calculated using all elevation change data (Table 8) indicate that volume loss in adjacent habitats was 40% to 100% of losses observed in coral-dominated
substrate. We calculated the percent of each study site classified as coral-dominated substrate to estimate the potential contribution of loss of framework-building coral species to total net volume loss (coral-dominated habitat classes were denoted in Tables 9 and 10 with an asterisk). For example, the Upper Florida Keys coral-dominated habitat classes included scattered coral rock in unconsolidated sediment, aggregate reef, reef rubble, individual or aggregate patch reefs, and spur and groove habitat. The other study sites included similar lists of coral-dominated habitats. We included reef rubble as a coral-
dominated substrate in our calculations because, for example, in the Florida Keys many areas of reef rubble contain large skeletal fragments of *Acropora palmata* and other coral species, and we note that framework-building coral species such as *Acropora sp*. declined significantly throughout the Caribbean and Western Atlantic during the time periods of our study

(Jackson et al, 2014). Areal extent of coral-dominated substrate was only 8% to 15% of the total study area in the UFK, LFK, and STT study sites, and contributed up to only 26% of the total net volume loss. Most of the volume loss in these areas was associated with sediment loss from non-coral dominated (adjacent) habitats such as areas of unconsolidated sediments, seagrass, and uncolonized pavement. Areal extent of coral-dominated substrate was higher at the Maui study site (57%) and contributed more than 50% to net volume loss; and adjacent habitats accounted for the remaining volume loss. The considerable loss of seafloor volume in adjacent habitats at these study sites as well as in coral-dominated habitats, and large total net volume losses suggests that physical erosion and export of sediments is a likely driver of much of the volume loss we observed.

The BI study site had the highest areal extent of coral-dominated substrate (91%) that contributed more than 90% to the total net volume loss suggesting that degradation of framework-building corals may be the primary contributor to volume loss at this study site. Recent photographic surveys (C. Storlazzi, USGS, 2015) along the eastern coast of Buck Island show large stands of *Acropora palmata* coral colonies that have died and toppled over, but remain largely in tact and in place on the seafloor along the northeastern side of the island from approximately the eastern-most point to at least the mid-point of the island (Fig. 8a, b, and c), consistent with our observation of greater elevation loss in that area (Fig. 6c). The southeastern side of the island, from the eastern-most point to at least the barrier break, was characterized by much more live *A. palmata* coral (Fig. 8d, e, and f), also consistent with our observation of increased elevation in that area (Fig. 6c). These photos illustrate how loss of topographic complexity (as described by Alvarez-Fillip et al., 2009) can cause elevation and volume loss without material export. The live coral colonies clearly show relatively high elevation and colony volume that consists of both coral branches as well as very large open (pore) spaces between the branches. As large colonies topple, the coral branches break and the colony compacts causing a loss in elevation and volume as the open spaces between the branches are minimized. Large coral fragments are much harder to transport than sediments, typically require high-energy storm events for movement, and are, thus, not as easily exported from the system. Additionally, coral rubble-fields generated from toppling corals can create a rubble pavement on top of sand making it more difficult to transport. Mean elevation and volume losses were much lower at the BI study site than at the other study sites indicating that much less sediment was exported from the system, consistent with our observations of large areas of physically-toppled coral colonies that remain mostly in place on the seafloor. This type of physical coral degradation is very different from degradation of reef structure due to bioerosion by reef grazers such as parrotfish and *Diadema* sp. sea urchins that directly reduce coral and reef structure to sand-sized sediments that can then be more easily transported and exported by physical processes, and from chemical bioerosion caused by some species of sponges and endolithic algae that dissolve carbonate material (Decarlo et al., 2015; Enochs et al., 2015; Reyes-Nivia et al., 2013; Wisshak et al., 2013; Wisshak et al., 2012; Tribollet et al., 2009). As population-abundances of bioeroders increase and decrease in a given location, so do rates of bioerosion caused by these organisms (Mumby, 2009). Populations of grazing bioeroders have decreased at the BI study site and throughout most of the Caribbean during the time periods of our study (Table 1) that may account for the lower export rates at this study site because coral and coral rubble

may be reduced to sand at a slower rate. Visual inspection of study sites in the Florida Keys shows similar evidence for physical toppling of corals in rubble fields. However, parrotfish populations recently increased in the Florida Keys (Table 1) and rates of chemical bioerosion have increased since pre-industrial times (Enochs et al., 2015) which may help expedite breakdown and physical transport of reef materials out of the system, consistent with the higher volume losses observed in

the Florida Keys.

Along the Florida Keys reef tract within Biscayne National Park, Florida and the Florida Keys National Marine Sanctuary, mean elevation and volume losses occurred in 9 of 11 habitat classes in the Upper Florida Keys (UFK) reef tract (Fig. 5a, b) and in 6 of 9 classes in the Lower Florida Keys (LFK) reef tract (Fig. 5c, d) including coral habitats, and adjacent habitats

such as unconsolidated sediments, pavement previously covered by sediments or coral, and seagrass. Largest mean elevation losses occurred at shallow patch and aggregate reefs, coral-dominated and reef rubble habitats, consistent with documented declines in abundance of large framework-building corals over the past several decades (Jackson et al., 2014). Largest net volume losses occurred in seagrass and unconsolidated sediment habitats. Mean elevation and volume gains occurred in deep water habitats of the UFK and LFK sites including offshore aggregate reefs in the LFK near a sanctuary preserve area, at the

base of spur-and-groove habitat along fore-reef slopes, and where relic spur-and-groove formations in-filled with sediments indicating transport of reef sediments down the fore-reef slope and export offshore (Fig. 9). Mean total elevation loss was lowest at the UFK study site. However, mean elevation losses decreased from upper (-0.4 m) to central (-0.3 m) sub-regions of the UFK, and mean elevation increased slightly in the lower sub-region (0.1 m) primarily associated with seagrass habitat (Fig. 5a, b). Notably, the lower sub-region is further away from high-density population areas north of the study site and near

an area of the middle Florida Keys identified as a possible refuge from ocean acidification due to seagrass productivity (Manzello et al., 2012). The LFK site included the Looe Key National Marine Sanctuary and Special Protection Area (Fig. 5c). Net elevation and volume losses were similar to the central sub-region of the UFK possibly due to close proximity to high-density population areas in the Lower Florida Keys. However, the LFK was the only study site that showed net accretion in combined coral habitat classes. Accretion is partially attributed to deep reef and spur-and-groove habitat that has

been buried by sand (Lidz et al., 2007). Redistribution of reef materials and erosion of the Florida reef tract is corroborated by field observations of movement and deposition of meters-thick sand deposits likely due to hurricanes and by exposure of older reef material from erosion and transport (Shinn et al., 2003). However, accretion was also observed on offshore aggregate reef habitat near the Special Protection Area.

Mean elevation and volume loss occurred in 16 of 17 habitat classes in STT (Fig. 6a, b). In STT, small, localized areas of coral-dominated substrate showed gains in elevation. However, greatest mean elevation losses occurred in coral-dominated habitats and near the central coastline where harbour and shipping channels exist.  Mean elevation and volume gains only occurred on un-colonized bedrock possibly due to transport and deposition of sediments. However, uninhabited Buck Island National Reef Monument and Marine Park, St. Croix USVI (BI) showed less mean elevation and volume loss than all other

sites with losses limited to 4 of 11 habitat classes including 3 coral habitats (Fig. 6c, d). The study period for BI is approximately 10 years shorter than for nearby STT. Greatest mean elevation gains in BI occurred along linear reef near the shoreline. Most elevation and volume gains occurred within Buck Island National Reef Monument and Marine Park boundaries (Fig. 6c) suggesting that distance from populated areas combined with managed protection has limited losses at

this site.

Mean elevation and volume loss occurred in 10 of 12 classes in Maui (Fig. 7a, b). Mean elevation and area-normalized volume losses were at least 2 to 3 times greater, respectively, than all other study sites and occurred over a shorter time-frame. These greater losses may be caused by higher sediment export rates due to a combination of higher wave energy,

physical erosion and a narrow shallow shelf surrounding the island allowing sediment to be more easily transported offshore into deep water, as has been observed in other high energy reef environments (Morgan and Kench, 2014; Perry et al., 2015). Greatest mean elevation losses were associated with coral-dominated habitat as well as 'not classified' habitat located in the bank/shelf zone that may be impacted by slumping of materials along steeply sloping areas. Elevation and volume gains occurred in mud and rubble habitats, and may be associated with terrigenous sediment transport from the island.

Examination of histograms for pavement analysis data support the general observations for each of our study sites. Histograms for the UFK, LFK, STT and Maui (Fig. 4a, b, c, e) are all skewed toward negative values, consistent with regional-scale trends of seafloor elevation loss and export of sediments at these sites. While BI (Fig. 4d) showed a more even distribution of points among elevation losses and gains consistent with the lower regional-scale mean elevation losses

observed at this study site, and suggestive of redistribution and less export of sediments within the system. Notably, of the five study sites we examined, the 'colonized pavement' in BI was the only pavement habitat to show a very small increase in elevation that could be due either to coral growth on the pavement or accumulation of sediments.

**4 Discussion**

Our results include elevation and volume changes caused by chronic erosion and accretion processes that occur slowly over

time frames of months to decades such as changes in carbonate production rates, bioerosion, chemical erosion from carbonate dissolution, degradation of large framework building coral colonies, and physical movement of reef sediments due to persistent oceanographic conditions such as waves and currents. Our results also include changes caused by episodic events that occur over very short time frames of minutes to days and often cause large changes in elevation that alter habitat distribution and affect process modelling for coastal hazards. Examples include dredging and infilling of channels and

coastal harbours, deposition of terrigenous materials from landslides and run-off, slumping and relocation of seafloor materials at steeply sloping locations, storm erosion and deposits. These multiple processes cause direct changes in seafloor accretion and erosion; and multiple factors drive changes in the rates of these processes (including, for example, impacts

from coastal development, overfishing, nutrient enrichment, coral bleaching and disease, climate change, ocean acidification, cultural and economic resource values, etc). Physical changes in seafloor structure across reef ecosystems and its causes are highly variable due to the large spatial and temporal variability of these processes, the complex interactions among processes within and among adjacent habitats, and uneven responses within and between species and habitats to multiple local,

regional and global impacts. Accurately measuring these physical changes are difficult due to the diversity and complexity of the resulting habitat structure within a coral reef ecosystem. The complex spatial distribution and large range of elevation gains and losses we observed across our study sites, as well as our assessments of error, indicate that our methods provide an effective means for measuring the changing dynamics of accretion and erosion at whole ecosystem scales with high enough spatial resolution to capture changes that reflect the diverse responses of accretion and erosion processes to multiple driving

factors. Our results can, therefore, be used to measure changes in whole system accretion and erosion required to determine regional-scale impacts from sea level rise. Additionally, our methods can be used for characterizing changes in the complex seafloor structure of coral reef ecosystems required for development of hydrodynamic and other numerical models for assessing and predicting the impact of reef degradation on the vulnerability of coastal communities to hazards caused by storms, waves, sea level rise and erosion.

More difficult is the attribution of change to cause because physical change in seafloor structure represents the integrated, and sometimes delayed, result of many combined processes of and impacts to carbonate production, accretion and erosion. Carbonate production and erosion rates developed from small-scale and/or short-term process studies provide local snap-shots in time that may not capture the variability of these rates in the whole system and, therefore, may skew carbonate

budget results when extrapolated to larger areas or longer time periods. Integrative process-studies conducted over large areas can indicate the net response of a system, but often do not accurately capture the contribution from individual processes in a system required to attribute change to cause. Furthermore, our knowledge of multi-stressor impacts to carbonate production and erosion is emerging, but largely limited to experimental results from laboratory or mesocosm experiments that are difficult to extrapolate to whole ecosystem scales. Many geological studies of reef systems measure

long-term changes in carbonate accretion, erosion and carbonate budgets. However, there is a disconnect between the temporal resolution of geologic studies (decades to millions of years) and the temporal resolution of most process studies (from minutes to decades) that makes it difficult to attribute change in the geologic record to contemporary processes over time periods that are informative for management decisions.

We emphasize that a key limitation of our elevation-change analyses is that they do not attribute change to cause. Detailed analysis of the processes causing elevation-change in these systems is beyond the scope of this paper and should be undertaken in future studies. However, we provide general comparisons of our elevation and volume-loss measurements to individual process rates from the literature to provide context for the magnitude of our results. These comparisons also demonstrate the potential value of knowing net whole system change when accounting for and identifying individual

processes, for identifying missing gaps in erosion/accretion budgets, and serve as an example of how our results may be used in future studies to improve erosion and accretion budgets in coral reef ecosystems.

Over 90% of the habitats we analysed in our study showed statistically-significant elevation changes with a net result of elevation loss. Accretion of a reef ecosystem over time occurs when the balance of accumulation of reef materials and sediments (that increases material elevation and volume) exceeds erosion and loss of the eroded material from the system (that decreases materials elevation and volume). Our observations of mean seafloor elevation and volume loss indicate that more materials are being eroded and exported from these coral reef systems than are accumulating. Erosion of both coral-dominated substrate and non-coral substrate suggests that the current rate of carbonate production is no longer sufficient to support net accretion of coral reefs or adjacent habitats. Calculation of annual mean seafloor elevation losses for each study site from our total mean elevation change values in Table 8 (column 3) and the historical-to-modern time periods of data sets for each site, Eqs. (3 - 7), shows that mean seafloor elevation decreased by -1.5 to -6.3 mm yr$^{-1}$ over 33 to 68 years at Atlantic and Caribbean sites, and by -21.1 mm yr$^{-1}$ over 38 years at the Pacific site (Fig. 10).

$$UFK = -0.1 \text{ m} / 68 \text{ years} = -1.5 \text{ mm yr}^{-1} \quad (-5.9 \text{ mm yr}^{-1} \text{ upper sub-region, } -4.4 \text{ mm yr}^{-1} \text{ central sub-region}) \quad (3)$$
$$LFK = -0.3 \text{ m} / 66 \text{ years} = -4.5 \text{ mm yr}^{-1} \quad (4)$$
$$STT = -0.3 \text{ m} / 48 \text{ years} = -6.3 \text{ mm yr}^{-1} \quad (5)$$
$$BI = -0.09 \text{ m} / 33 \text{ years} = -2.7 \text{ mm yr}^{-1} \quad (6)$$
$$Maui = -0.8 \text{ m} / 38 \text{ years} = -21.0 \text{ mm yr}^{-1} \quad (7)$$

To estimate how many years of reef accretion may have been lost due to erosion, we divided our total mean elevation-losses (mm) from Table 7 (column 3) by published average Holocene reef accretion rates of 2.6 and 10 mm yr$^{-1}$ for Caribbean/Atlantic and Pacific reefs, respectively, in Eqs. (8 – 12) (Shinn et al., 1977; Buddemeier and Smith, 1988).

$$UFK = -0.1 \text{ m or } 100 \text{ mm} / 2.6 \text{ mm yr}^{-1} = 38 \text{ yrs} \quad (8)$$
$$LFK = -0.3 \text{ m or } 300 \text{ mm} / 2.6 \text{ mm yr}^{-1} = 115 \text{ yrs} \quad (9)$$
$$STT = -0.3 \text{ m or } 300 \text{ mm} / 2.6 \text{ mm yr}^{-1} = 115 \text{ yrs} \quad (10)$$
$$BI = -0.09 \text{ m or } 90 \text{ mm} / 2.6 \text{ mm yr}^{-1} = 35 \text{ yrs} \quad (11)$$
$$Maui = -0.8 \text{ m or } 800 \text{ mm} / 10 \text{ mm yr}^{-1} = 80 \text{ yrs} \quad (12)$$

These calculations indicate that total mean elevation losses that occurred over 33 to 68 years at our study sites represent the loss of approximately 35 to 115 years of reef accretion. These coral reef ecosystems have lost pace with historical global mean sea level rise of 1.7 mm yr$^{-1}$ between 1901 and 2010, and will be unable to keep up or catch up with current or projected rates of 3.2 mm yr$^{-1}$ (1993-2012) and 4.5 mm yr$^{-1}$ (mid-century, RCP4.5), respectively (Church et al., 2013).

Perry et al. (2013) estimated modern reef accretion and erosion rates on 19 Caribbean reefs based on carbonate production and bioerosion, and reported a maximum erosion rate of -1.17 mm yr$^{-1}$. This maximum erosion rate could account for only 20% and 27% of erosion observed in the upper and lower sub-regions of the UFK, respectively; for only 26% of erosion observed in the LFK, 43% at BI, and 19% at STT. Such a comparison illustrates that this maximum bioerosion rate (1) cannot fully account for erosion loss at these study sites; (2) estimates the amount of erosion that is unaccounted for in these examples; and (3) identifies missing process gaps in that analysis. Similarly, recent regional-scale measurements of carbonate production and dissolution in the Florida Reef Tract show that dissolution alone could account for chemical erosion of up to approximately -0.7 mm yr$^{-1}$ in the northernmost reef tract (Muehllehner, 2016), or approximately 12% and 16% of the erosion we observed in the upper and central sub-regions of the UFK study site, respectively. Applying combined maximum estimates for erosion considering bioerosion (Perry et al., 2013) and chemical dissolution (Muehllehner, 2016) could account for only 32% to 43% of erosion in these sub-regions of the UFK. Enochs et al. (2015) developed a carbonate budget model of 37 reefs along the Florida Reef Tract that accounted for bioerosion by boring sponges, endolithic algae, and parrotfish as well as calcification by coral and crustose coralline algae. Eighty-nine percent of the reefs in their study showed net erosion, consistent with our observation of net erosion at the Florida Keys study sites. Their erosion rates for the Upper Keys, and Lower Keys were -1.474 kg m$^{-2}$ yr$^{-1}$ (or 0.00092 m$^3$ m$^{-2}$ yr$^{-1}$) and -1.556 kg m$^{-2}$ yr$^{-1}$ (or 0.00097 m$^3$ m$^{-2}$ yr$^{-1}$), respectively (assuming a bulk density of 1600 kg CaCO$_3$ m$^{-3}$ from Hubbard (1992) for volume calculations). Our area-normalized, annual volume change (calculated from maximum net volume change in Table 8) for the UFK and LFK was 0.002 m$^3$ m$^{-2}$ yr$^{-1}$ and 0.004 m$^3$ m$^{-2}$ yr$^{-1}$, respectively. Enochs et al. (2015) erosion rates could account for 24% to 46% of the erosion we observed at the Florida Keys study sites. Harney and Fletcher (2003) determined sediment production from reef bioerosion rates for Kailua Bay, Oahu, Hawaii of 0.33 kg m$^{-2}$ yr$^{-1}$ (or 0.0002 m$^3$ m$^{-2}$ yr$^{-1}$, assuming a bulk density from Harney and Fletcher (2003) of 1480 kg CaCO$_3$ m$^{-3}$). Export of all bioeroded sediment based on their sediment production rate could account for less than 1% of our area normalized, annual volume change for Maui of 0.03 m$^3$ m$^{-2}$ yr$^{-1}$. In these examples, results indicate that much of the erosion we observed at our study sites could be attributed to causes other than export of bioeroded sediments such as physical erosion and export of sediments that has been largely unaccounted for in these systems.

Previous studies indicate that, in some cases, higher sea level will facilitate production and transport of unconsolidated sediments shoreward, and growth of reef islands (Hopley and Kinsey, 1988; Hopley, 1992; Kench et al., 2014). Recent studies on 146 reef islands on 12 atolls in the central and western Pacific show that 106 of these islands have remained stable, while 40 have increased in size, and only 12 have decreased in size over the past few decades when sea level increased three to four times more than the global average (McLean and Kench, 2015). Kench et al. (2015) performed a detailed analysis of shoreline change on 29 islands of the Funafuti Atoll in the central, tropical Pacific. Their results showed no evidence of erosion with increasing sea level during the past 50 years, and that many of those islands have increased in

size. Kench et al., (2015) also noted that stable and growing islands have a more erosion resistant island core (e.g. gravel islands) and periodic pulses of new sediment input, while those islands that have decreased in area have lost original island core material that is often less resilient (e.g., sand islands), and have no new sediment input. Our Maui study site showed the greatest volume loss over the shortest period of time (Table 8). Although Maui is a volcanic island, the coastal plains are composed primarily of late middle to late Holocene carbonate sands (Fletcher, et al., 2003). Studies of beach erosion along the Maui coastline show that beach width narrowed by 19% island-wide from 1949/50 to 1997/2002 due to erosion and alongshore transport of sand; and radiocarbon dating showed that no modern sand is transported onshore from the reefs (Fletcher et al., 2003). To our knowledge, no comprehensive studies that quantify seafloor sediment export have been undertaken in Maui. However, studies on shallow, fringing reefs along the west coast of Maui and south coast of Molokai, Hawaii indicate that strong currents produced by trade winds cause daily sediment resuspension and transport primarily alongshore, and offshore-directed currents and ebbing tides cause offshore transport of suspended sediments (Ogston et al., 2004; Storlazzi et al., 2004; Presto et al., 2006; Storlazzi et al., 2008). Large, high-energy winter and storm waves cause strong offshore transport of suspended material, and terrigenous sediment from the shore is also primarily transported offshore across the reef flat and fore reef (Storlazzi et al., 2008). Our results indicate net export of sediments from the coastal seafloor of Maui and support these previous observations that reef and terrigenous sediment is transported offshore and no new sediment from the reef is contributed to coastal beaches.

Sediment transport measurements in other coral reef ecosystems show large amounts of sediment export primarily due to physical oceanographic processes during both fair-weather (persistent) and storm conditions (Hubbard et al., 1981; Hubbard, 1986; Hubbard et al., 1990; Hubbard, 1992; Kench and McLean, 2004; Morgan and Kench, 2014), and suggest that physical erosion and export of sediments could account for much of the volume loss we observed at our study sites. We estimated sediment volumes for studies that reported mass (kg or metric tons) of sediment export based on a bulk density for reef sediment of 1600 kg m$^{-3}$ (Hubbard, 1992) for comparative purposes; however, bulk densities are likely variable from location to location (e.g., Harney and Fletcher, 2003). For example, several studies on reefs around the island of St. Croix, U.S. Virgin Islands show that wave-induced oscillatory and unidirectional currents are the dominant process causing physical transport of sediments offshore, particularly during storms (Hubbard et al., 1981); and that annual storms can increase sediment transport by an order of magnitude higher than occurs during fair weather conditions (Hubbard,1986). Short-term sediment transport measurements from 15 locations around St. Croix (including two sites near Buck Island) showed transport rates ranging from 14 to 437 kg m$^2$ yr$^{-1}$ (~0.009 to 0.3 m$^3$ m$^{-2}$ yr$^{-1}$) during non-storm conditions, and 146 to 2431 kg m$^2$ yr$^{-1}$ (~0.09 to 1.5 m$^3$ m$^{-2}$ yr$^{-1}$) during storm conditions (Hubbard et al., 1981). Results from studies that incorporated carbonate production, bioerosion, reef accretion and sediment transport measurements in an open reef system in Cane Bay along the north shore of St. Croix showed average sediment export rates of 153 kg m$^2$ yr$^{-1}$ (~0.1 m$^3$ m$^{-2}$ yr$^{-1}$) considering both fair weather and storm export rates (Hubbard et al., 1990). Fair weather export rates from Cane Bay and nearby Salt River Bay (St. Croix, USVI) are 33 kg day$^{-1}$ (~0.02 m$^3$ day$^{-1}$) and 64 kg day$^{-1}$ (~0.04 m$^3$ day$^{-1}$), respectively; but

increased to 440 kg day$^{-1}$ (~0.3 m$^3$ day$^{-1}$) and 1115 kg day$^{-1}$ (~0.7 m$^3$ day$^{-1}$), respectively, during normal heavy weather that occurs several times a year (Hubbard 1992). However, passage of Hurricane Hugo in 1992, caused export of 336 metric tons (~210 m$^3$) and 2000 metric tons (~1250 m$^3$) of sediment from Salt River Canyon and Cane Bay, respectively, in only 4 to 6 hours (Hubbard 1992). Based on our maximum net volume change values (Table 8), our results indicate annual volume loss

ranged from up to 103,030 m$^3$ yr$^{-1}$ (at BI) to 2,118,421 m$^3$ yr$^{-1}$ (Maui). Area-normalized, annual volume loss ranged from 0.002 m$^3$ m$^{-2}$ yr$^{-1}$ to 0.03 m$^3$ m$^{-2}$ yr$^{-1}$, and was lower than the observed ranges in the St. Croix studies (Hubbard et al., 1981; Hubbard et al., 1990). Our lower values may result because our measurements reflect integration of variable short- and long-term processes over much longer time periods (decades) and much larger geographic areas that include transport and redistribution of sediments within these systems as well erosion and export. Our area-normalized, annual volume losses were

much higher those measured during sediment flux studies on a 3.23 km$^2$ Maldivian reef platform system that showed an annual sediment export rate of 127,120 kg yr$^{-1}$ (~2 x 10$^{-5}$ m$^3$ yr$^{-1}$, Morgan and Kench, 2014), and in eleven hoas of reef systems in the Keeling Islands (Indian Ocean) that showed annual sediment flux rates ranging from 44 metric tons per year (~28 m$^3$ yr$^{-1}$) to 223 metric tons per year (~139 m$^3$ yr$^{-1}$) (Kench and McLean, 2004). However, these studies were performed over very small areas and very short time periods (108 days to 13 months). Our long-term sediment export rates fall within

the range of the few studies that quantified short-term sediment flux and export rates in other reef systems.

Studies of ancient reefs indicate that many were able to keep up or catch up with sea level rise (Neumann and Macintyre, 1985). However, ancient reefs that were exposed to local, regional and global environmental stresses were unable to keep up and drowned (Schlager, 1981; Hubbard, 1997; Kiessling and Simpson, 2011). Recent projections of reef response to sea

level rise in the northern Great Barrier Reef using carbonate production rates primarily from the 1970's indicate that shallower reef flats could become colonized by corals with a rise in sea level of 0.5 m, but begin to drown after 30 years with a rise of 1.2 m (Hamylton et al., 2014). However, these projections do not consider reduced rates of calcification due to local and global stressors and loss of carbonates from erosion processes. A recent study by Perry et al. (2015) showed that while remote coral reefs (such as the Chagos Archipelago reef systems in the Indian Ocean) that are largely isolated from human

influence experience severe coral mortality from climate driven impacts like coral bleaching, most of these reefs recover very rapidly and continue to produce enough carbonate to keep up with present and future sea level rise. While our Maui study site is remote (geographically isolated), it is not isolated from human influence; and our results showed large erosion rates at this site suggesting that these reefs systems have not recovered well from impacts/degradation despite geographic isolation. We observed lower erosion rates in reef ecosystems (and net accretion in some localized areas in the UFK and BI)

that were managed, distant from human population centers, or associated with natural refuge zones. However, these lower rates did not prevent net regional-scale losses of seafloor elevation and volume at our study sites. Vertical accretion rates vary from site to site within a given region, some localized areas may continue accreting, and other coral reef ecosystems may be able to maintain pace with sea level rise (Hopley and Kinsey., 1988; Hopley, 1992; Kench et al., 2014; McLean and Kench, 2015; Kench et al., 2015). However, modern carbonate production rates are an order of magnitude lower than

Holocene averages (Perry et al., 2013), and are estimated to decrease by as much as 60% by mid-century (Langdon and Atkinson, 2005). Bioerosion and chemical dissolution of carbonates are projected to increase with ocean acidification (Hoegh-Guldberg et al., 2007; Eyre et al., 2014; Enochs et al., 2015). Therefore, reef erosion rates are likely to accelerate over the coming decades.

The magnitude of regional-scale erosion we observed in these reef systems (Table 8) following world-wide declines and transitions in reef species composition, combined with projections for continued global reef degradation suggests the onset in the geologic record of an Anthropocene reef crisis as marked by significant reef volume loss in ancient reef systems (Kiessling and Simpson, 2011). As an example, we calculated the amount of the total Holocene reef deposit at the LFK site that has been lost since 1938 using geophysical data sets from Lidz (2000) and Lidz et al. (2007). The geologic history and evolution of the Florida Keys reef tract has been well characterized (e.g., Lidz 2008). The Florida Keys Holocene reef deposit lies on top of Pleistocene carbonate bedrock. The formation of late Pleistocene bedrock began more than 125,000 years ago when South Florida was submerged during a sea level high stand. During this time period carbonate bedrock production resulted from the formation and accretion of coral reefs and carbonate sediments. The development of Pleistocene bedrock was marked by alternating periods of submersion and sub-aerial exposure as sea level fluctuated from approximately 125,000 to 10,000 years ago; and the Holocene reef of South Florida began to form as the Florida shelf flooded when the most recent rise in sea level began approximately 10,000 years ago (Lidz et al. 2008). The Holocene reef of the Lower Florida Keys began forming approximately 6000 to 7000 years ago and lies on top of Pleistocene bedrock that was exposed in this area prior to 7000 years ago (Lidz et al. 1985). Seismic profiles were collected throughout the Florida Keys from 1997 through the early 2000's to measure the depth below the seafloor of the Pleistocene bedrock surface (Lidz et al. 2007); and a depth-to-Pleistocene-bedrock (DPB) surface was constructed from these seismic data and cores along the Florida Keys reef tract (available from the U.S. Geological Survey at http://pubs.usgs.gov/pp/2007/1751/data/PP1751.zip). The difference between the elevation of the seafloor surface and the elevation of the Pleistocene bedrock surface below the seafloor estimates sediment thickness of the Holocene reef deposit along the Florida Keys reef tract.

We estimated the sediment volume of the Holocene reef and sediment deposit that existed at the Lower Florida Keys study site in 1938 by calculating the sediment thickness between the elevation of the Pleistocene bedrock surface and the elevation of the 1938 seafloor surface. We created a DEM for the Lower Florida Keys historical hydrographic data from 1938 using the Delaunay triangulation method described in Section 2.8. We transformed the horizontal datum of the DPB surface map to match that of the 1938 DEM then clipped the area of the DPB surface map to match the area of the LFK study site. We converted the depth polygons of the DPB surface map to elevations by assigning XYZ coordinates to them and converting the DPB vertical datums to match those of the 1938 DEM. We then performed a difference analysis between the 1938 DEM surface and the updated DPB elevation layer. We calculated a Holocene reef deposit volume in 1938 of 101.2 Mm$^3$ at the LFK site (see Supplemental Methods Section 1.9 for data processing steps). Comparison of total Holocene reef-deposit

volume in 1938 to the amount of net volume loss that occurred between 1938 and 2004 at the study site of 5.7 mM$^3$ (Table 8) indicates that Holocene reef-deposit volume has decreased by as much as 5% over the past 66 years. Assuming a modern reef age of 6000 years (Lidz, 2000) and a constant erosion rate of 5% per 66 years, total reef volume at this location could completely erode down to the Pleistocene-bedrock-surface in approximately 1250 years. This suggests that the Holocene reef

deposit may be eroding up to 5 times faster than it accreted throughout the Holocene in the LFK.

The seafloor elevation and volume losses observed in our study indicate the extent of erosion that has already occurred in many coral reef ecosystems may be largely under-estimated without comprehensive analysis of total elevation and volume change. Without such analyses, assessments of risk levels and impact from coastal hazards due to coral reef degradation may

also be under-estimated. Numerical models indicate that loss of reef structure and seafloor-elevation increases coastal vulnerability to erosion, storm surge, waves, sea level rise and tsunami hazards that are predicted to intensify and become more frequent with global climate change (Lowe et al., 2005; Quataert et al., 2015). An increase in water depth of 0.5 to 1.0 m from rising sea level by 2100 is projected to cause larger waves and accelerated physical erosion of sediments on fringing reef flats and adjacent coastlines (Storlazzi et al., 2011). Our results show that degradation of coral reef ecosystems in

Atlantic, Caribbean and Pacific regions has accelerated the relative increase in water depth from sea level rise, and has already increased water depths to levels that were not expected until near 2100. We projected increases in water depth at our study sites based on our measured rates of mean elevation loss combined with sea level rise. Our projections indicate that seafloor erosion will increase water depths by 2 to 8 times more than levels predicted from sea level rise alone by the year 2100 at all study sites (Fig. 11, Table S6). This divergence between rising sea level and declining seafloor elevation has

already increased the risk to coastlines in these regions from long-term, persistent oceanographic pressures and periodic events such as storms. The combination of human impacts to reefs and unprecedented rates of global climate change are likely to intensify coastal hazard vulnerabilities caused by seafloor erosion in coral reef ecosystems over the next century.

**5 Conclusions**

Results of our study show that ecosystem- to regional-scale elevation-change analyses are an effective means for quantifying

whole system accretion and erosion, and for characterizing change in the physical structure of the seafloor at spatial resolutions needed for numerical modelling of physical oceanographic processes. Furthermore, results from these analyses are valuable for examining the relative contribution of erosion and accretion processes and identifying missing gaps in process accounting. We suggest that combining comprehensive elevation-change analyses with coordinated ecosystem process studies and appropriately scaled geologic characterization can better establish the links among contemporary

processes causing coral reef ecosystem degradation, long-term changes in the physical structure of the environment, and future implications for physical and societal impacts. There are, however, some key limitations that should be considered in future applications of these methods including:

1) Elevation-change analyses alone do not attribute physical change in seafloor structure to cause, but can be combined with results from field observations and process studies to help identify and account for the causes.

2) Calculation of accretion and erosion from these methods is limited to volumetric values and cannot be used to determine mass of carbonate without more detailed analysis of sediment and materials bulk densities.

3) Not all historical or contemporary bathymetric data sets are appropriate for use in these studies. The coarse horizontal resolution of most historical data limits assessments to large spatial scales. Careful evaluation of the potential error due to data resolution must be performed.

The findings of our study show that coral reef ecosystems in the Florida Keys, U.S. Virgin Islands and Maui, Hawaii are net erosional. Elevation and volume losses observed in our study indicate the amount of erosion that has occurred over the past several decades has likely been underestimated in previous studies. Substantial erosion was observed both in coral-dominated and non-coral dominated habitats. However, erosion in non-coral habitats accounted for most of the volume loss at these study sites suggesting that carbonate production at these sites can no longer support accretion of coral reefs or their adjacent habitats. Select habitats that were more distant from large population centers, or were located in protected areas or near refuges, showed either lower rates of erosion or net accretion illustrating the value of managed protection in these areas. However, those areas showing net accretion could not compensate for ecosystem-scale, mean-loss trends. The magnitude of reef volume lost due to erosion provides evidence for the onset of an Anthropocene reef crisis similar to ancient reef crises caused by climate change and marked in the geologic record by regional and global declines in reef volume. Losses in mean seafloor elevation over the past few decades have already increased local sea level rise to water depths not predicted to occur until near 2100, and to levels that exacerbate impacts of coastal erosion, storm surge, waves, and tsunami hazards. The magnitude of erosion that has already occurred, trajectories for continued coral reef degradation and increasing sea level place these ecosystems and nearby communities at elevated and accelerating risk to coastal hazards.

**Author contributions**

K.K. Yates and D.G. Zawada conceived of and designed the research project, interpreted the results, and wrote the manuscript. N.A. Smiley acquired and processed the historical data sets. G. Tiling-Range created the ArcGIS models to automate computation of the habitat-related statistics.

**Acknowledgements**

We thank N. Buster for assistance with preparation of historical data for geospatial analysis and review of methods sections, and X. Fredericks for assistance with development of LiDAR digital elevation models. N. Plant provided advice and

comments on elevation surface modeling and error propagation statistics. We thank C. Moore for help with manuscript formatting. M. Michalski, J. Campagnoli, M. Cole, S. White, T. Ehret provided assistance with and information for validation of metadata for historical bathymetry data from NOAA. N. Johnston provided assistance with LiDAR data access and metadata from the Joint Airborne Lidar Bathymetry Center of Expertise (JALBTCX). This work was supported and

funded by the U.S. Geological Survey - Coastal and Marine Geology Program. Use of trade, firm or product names is for descriptive purposes only and does not imply endorsement by the U.S. Government.

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

**Table 1** Area, benthic cover and fish survey statistics for study sites.

| Location | Study site area (km$^2$) | Δ % live coral | Δ % macroalgae | Δ *Diadema* sp. density (# m$^{-2}$) | Δ parrotfish biomass (g m$^{-2}$) |
|---|---|---|---|---|---|
| UFK | 241.1 | -23 (1975 – 2011) | +15 (1974 – 2011) | -2 (density near 0) (1970's – 2011) | +5 (late 1990's – 2011) |
| LFK | 19.0 | -25 (1984 – 2010) | +15 (1974 – 2011) | -3 to -17 (1970's – 2011) | +5 (late 1990's – 2011) |
| STT | 116.9 | -10 (1978 – 2011) | +5 (2001 – 2010) | -23 (1973 – 2011) | +13 (2008 – 2010) |
| BI | 51.8 | -14 (1976 – 2011) | 0 (1982 – 2011) | -23 (1973 – 2011) | -2 (2003 – 2010) |
| Maui | 84.5 | -4 (1999 – 2015) | Decrease (1999 – 2015) | N | 0 to increase (2008 – 2015) |

Values are estimated from data reported in Jackson et al. (2014) and Sparks et al. (2015). N = no data. Parentheses indicate time period of data collection.

**Table 2.** Population at study sites.

| Study region | Census Date | Population |
|---|---|---|
| Florida Keys (Monroe County) | 1940 | 14078 |
| Florida Keys | 2000 | 79535 |
| Key Largo, Upper Florida Keys | 2000 | 12971 |
| Key West, Lower Florida Keys | 2000 | 25431 |
| St. Thomas, US Virgin Islands | 1970 | 28960 |
| | 2010 | 51634 |
| Buck Island, US Virgin Islands | 1980 | Uninhabited |
| | 2010 | Uninhabited (50,000 visitors/yr) |
| Maui, Hawaii | 1960 | 35717 |
| | 2000 | 117644 |

Notes. Data acquired from the U.S. Census Bureau (2016).

**Table 3.** Bathymetric and habitat data sources and descriptions.

| Study site | Data type | Source | Source reference | Source title | Year |
|---|---|---|---|---|---|
| UFK | Bathymetry | NOAA | (NOAA Office of Coast Survey) | H05536 | 1934 |
| (68 yrs) | Bathymetry | NOAA | (NOAA Office of Coast Survey) | H05578 | 1934 |
| | Bathymetry | NOAA | (NOAA Office of Coast Survey) | H05726A | 1934 |
| | Bathymetry | NOAA | (NOAA Office of Coast Survey) | H05878A | 1935 |
| | Bathymetry | NOAA | (NOAA Office of Coast Survey) | H05879 | 1935 |
| | LiDAR | USGS | (Brock et al., 2007; Brock et al., 2006a) | EAARL | 2002 |
| | Habitat | FWC | (Florida Fish and Wildlife Conservation Commission, 2015) | UFRT Habitat Map v1.2 | 2014 |
| LFK | Bathymetry | NOAA | (NOAA Office of Coast Survey) | H06323 | 1938 |
| (66 yrs) | LiDAR | USACE | (U.S. Army Corps of Engineers-JALBTCX, 2004) | SHOALS | 2004 |
| | Habitat | FWC | (Florida Fish and Wildlife Conservation Commission, 2015) | UFRT Habitat Map v1.2 | 2014 |
| STT | Bathymetry | NOAA | (NOAA Office of Coast Survey) | H08877 | 1966 |
| (48 yrs) | Bathymetry | NOAA | (NOAA Office of Coast Survey) | H09271 | 1972 |
| | Bathymetry | NOAA | (NOAA Office of Coast Survey) | H09272 | 1972 |
| | Bathymetry | NOAA | (NOAA Office of Coast Survey) | H09353 | 1973 |
| | LiDAR | USGS | (Fredericks et al., 2015b) | EAARL | 2014 |
| | Habitat | NOAA | (Kendall, 2001a) | NOAA | 1999 |
| STC | Bathymetry | NOAA | (NOAA Office of Coast Survey) | H09936 | 1981 |
| (33 yrs) | Bathymetry | NOAA | (NOAA Office of Coast Survey) | H10002 | 1982 |
| | Bathymetry | NOAA | (NOAA Office of Coast Survey) | H10003 | 1982 |
| | LiDAR | USGS | (Fredericks et al., 2015a) | EAARL | 2014 |
| | Habitat | NOAA | (Kendall, 2001b) | NOAA | 1999 |
| Maui | Bathymetry | NOAA | (NOAA Office of Coast Survey) | H08576 | 1961 |
| (38 yrs) | Bathymetry | NOAA | (NOAA Office of Coast Survey) | H08577 | 1961 |
| | Bathymetry | NOAA | (NOAA Office of Coast Survey) | H08578 | 1961 |
| | Bathymetry | NOAA | (NOAA Office of Coast Survey) | H08579 | 1961 |
| | Bathymetry | NOAA | (NOAA Office of Coast Survey) | H08580 | 1961 |
| | Bathymetry | NOAA | (NOAA Office of Coast Survey) | H08581 | 1961 |
| | Bathymetry | NOAA | (NOAA Office of Coast Survey) | H08680 | 1962 |
| | Bathymetry | NOAA | (NOAA Office of Coast Survey) | H08681 | 1963 |
| | Bathymetry | NOAA | (NOAA Office of Coast Survey) | H08682 | 1962 |
| | Bathymetry | NOAA | (NOAA Office of Coast Survey) | H08683 | 1962 |
| | Bathymetry | NOAA | (NOAA Office of Coast Survey) | H08684 | 1962 |
| | Bathymetry | NOAA | (NOAA Office of Coast Survey) | H08685 | 1962 |
| | Bathymetry | NOAA | (NOAA Office of Coast Survey) | H08686 | 1962 |
| | Bathymetry | NOAA | (NOAA Office of Coast Survey) | H08687 | 1962 |
| | Bathymetry | NOAA | (NOAA Office of Coast Survey) | H08717 | 1963 |
| | Bathymetry | NOAA | (NOAA Office of Coast Survey) | H08718 | 1966 |
| | Bathymetry | NOAA | (NOAA Office of Coast Survey) | H08719 | 1965 |
| | Bathymetry | NOAA | (NOAA Office of Coast Survey) | H08720 | 1963 |
| | Bathymetry | NOAA | (NOAA Office of Coast Survey) | H08721 | 1963 |
| | Bathymetry | NOAA | (NOAA Office of Coast Survey) | H08723 | 1963 |
| | Bathymetry | NOAA | (NOAA Office of Coast Survey) | H08792 | 1964 |

| | | | | |
|---|---|---|---|---|
| Bathymetry | NOAA | (NOAA Office of Coast Survey) | H08793 | 1964 |
| Bathymetry | NOAA | (NOAA Office of Coast Survey) | H08825 | 1965 |
| Bathymetry | NOAA | (NOAA Office of Coast Survey) | H08826 | 1965 |
| LiDAR | USACE | (U.S. Army Corps of Engineers-JALBTCX, 1999) | SHOALS | 1999 |
| Habitat | NOAA | (Battista and Christensen, 2007) | NOAA | 2007 |

**Table 4.** Geospatial information and transformations for historical bathymetry and LiDAR data.

| Study region | Source title | Vertical datum source | Geoid | Vertical datum final | Geoid | Horizontal datum source | UTM zone | Horizontal datum final | UTM zone | NOAA sea level trend station | NOAA sea-level rise rate (mm/yr) | Yrs | Relative sea-level rise correction (m) |
|---|---|---|---|---|---|---|---|---|---|---|---|---|---|
| UFK | H05536 | MLW | - | NAVD88 | 03 | Geo NAD83 | - | UTM NAD83 | 17 | 8724580 | 2.33 | 67 | 0.15611 |
| | H05578 | MLW | - | NAVD88 | 03 | Geo NAD83 | - | UTM NAD83 | 17 | 8724580 | 2.33 | 67 | 0.15611 |
| | H05726A | MLW | - | NAVD88 | 03 | Geo NAD83 | - | UTM NAD83 | 17 | 8724580 | 2.33 | 67 | 0.15611 |
| | H05878A | MLW | - | NAVD88 | 03 | Geo NAD83 | - | UTM NAD83 | 17 | 8724580 | 2.33 | 66 | 0.15378 |
| | H05879 | MLW | - | NAVD88 | 03 | Geo NAD83 | - | UTM NAD83 | 17 | 8724580 | 2.33 | 66 | 0.15378 |
| | EAARL | NAVD88 | 03 | NAVD88 | 03 | Geo NAD83 | - | UTM NAD83 | 17 | - | - | - | - |
| LFK | H06323 | MLW | - | NAVD88 | 03 | Geo NAD83 | - | UTM NAD83 | 17 | 8724580 | 2.33 | 66 | 0.15378 |
| | SHOALS | NAVD88 | 03 | NAVD88 | 03 | UTM NAD83 | 17 | UTM NAD83 | 17 | - | - | - | - |
| STT | H08877 | MLLW | - | VIVD09 | - | Geo NAD83 | - | UTM NAD83 | 20 | 9751639 | 1.77 | 48 | 0.08496 |
| | H09271 | MLLW | - | VIVD09 | - | Geo NAD83 | - | UTM NAD83 | 20 | 9751639 | 1.77 | 42 | 0.07434 |
| | H09272 | MLLW | - | VIVD09 | - | Geo NAD83 | - | UTM NAD83 | 20 | 9751639 | 1.77 | 42 | 0.07434 |
| | H09353 | MLLW | - | VIVD09 | - | Geo NAD83 | - | UTM NAD83 | 20 | 9751639 | 1.77 | 41 | 0.07257 |
| | EAARL | NAVD88 | 12B | NAVD88 | 12B | UTM NAD83 | 20 | UTM NAD83 | 20 | - | - | - | - |
| STC | H09936 | MLLW | - | VIVD09 | - | Geo NAD83 | - | UTM NAD83 | 20 | 9751401 | 2.29 | 33 | 0.07557 |
| | H10002 | MLLW | - | VIVD09 | - | Geo NAD83 | - | UTM NAD83 | 20 | 9751401 | 2.29 | 32 | 0.07328 |
| | H10003 | MLLW | - | VIVD09 | - | Geo NAD83 | - | UTM NAD83 | 20 | 9751401 | 2.29 | 32 | 0.07328 |
| | EAARL | NAVD88 | 12B | NAVD88 | 12B | UTM NAD83 | 20 | UTM NAD83 | 20 | - | - | - | - |
| Maui | H08576 | MLLW | - | MLLW | - | Geo NAD83 | - | UTM NAD83 | 4 | 1615680 | 2.02 | 38 | 0.07676 |
| | H08577 | MLLW | - | MLLW | - | Geo NAD83 | - | UTM NAD83 | 4 | 1615680 | 2.02 | 38 | 0.07676 |
| | H08578 | MLLW | - | MLLW | - | Geo NAD83 | - | UTM NAD83 | 4 | 1615680 | 2.02 | 38 | 0.07676 |
| | H08579 | MLLW | - | MLLW | - | Geo NAD83 | - | UTM NAD83 | 4 | 1615680 | 2.02 | 38 | 0.07676 |
| | H08580 | MLLW | - | MLLW | - | Geo NAD83 | - | UTM NAD83 | 4 | 1615680 | 2.02 | 38 | 0.07676 |
| | H08581 | MLLW | - | MLLW | - | Geo NAD83 | - | UTM NAD83 | 4 | 1615680 | 2.02 | 38 | 0.07676 |
| | H08680 | MLLW | - | MLLW | - | Geo NAD83 | - | UTM NAD83 | 4 | 1615680 | 2.02 | 37 | 0.07474 |
| | H08681 | MLLW | - | MLLW | - | Geo NAD83 | - | UTM NAD83 | 4 | 1615680 | 2.02 | 36 | 0.07272 |
| | H08682 | MLLW | - | MLLW | - | Geo NAD83 | - | UTM NAD83 | 4 | 1615680 | 2.02 | 37 | 0.07474 |
| | H08683 | MLLW | - | MLLW | - | Geo NAD83 | - | UTM NAD83 | 4 | 1615680 | 2.02 | 37 | 0.07474 |
| | H08684 | MLLW | - | MLLW | - | Geo NAD83 | - | UTM NAD83 | 4 | 1615680 | 2.02 | 37 | 0.07474 |
| | H08685 | MLLW | - | MLLW | - | Geo NAD83 | - | UTM NAD83 | 4 | 1615680 | 2.02 | 37 | 0.07474 |
| | H08686 | MLLW | - | MLLW | - | Geo NAD83 | - | UTM NAD83 | 4 | 1615680 | 2.02 | 37 | 0.07474 |
| | H08687 | MLLW | - | MLLW | - | Geo NAD83 | - | UTM NAD83 | 4 | 1615680 | 2.02 | 37 | 0.07474 |
| | H08717 | MLLW | - | MLLW | - | Geo NAD83 | - | UTM NAD83 | 4 | 1615680 | 2.02 | 36 | 0.07272 |
| | H08718 | MLLW | - | MLLW | - | Geo NAD83 | - | UTM NAD83 | 4 | 1615680 | 2.02 | 33 | 0.06666 |
| | H08719 | MLLW | - | MLLW | - | Geo NAD83 | - | UTM NAD83 | 4 | 1615680 | 2.02 | 34 | 0.06868 |
| | H08720 | MLLW | - | MLLW | - | Geo NAD83 | - | UTM NAD83 | 4 | 1615680 | 2.02 | 36 | 0.07272 |
| | H08721 | MLLW | - | MLLW | - | Geo NAD83 | - | UTM NAD83 | 4 | 1615680 | 2.02 | 36 | 0.07272 |
| | H08723 | MLLW | - | MLLW | - | Geo NAD83 | - | UTM NAD83 | 4 | 1615680 | 2.02 | 36 | 0.07272 |
| | H08792 | MLLW | - | MLLW | - | Geo NAD83 | - | UTM NAD83 | 4 | 1615680 | 2.02 | 35 | 0.0707 |
| | H08793 | MLLW | - | MLLW | - | Geo NAD83 | - | UTM NAD83 | 4 | 1615680 | 2.02 | 35 | 0.0707 |
| | H08825 | MLLW | - | MLLW | - | Geo NAD83 | - | UTM NAD83 | 4 | 1615680 | 2.02 | 34 | 0.06868 |
| | H08826 | MLLW | - | MLLW | - | Geo NAD83 | - | UTM NAD83 | 4 | 1615680 | 2.02 | 34 | 0.06868 |
| | SHOALS | MLLW | - | MLLW | - | UTM NAD83 | 4 | UTM NAD83 | 4 | - | - | - | - |

Notes: Dashes indicate not applicable. Years = number of years by which sea-level rise rates were multiplied to calculate relative sea-level rise correction.

**Table 5.** Vertical error analysis results.

| Study site | Historical Data | | | Modern Data | VDatum | Study Site Totals | |
|---|---|---|---|---|---|---|---|
| | # paired points | $RMSE_{Sounding}$ (m) | T-test results (±, P) | $RMSE_{Lidar}$ (m) | $RMSE_{VDatum}$ (m) | $RMSE_{Total}$ (m) | *1.65 x study site $RMSE_{Total}$ (m) |
| UFK | 38 | 0.23 | +, 0.96 | 0.15 | 0.081 | 0.29 | 0.47 |
| STT | 184 | 0.26 | +, 0.81 | 0.135 | 0.114 | 0.31 | 0.52 |
| STC | 29 | 0.32 | +, 0.93 | 0.135 | 0.114 | 0.37 | 0.60 |
| Maui | 46 | 0.13 | +, 0.99 | 0.15 | - | 0.20 | 0.33 |
| Average | - | - | - | - | - | 0.29 | 0.48 |

Notes: LFK was excluded from the error analysis because no adjacent points were separated by 0.5 m or less. For t-test results, + = accept $H_0$. *1.65 x study site $RMSE_{Total}$ = 90% of variance.

**Table 6**. Results of pavement elevation-data analysis.

| Study site | Total # habitat data points | # Points within +/- 0.5 m elevation change | Average difference (m) | Stdev. |
|---|---|---|---|---|
| UFK - pavement | 1901 | 958 | -0.06 | 0.27 |
| LFK - pavement | 198 | 34 | -0.03 | 0.1 |
| STT – uncolonized pavement | 33 | 28 | -0.04 | 0.2 |
| BI – colonized pavement | 3286 | 2543 | 0.04 | 0.26 |
| Maui – coral pavement | 4412 | 1013 | 0.05 | 0.14 |

Average elevation differences between modern and historical elevation data (modern – historical) and standard deviation (Stdev.) are reported for those data points elevation changes within +/- 0.5 m (or 1.65 x $RMSE_{Total}$).

**Table 7.** Horizontal shift analysis results.

| Study site & relative location | Net volume change (Mm$^3$) | % difference from un-shifted value | Area- normalized volume change (Mm$^3$/km$^2$) | % difference from un-shifted value |
|---|---|---|---|---|
| **UFK** | | | | |
| Un-shifted | -37.9 | - | -0.16 | - |
| x + 10 m | -41.7 | 10 | -0.17 | 6 |
| x – 10 m | -34.9 | 8 | -0.15 | 6 |
| y + 10 m | -36.6 | 3 | -0.15 | 6 |
| y – 10 m | -39.5 | 4 | -0.16 | 0 |
| **LFK** | | | | |
| Un-shifted | -5.7 | - | -0.30 | - |
| x + 10 m | -5.8 | 2 | -0.31 | 3 |
| x – 10 m | -5.4 | 5 | -0.29 | 3 |
| y + 10 m | -4.5 | 21 | -0.24 | 20 |
| y – 10 m | -6.7 | 18 | -0.35 | 17 |

Un-shifted results are derived from the original point to DEM data sets, correctly aligned to horizontal datums, for each study site. Experimental data sets were created by shifting UTM northings and eastings of historical elevation XYZ data sets by plus or minus 10 m and recalculating volume change relative to unshifted LiDAR elevation data.

**Table 8.** Seafloor elevation and volume change for Atlantic, Pacific and Caribbean study sites.

| Location, time period | Study area (km²) | Mean elevation change (m) | Standard deviation (m) | Gross erosion (Mm³) | | Gross accretion (Mm³) | | Net change (Mm³) | | Area-normalized volume change (Mm³/km²) | |
|---|---|---|---|---|---|---|---|---|---|---|---|
| | | | | Min. | Max. | Min. | Max. | Min. | Max. | Min. | Max. |
| UFK, 1934/35-2002 | 241.1 | -0.1 | 0.8 | -22.2 | -69.9 | 7.5 | 32.1 | -14.6 | -37.9 | -0.06 | -0.2 |
| LFK, 1938-2004 | 19.0 | -0.3 | 0.8 | -1.8 | -8.2 | 1.6 | 2.6 | -0.2 | -5.7 | -0.01 | -0.3 |
| STT, 1966/73-2014 | 116.9 | -0.3 | 0.9 | -7.4 | -29.1 | 1.4 | 7.5 | -6.0 | -21.6 | -0.05 | -0.2 |
| BI, 1981/82-2014 | 51.8 | -0.09 | 0.7 | -2.3 | -9.6 | 0.7 | 6.1 | -1.6 | -3.4 | -0.03 | -0.07 |
| Maui, 1961/65-1999 | 84.5 | -0.8 | 1.5 | -56.6 | -88.5 | 3.2 | 7.2 | -52.8 | -80.5 | -0.6 | -0.9 |
| UFK coral habitats (5) | 36.1 | -0.2 | 1.1 | -5.9 | -14.3 | 2.3 | 6.3 | -3.5 | -7.9 | -0.1 | -0.2 |
| UFK adjacent habitats | 204.9 | -0.1 | -0.7 | -15.6 | -54.9 | 4.8 | 25.0 | -10.8 | -30 | -0.05 | -0.15 |
| LFK coral habitats (4) | 1.6 | 0.2 | 1.4 | -0.1 | -0.4 | 0.9 | 1.3 | 0.8 | 0.9 | 0.50 | 0.55 |
| LFK adjacent habitats | 17.3 | -0.4 | 0.6 | -1.6 | -7.8 | 0.7 | 1.3 | -1.0 | -6.5 | -0.06 | -0.40 |
| STT coral habitats (9) | 14.3 | -0.5 | 1.3 | -4.0 | -7.9 | 0.6 | 1.7 | -3.4 | -6.2 | -0.2 | -0.4 |
| STT adjacent habitats | 102.7 | -0.2 | 0.6 | -3.3 | -21.2 | 0.7 | 5.8 | -2.6 | -15.4 | -0.03 | -0.2 |
| BI coral habitats (7) | 47.1 | -0.09 | 0.72 | -2.0 | -8.7 | 0.6 | 5.6 | -1.4 | -3.1 | -0.03 | -0.07 |
| BI adjacent habitats | 4.7 | -0.09 | 0.68 | -0.3 | -0.9 | 0.1 | 0.6 | -0.2 | -0.3 | -0.05 | -0.07 |
| Maui coral habitats (7) | 48.4 | -0.9 | 1.5 | -32.2 | -50.0 | 1.9 | 4.2 | -30.4 | -45.8 | -0.6 | -0.9 |
| Maui adjacent habitats | 36.2 | -0.8 | 1.5 | -23.5 | -37.3 | 1.1 | 2.6 | -22.4 | -34.7 | -0.6 | -1.0 |

Minimum (Min.) and maximum (Max.) volumes are based on vertical error analysis (Table 5). Parenthetical numbers indicate number of combined habitat classes for each location (Tables 9 and 10). Mm³ = millions of cubic meters.

**Table 9.** Elevation-change by habitat type.

| Location | Mean elev-ation (m) | Total points (#) | Mean elevation change (m) | Stdev (m) | Max loss (m) | Max gain (m) | Elevation loss points (#) | Mean loss (m) | Stdev (m) | Elevation gain points (#) | Mean gain (m) | Stdev (m) |
|---|---|---|---|---|---|---|---|---|---|---|---|---|
| Upper Florida Keys | | | | | | | | | | | | |
| Total study site | -6.7 | 26341 | -0.1 | 0.8 | -8.3 | 6.5 | 14996 | -0.6 | 0.7 | 11345 | 0.44 | 0.51 |
| *Scattered coral/rock in uncons. sed. | -6.9 | 17 | -0.8 | 0.9 | -2.0 | 1.2 | 13 | -1.1 | 0.6 | 4 | 0.5 | 0.4 |
| Pavement | -5.9 | 1901 | -0.2 | 1.0 | -6.2 | 4.9 | 1218 | -0.7 | 0.7 | 683 | 0.7 | 0.7 |
| *Aggregate reef | -6.6 | 1435 | -0.2 | 1.1 | -5.1 | 4.7 | 857 | -0.8 | 0.8 | 578 | 0.7 | 0.7 |
| *Reef rubble | -3.6 | 291 | -0.2 | 0.8 | -2.6 | 3.2 | 196 | -0.6 | 0.5 | 95 | 0.6 | 0.7 |
| Unconsolidated sediments | -8.7 | 5498 | -0.2 | 0.6 | -6.8 | 3.4 | 3526 | -0.5 | 0.5 | 1972 | 0.3 | 0.3 |
| *Individual or aggregate patch reef | -5.8 | 2078 | -0.2 | 1.1 | -6.6 | 5.2 | 1170 | -0.8 | 0.9 | 908 | 0.6 | 0.7 |
| Seagrass continuous | -6.7 | 6826 | -0.2 | 0.8 | -8.0 | 3.2 | 3604 | -0.6 | 0.8 | 3322 | 0.4 | 0.3 |
| Seagrass discontinuous | -5.2 | 6944 | -0.1 | 0.5 | -8.3 | 2.6 | 3721 | -0.4 | 0.4 | 3223 | 0.3 | 0.3 |
| *Spur and groove | -8.7 | 613 | -0.02 | 1.3 | -4.6 | 6.5 | 343 | -0.8 | 0.8 | 270 | 1.0 | 1.0 |
| Pavement w/sand channels | -12.1 | 623 | 0.2 | 1.2 | -3.2 | 5.7 | 313 | -0.8 | 0.7 | 310 | 1.1 | 0.9 |
| Not classified | -6.2 | 113 | 0.2 | 0.5 | -2.0 | 2.0 | 34 | -0.4 | 0.4 | 79 | 0.4 | 0.3 |
| Lower Florida Keys, Looe Key | | | | | | | | | | | | |
| Total study site | -8.7 | 1688 | -0.3 | 0.8 | -4.0 | 7.3 | 1361 | -0.6 | 0.4 | 327 | 0.7 | 1.1 |
| *Individual or aggregate patch reef | -10.1 | 14 | -0.8 | 0.8 | -3.5 | -0.2 | 14 | -0.8 | 0.8 | - | - | - |
| *Reef rubble | -4.7 | 68 | -0.7 | 0.8 | -2.9 | 0.9 | 55 | -0.9 | 0.6 | 13 | 0.3 | 0.3 |
| Seagrass continuous | -8.6 | 285 | -0.5 | 0.4 | -3.1 | 0.4 | 264 | -0.5 | 0.3 | 21 | 0.2 | 0.1 |
| Seagrass discontinuous | -7.4 | 488 | -0.5 | 0.5 | -1.9 | 2.0 | 427 | -0.6 | 0.3 | 61 | 0.3 | 0.3 |
| Pavement | -9.4 | 198 | -0.4 | 0.7 | -2.2 | 1.9 | 151 | -0.6 | 0.4 | 47 | 0.5 | 0.5 |
| Unconsolidated sediments | -9.8 | 533 | -0.4 | 0.7 | -4.0 | 6.3 | 428 | -0.6 | 0.4 | 105 | 0.7 | 1.0 |
| *Spur and groove | -8.3 | 68 | 0.5 | 1.1 | -1.4 | 4.2 | 21 | -0.5 | 0.4 | 47 | 1.0 | 1.0 |
| *Aggregate reef | -15.1 | 29 | 2.0 | 1.9 | -1.4 | 7.3 | 1 | -1.4 | 0.0 | 28 | 2.2 | 1.9 |
| Not classified | -19.8 | 5 | 3.0 | 1.3 | 0.7 | 4.1 | - | - | - | 5 | 2.9 | 1.3 |
| St. Thomas, US Virgin Islands | | | | | | | | | | | | |
| Total study site | -18.6 | 49269 | -0.3 | 0.9 | -11.2 | 25.6 | 35332 | -0.5 | 0.7 | 13937 | 0.4 | 0.9 |
| *Spur and groove | -7.3 | 75 | -1.1 | 1.6 | -5.9 | 3.1 | 60 | -1.6 | 1.2 | 15 | 1.1 | 0.9 |
| *Patch reef (aggregate) | -19.9 | 163 | -1.0 | 1.6 | -5.9 | 4.9 | 138 | -1.3 | 1.4 | 25 | 0.9 | 1.2 |
| *Linear reef | -15.7 | 2244 | -0.7 | 1.5 | -11.2 | 7.0 | 1632 | -1.2 | 1.4 | 612 | 0.7 | 0.8 |
| *Colonized bedrock | -5.7 | 1968 | -0.6 | 1.5 | -8.3 | 6.1 | 1358 | -1.2 | 1.2 | 610 | 0.9 | 1.0 |
| *Colonized pave.w/sand channels | -17.0 | 591 | -0.5 | 1.0 | -5.7 | 4.3 | 438 | -0.9 | 0.8 | 153 | 0.5 | 0.6 |
| Uncolonized pave. w/sand channels | -11.5 | 41 | -0.5 | 0.9 | -3.0 | 0.9 | 27 | -0.9 | 0.9 | 14 | 0.3 | 0.3 |
| Mud | -8.3 | 1857 | -0.5 | 1.1 | -5.6 | 3.4 | 1262 | -1.0 | 0.9 | 595 | 0.5 | 0.5 |
| *Scattered coral/rock in uncons. sed. | -11.4 | 324 | -0.5 | 1.3 | -8.4 | 4.3 | 228 | -0.9 | 1.2 | 96 | 0.5 | 0.7 |
| *Colonized pavement | -11.5 | 3195 | -0.4 | 1.0 | -7.9 | 8.1 | 2249 | -0.8 | 0.9 | 946 | 0.5 | 0.6 |
| *Patch reef (individual) | -19.0 | 280 | -0.3 | 1.1 | -5.3 | 3.2 | 193 | -0.8 | 0.8 | 87 | 0.7 | 0.8 |
| Seagrass | -12.6 | 11113 | -0.2 | 0.6 | -7.5 | 13.2 | 7914 | -0.4 | 0.6 | 3199 | 0.3 | 0.5 |
| Macroalgae | -23.4 | 13973 | -0.2 | 0.5 | -11.1 | 4.0 | 10360 | -0.3 | 0.5 | 3613 | 0.2 | 0.2 |
| Sand | -8.6 | 961 | -0.2 | 0.8 | -4.2 | 3.9 | 571 | -0.6 | 0.6 | 390 | 0.4 | 0.5 |
| Unknown | -25.6 | 11992 | -0.2 | 0.6 | -8.6 | 11.5 | 8533 | -0.3 | 0.5 | 3459 | 0.2 | 0.5 |
| *Reef rubble | -0.6 | 1 | -0.1 | 0 | -0.1 | - | 1 | -0.1 | 0 | - | - | - |
| Uncolonized pavement | -10.6 | 33 | -0.1 | 0.4 | -1.2 | 0.7 | 20 | -0.3 | 0.4 | 13 | 0.2 | 0.2 |
| Uncolonized bedrock | -1.9 | 6 | 1.1 | 1.0 | - | 2.3 | - | - | - | 6 | 1.1 | 1.0 |
| Buck Island, US Virgin Islands | | | | | | | | | | | | |
| Total study site | -17.9 | 13894 | -0.1 | 0.7 | -8.6 | 6.8 | 7200 | -0.5 | 0.7 | 6694 | 0.4 | 0.4 |
| *Patch reef (aggregated) | -11.4 | 1086 | -0.5 | 1.5 | -8.6 | 6.8 | 629 | -1.3 | 1.5 | 457 | 0.5 | 0.9 |
| *Patch reef (individual) | -15.5 | 17 | -0.3 | 0.5 | -1.5 | 0.7 | 13 | -0.5 | 0.4 | 4 | 0.3 | 0.3 |
| Unknown | -29.7 | 551 | -0.3 | 0.9 | -4.9 | 3.5 | 326 | -0.8 | 0.8 | 225 | 0.5 | 0.5 |
| *Colonized pave./w sand channels | -18.7 | 6914 | -0.1 | 0.6 | -5.9 | 3.3 | 3809 | -0.4 | 0.5 | 3105 | 0.3 | 0.3 |
| *Scattered coral/rock in uncons. sed. | -21.1 | 1152 | -0.01 | 0.6 | -4.7 | 2.8 | 490 | -0.4 | 0.6 | 662 | 0.3 | 0.3 |
| *Reef rubble | -12.2 | 10 | 0 | 0.3 | -0.5 | 0.5 | 5 | -0.3 | 0.2 | 5 | 0.3 | 0.2 |
| *Colonized pavement | -16.4 | 3286 | 0.01 | 0.5 | -3.0 | 4.5 | 1529 | -0.4 | 0.4 | 1757 | 0.3 | 0.3 |
| Seagrass | -14.3 | 610 | 0.02 | 0.4 | -2.2 | 0.9 | 270 | -0.2 | 0.3 | 340 | 0.2 | 0.2 |
| Sand | -15.3 | 88 | 0.1 | 0.6 | -3.8 | 0.9 | 27 | -0.5 | 0.8 | 61 | 0.3 | 0.2 |
| Macroalgae | -10.0 | 26 | 0.2 | 0.5 | -1.6 | 0.7 | 4 | -0.6 | 0.7 | 22 | 0.3 | 0.2 |

| | | | | | | | | | | | | |
|---|---|---|---|---|---|---|---|---|---|---|---|---|
| *Linear reef | 0.1 | 83 | 0.6 | 2.1 | -3.0 | 5.7 | 43 | -0.9 | 0.8 | 40 | 2.2 | 1.9 |

Maui, Hawaii

| | | | | | | | | | | | | |
|---|---|---|---|---|---|---|---|---|---|---|---|---|
| Total study site | -17.0 | 27518 | -0.8 | 1.5 | -28.7 | 11.8 | 22111 | -1.2 | 1.4 | 5407 | 0.7 | 0.9 |
| *Individual patch reef | -14.1 | 166 | -1.3 | 2.0 | -7.4 | 4.9 | 140 | -1.9 | 1.5 | 26 | 1.6 | 1.3 |
| *Coral reef and hardbottom | -12.4 | 5610 | -1.3 | 2.0 | -16.7 | 8.3 | 4592 | -1.8 | 1.8 | 1018 | 1.0 | 1.2 |
| Unknown | -12.1 | 4548 | -1.2 | 2.2 | -28.7 | 9.2 | 3520 | -1.8 | 2.1 | 1028 | 0.8 | 1.0 |
| *Spur and groove | -8.2 | 369 | -0.8 | 1.1 | -9.2 | 2.7 | 300 | -1.1 | 1.0 | 69 | 0.5 | 0.5 |
| *Aggregate patch reef | -14.5 | 170 | -0.7 | 0.8 | -3.9 | 1.7 | 148 | -0.9 | 0.7 | 22 | 0.6 | 0.5 |
| *Aggregate reef | -10.1 | 3666 | -0.7 | 1.1 | -9.4 | 8.6 | 2892 | -1.0 | 0.9 | 774 | 0.6 | 0.8 |
| Sand | -13.6 | 8381 | -0.6 | 0.8 | -11.5 | 9.8 | 7272 | -0.8 | 0.7 | 1109 | 0.5 | 0.7 |
| *Scattered coral/rock | -4.0 | 17 | -0.6 | 0.7 | -2.1 | 0.9 | 15 | -0.7 | 0.5 | 2 | 0.7 | 0.3 |
| *Coral pavement | -9.3 | 4412 | -0.4 | 0.9 | -10.4 | 6.4 | 3120 | -0.8 | 0.8 | 1292 | 0.5 | 0.6 |
| Pavement w/sand channels | -13.8 | 158 | -0.4 | 0.9 | -4.3 | 1.8 | 105 | -0.8 | 0.8 | 53 | 0.5 | 0.4 |
| Rubble | -2.6 | 10 | 0.03 | 0.5 | -0.5 | 0.8 | 5 | -0.4 | 0.1 | 5 | 0.5 | 0.3 |
| Mud | -2.2 | 4 | 0.2 | 1.3 | -0.6 | 2.1 | 3 | -0.4 | 0.2 | 1 | 2.1 | 0.0 |

Notes: * = coral-dominated substrate. Uncons. sed. = unconsolidated sediment. Pave. = pavement. /w = with. Mean elevations are based on LiDAR bathymetry. Dashes indicate not applicable. Data points were excluded from habitat analysis where no habitat delineations were available. Total study site values include all data points.

**Table 10.** Volume change by habitat type

| Location | Habitat area (km$^2$) | Gross erosion (Mm$^3$) | | Gross accretion (Mm$^3$) | | Net volume change (Mm$^3$/study area) | | Area-normalized volume change (Mm$^3$/km$^2$) | |
|---|---|---|---|---|---|---|---|---|---|
| | | Lower | Upper | Lower | Upper | Lower | Upper | Lower | Upper |
| Upper Florida Keys | | | | | | | | | |
| Total study site | 241.1 | -21.5 | -69.2 | 7.1 | 31.3 | -14.3 | -37.9 | -0.1 | -0.2 |
| *Scattered coral/rock in uncons. sed. | 0.2 | -0.1 | -0.1 | 0.00 | 0.01 | -0.1 | -0.1 | -0.3 | -0.7 |
| Pavement | 17.0 | -2.4 | -6.6 | 0.8 | 2.4 | -1.6 | -4.3 | -0.1 | -0.3 |
| *Aggregate reef | 11.8 | -1.9 | -4.5 | 0.8 | 2.3 | -1.1 | -2.2 | -0.1 | -0.2 |
| *Reef rubble | 1.5 | -0.2 | -0.6 | 0.1 | 0.2 | -0.1 | -0.4 | -0.1 | -0.2 |
| Unconsolidated sediments | 65.3 | -4.1 | -17.5 | 0.9 | 5.8 | -3.2 | -11.7 | -0.05 | -0.2 |
| *Individual or aggregate patch reef | 17.2 | -2.9 | -7.1 | 0.6 | 2.1 | -2.4 | -4.9 | -0.1 | -0.3 |
| Seagrass continuous | 63.5 | -6.4 | -18.7 | 1.1 | 8.5 | -5.2 | -10.2 | -0.1 | -0.2 |
| Seagrass discontinuous | 51.9 | -2.0 | -9.8 | 0.7 | 6.0 | -1.3 | -3.9 | -0.02 | -0.1 |
| *Spur and groove | 5.5 | -0.8 | -2.0 | 0.8 | 1.6 | 0.1 | -0.3 | 0.01 | -0.1 |
| Pavement with sand channels | 6.3 | -0.8 | -2.2 | 1.2 | 2.2 | 0.4 | -0.04 | 0.1 | -0.01 |
| Not classified | 0.7 | -0.01 | -0.04 | 0.03 | 0.2 | 0.02 | 0.1 | 0.03 | 0.2 |
| Lower Florida Keys, Looe Key | | | | | | | | | |
| Total study site | 19.0 | -1.7 | -8.2 | 1.6 | 2.6 | -0.1 | -5.6 | -0.01 | -0.3 |
| *Individual or aggregate patch reef | 0.1 | -0.02 | -0.1 | 0.00 | 0.00 | -0.02 | -0.1 | -0.2 | -0.6 |
| *Reef rubble | 0.4 | -0.1 | -0.2 | 0.00 | 0.01 | -0.1 | -0.2 | -0.2 | -0.5 |
| Seagrass continuous | 3.6 | -0.2 | -1.7 | 0.00 | 0.01 | -0.2 | -1.7 | -0.1 | -0.5 |
| Seagrass discontinuous | 4.6 | -0.5 | -2.3 | 0.0 | 0.1 | -0.5 | -2.2 | -0.1 | -0.5 |
| Pavement | 2.2 | -0.2 | -0.9 | 0.02 | 0.1 | -0.2 | -0.8 | -0.1 | -0.3 |
| Unconsolidated sediments | 6.8 | -0.7 | -2.9 | 0.5 | 0.9 | -0.2 | -2.0 | -0.02 | -0.3 |
| *Spur and groove | 0.7 | -0.01 | -0.1 | 0.2 | 0.4 | 0.2 | 0.3 | 0.3 | 0.4 |
| *Aggregate reef | 0.4 | 0.00 | -0.01 | 0.7 | 0.9 | 0.7 | 0.9 | 1.6 | 2.1 |
| Not classified | 0.1 | 0.00 | 0.00 | 0.1 | 0.2 | 0.1 | 0.2 | 1.7 | 2.2 |
| St. Thomas, US Virgin Islands | | | | | | | | | |
| Total study site | 116.9 | -7.4 | -29.1 | 1.4 | 7.5 | -6.0 | -21.6 | -0.05 | -0.2 |
| *Spur and groove | 0.1 | -0.1 | -0.2 | 0.00 | 0.01 | -0.1 | -0.2 | -0.8 | -1.2 |
| *Patch reef (aggregate) | 0.4 | -0.2 | -0.3 | 0.00 | 0.02 | -0.2 | -0.3 | -0.4 | -0.7 |
| *Linear reef | 4.2 | -1.3 | -2.4 | 0.2 | 0.5 | -1.1 | -1.9 | -0.3 | -0.5 |
| *Colonized bedrock | 2.4 | -1.0 | -1.8 | 0.2 | 0.4 | -0.8 | -1.3 | -0.3 | -0.6 |
| *Colonized pavement with sand channels | 1.5 | -0.3 | -0.8 | 0.02 | 0.1 | -0.3 | -0.7 | -0.2 | -0.5 |
| Uncolonized pavement with sand channels | 0.1 | 0.0 | -0.1 | 0.00 | 0.01 | -0.02 | -0.05 | -0.2 | -0.5 |
| Mud | 1.7 | -0.6 | -1.0 | 0.1 | 0.3 | -0.5 | -0.8 | -0.3 | -0.5 |
| *Scattered coral/rock in uncons. sed. | 0.5 | -0.1 | -0.2 | 0.02 | 0.1 | -0.1 | -0.1 | -0.2 | -0.3 |
| *Colonized pavement | 4.7 | -0.9 | -2.0 | 0.2 | 0.5 | -0.8 | -1.5 | -0.2 | -0.3 |
| *Patch reef (individual) | 0.4 | -0.1 | -0.2 | 0.02 | 0.1 | -0.1 | -0.2 | -0.2 | -0.4 |
| Seagrass | 16.0 | -0.9 | -3.6 | 0.1 | 0.9 | -0.7 | -2.7 | -0.05 | -0.2 |
| Macroalgae | 44.5 | -1.1 | -9.8 | 0.1 | 1.9 | -1.0 | -8.0 | -0.02 | -0.2 |
| Sand | 1.6 | -0.1 | -0.4 | 0.03 | 0.1 | -0.1 | -0.3 | -0.1 | -0.2 |
| Unknown | 38.8 | -0.6 | -6.2 | 0.3 | 2.6 | -0.2 | -3.6 | -0.01 | -0.1 |
| *Reef rubble | 0.02 | 0.00 | -0.01 | 0.00 | 0.00 | 0.00 | -0.01 | -0.1 | -0.5 |
| Uncolonized pavement | 0.1 | 0.00 | -0.01 | 0.00 | 0.01 | 0.00 | 0.00 | -0.02 | 0.0 |
| Uncolonized bedrock | 0.02 | 0.00 | 0.00 | 0.00 | 0.01 | 0.00 | 0.01 | 0.2 | 0.3 |
| Buck Island, US Virgin Islands | | | | | | | | | |
| Total study site | 51.8 | -2.3 | -9.6 | 0.7 | 6.1 | -1.6 | -3.4 | -0.03 | -0.07 |
| *Patch reef (aggregated) | 3.2 | -0.9 | -1.6 | 0.1 | 0.4 | -0.8 | -1.2 | -0.2 | -0.4 |
| *Patch reef (individual) | 0.1 | 0.00 | -0.01 | 0.00 | 0.00 | 0.00 | -0.01 | -0.03 | -0.2 |
| Unknown | 1.8 | -0.2 | -0.6 | 0.05 | 0.2 | -0.2 | -0.4 | -0.1 | -0.2 |
| *Colonized pavement with sand channels | 27.3 | -0.8 | -5.0 | 0.3 | 2.9 | -0.5 | -2.1 | -0.02 | -0.1 |
| *Scattered coral/rock in uncons. sed. | 4.5 | -0.1 | -0.5 | 0.00 | 0.6 | -0.1 | 0.0 | -0.01 | 0.01 |
| *Reef rubble | 0.03 | 0.00 | -0.01 | 0.00 | 0.00 | 0.00 | 0.00 | -0.03 | -0.1 |
| *Colonized pavement | 11.9 | -0.2 | -1.4 | 0.1 | 1.6 | -0.02 | 0.2 | 0.00 | 0.01 |
| Seagrass | 2.4 | -0.03 | -0.2 | 0.00 | 0.2 | -0.02 | 0.1 | -0.01 | 0.0 |

| | | | | | | | | | |
|---|---|---|---|---|---|---|---|---|---|
| Sand | 0.4 | -0.01 | -0.04 | 0.01 | 0.07 | 0.00 | 0.02 | -0.01 | 0.1 |
| Macroalgae | 0.1 | 0.00 | -0.01 | 0.00 | 0.02 | 0.00 | 0.00 | -0.02 | 0.0 |
| *Linear reef | 0.1 | -0.01 | -0.03 | 0.04 | 0.05 | 0.02 | 0.02 | 0.4 | 0.3 |
| Maui, Hawaii | | | | | | | | | |
| Total study site | 84.5 | -55.8 | -87.2 | 3.0 | 6.7 | -52.8 | -80.5 | -0.6 | -1.0 |
| *Individual patch reef | 0.1 | -0.1 | -0.1 | 0.01 | 0.01 | -0.1 | -0.1 | -0.9 | -1.2 |
| *Coral reef and hardbottom | 19.5 | -23.3 | -31.4 | 1.1 | 1.9 | -22.2 | -29.4 | -1.1 | -1.5 |
| Unknown | 13.2 | -15.0 | -19.9 | 0.7 | 1.5 | -14.2 | -18.3 | -1.1 | -1.4 |
| *Spur and groove | 2.1 | -0.9 | -1.7 | 0.03 | 0.1 | -0.8 | -1.6 | -0.4 | -0.8 |
| *Aggregate patch reef | 0.3 | -0.1 | -0.2 | 0.00 | 0.02 | -0.1 | -0.1 | -0.2 | -0.5 |
| *Aggregate reef | 12.8 | -4.7 | -9.4 | 0.3 | 0.8 | -4.4 | -8.7 | -0.3 | -0.7 |
| Sand | 22.5 | -8.5 | -17.2 | 0.3 | 1.0 | -8.2 | -16.2 | -0.4 | -0.7 |
| *Scattered coral/rock | 0.04 | -0.01 | -0.03 | 0.00 | 0.00 | -0.01 | -0.03 | -0.3 | -0.7 |
| *Coral pavement | 13.5 | -3.2 | -7.2 | 0.4 | 1.3 | -2.8 | -5.8 | -0.2 | -0.4 |
| Pavement with sand channels | 0.4 | -0.1 | -0.2 | 0.01 | 0.1 | -0.1 | -0.2 | -0.2 | -0.3 |
| Rubble | 0.03 | 0.00 | 0.00 | 0.00 | 0.01 | 0.00 | 0.00 | 0.01 | 0.1 |
| Mud | 0.005 | 0.00 | 0.00 | 0.00 | 0.00 | 0.00 | 0.00 | 0.02 | -0.04 |

* = coral-dominated substrate.

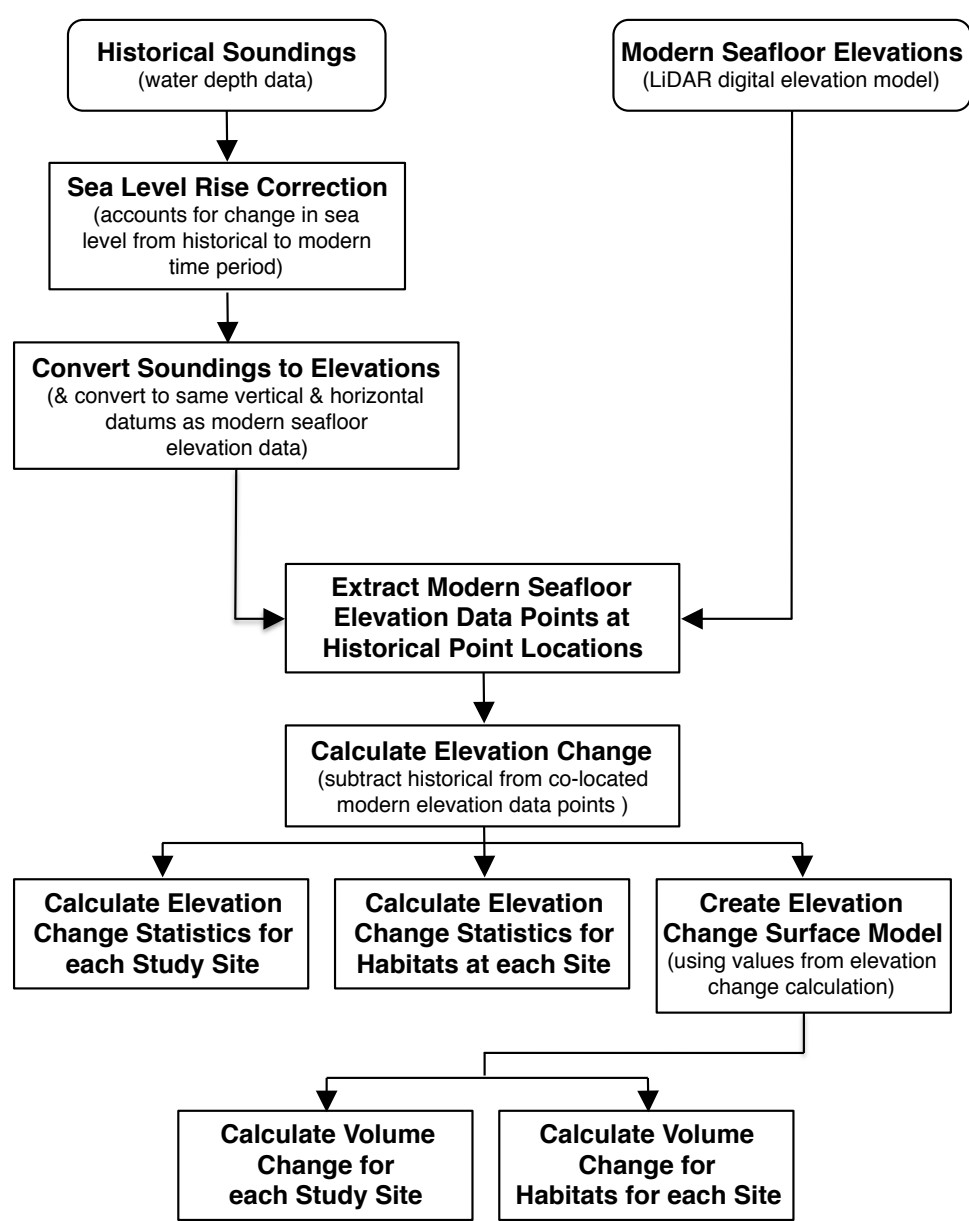

**Figure 1.** Methods flow diagram for elevation and volume change analysis. Negative values = losses. Positive values =gains. Net volume change = volume loss + volume gain. Using surface models, volume loss = 3D surface volume below 0

5    m reference; volume gain = 3D surface volume above 0 m reference.

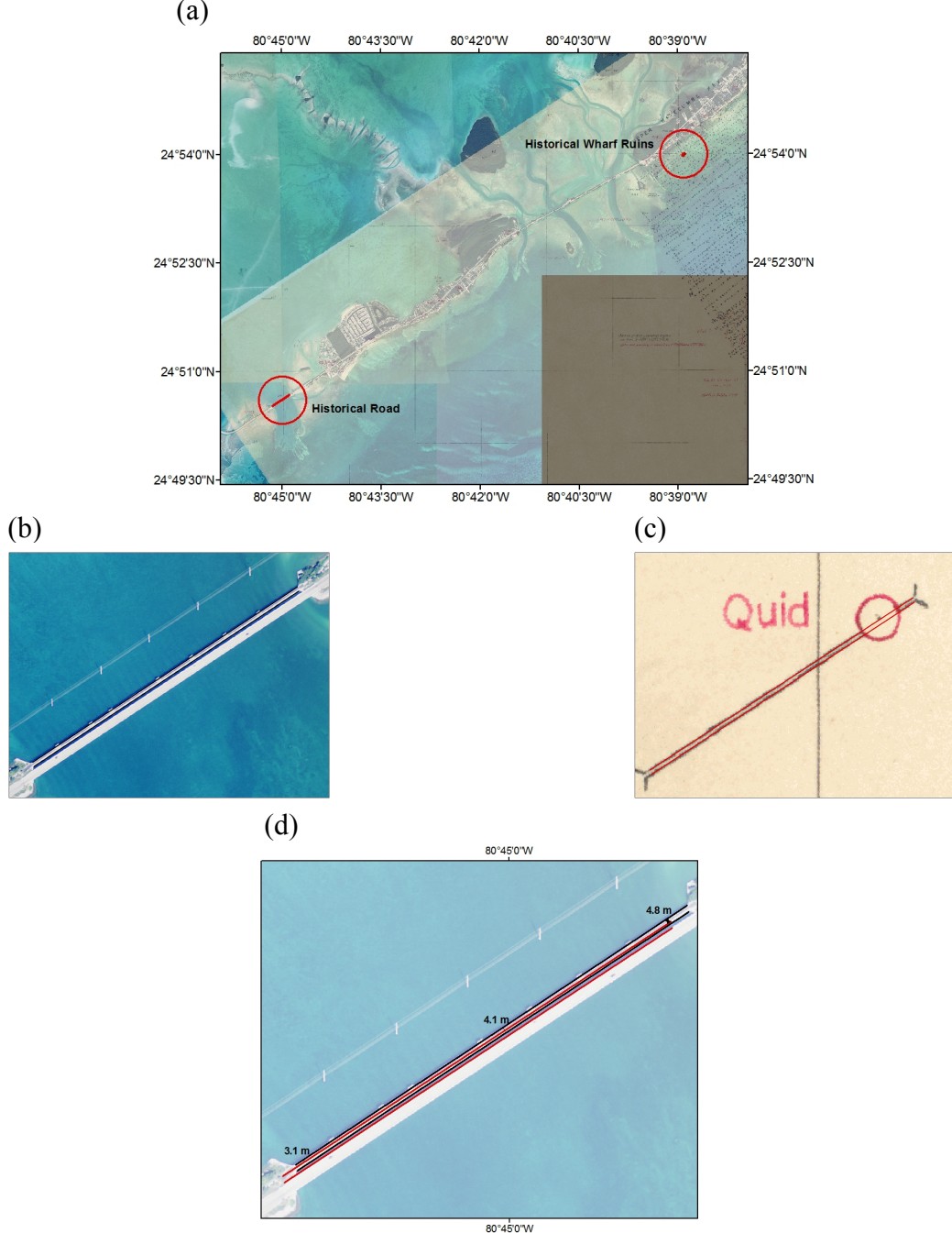

**Figure 2.** Example of visual inspection of historical Hsheets relative to modern aerial imagery as a preliminary 'check' for horizontal alignment. Historical bridge (a) overlay, merging the (b) 2016 aerial imagery with (c) 1934 H-sheet. ArcMap measurement lines (d).

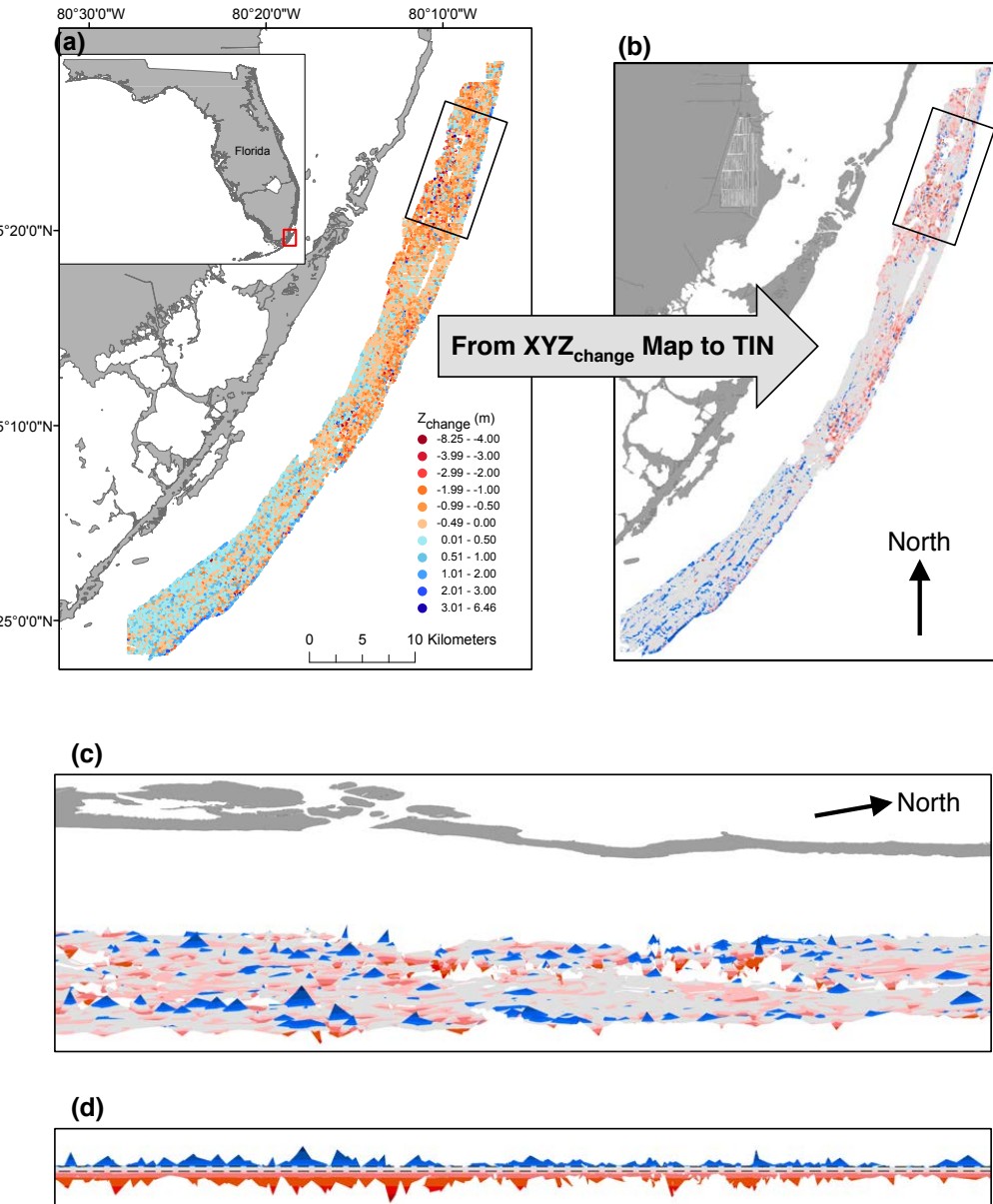

**Figure 3.** Examples of (a) the $XYZ_{change}$ data plot, (b) the three-dimensional triangulated irregular network (TIN) model developed from the $XYZ_{change}$ data, (c) a sub-section of the TIN model, rectangle in (b), rotated clockwise approximately 90° and tilted 15° from horizontal, and (d) a horizontal cross-section view of the TIN model sub-section for the Upper Florida Keys study site. Gray areas in (b), (c), and (d) and dashed areas in (d) indicate data within +/- 0.5m elevation change that were removed for minimum volume calculations. Red = elevation loss, blue = elevation gain in (b), (c), and (d).

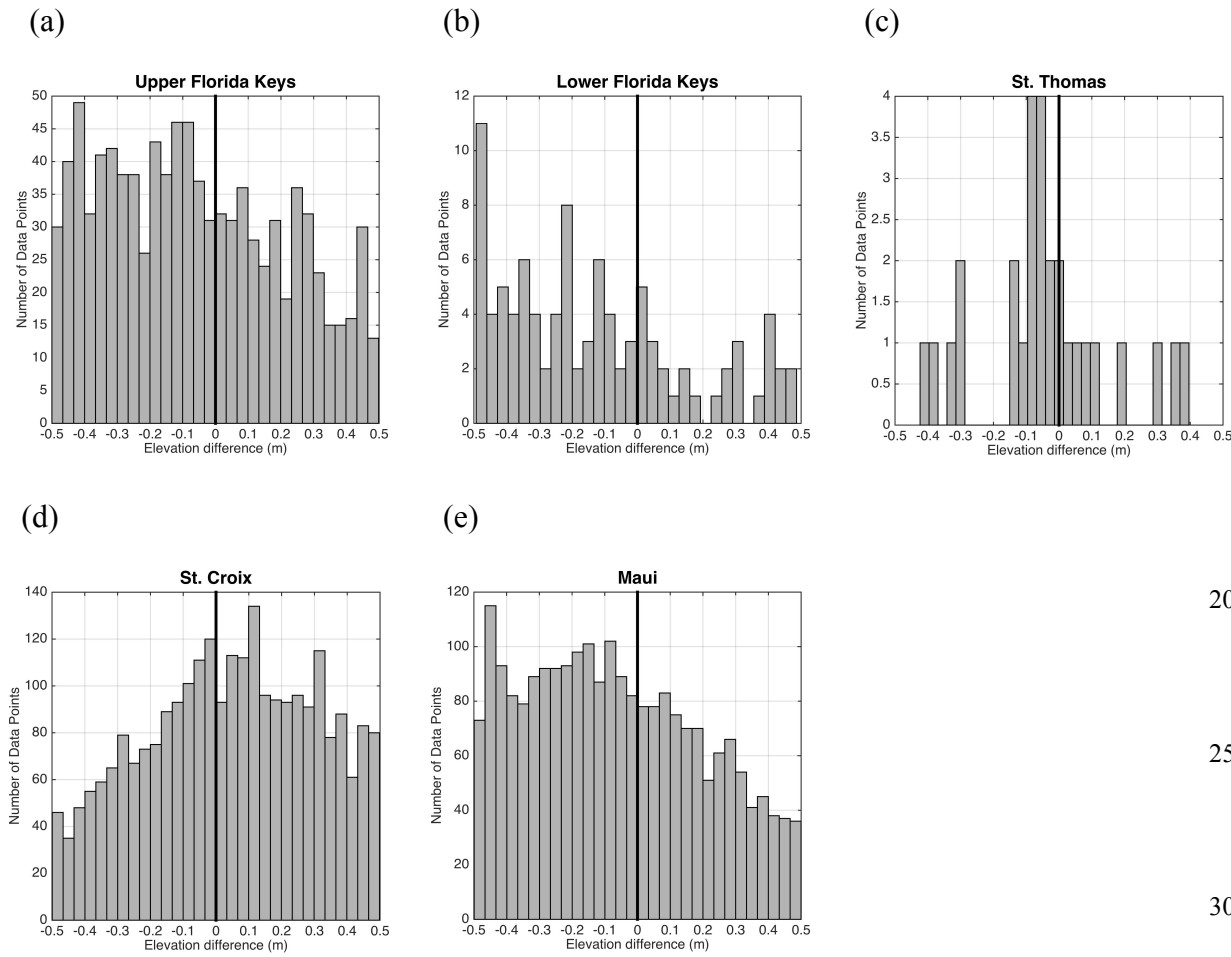

**Figure 4.** Histograms of elevation-change data for pavement analysis of the (a) Upper Florida Keys, (b) Lower Florida Keys, (c) St. Thomas, (d) Buck Island, St. Croix, and (e) Maui study sites.

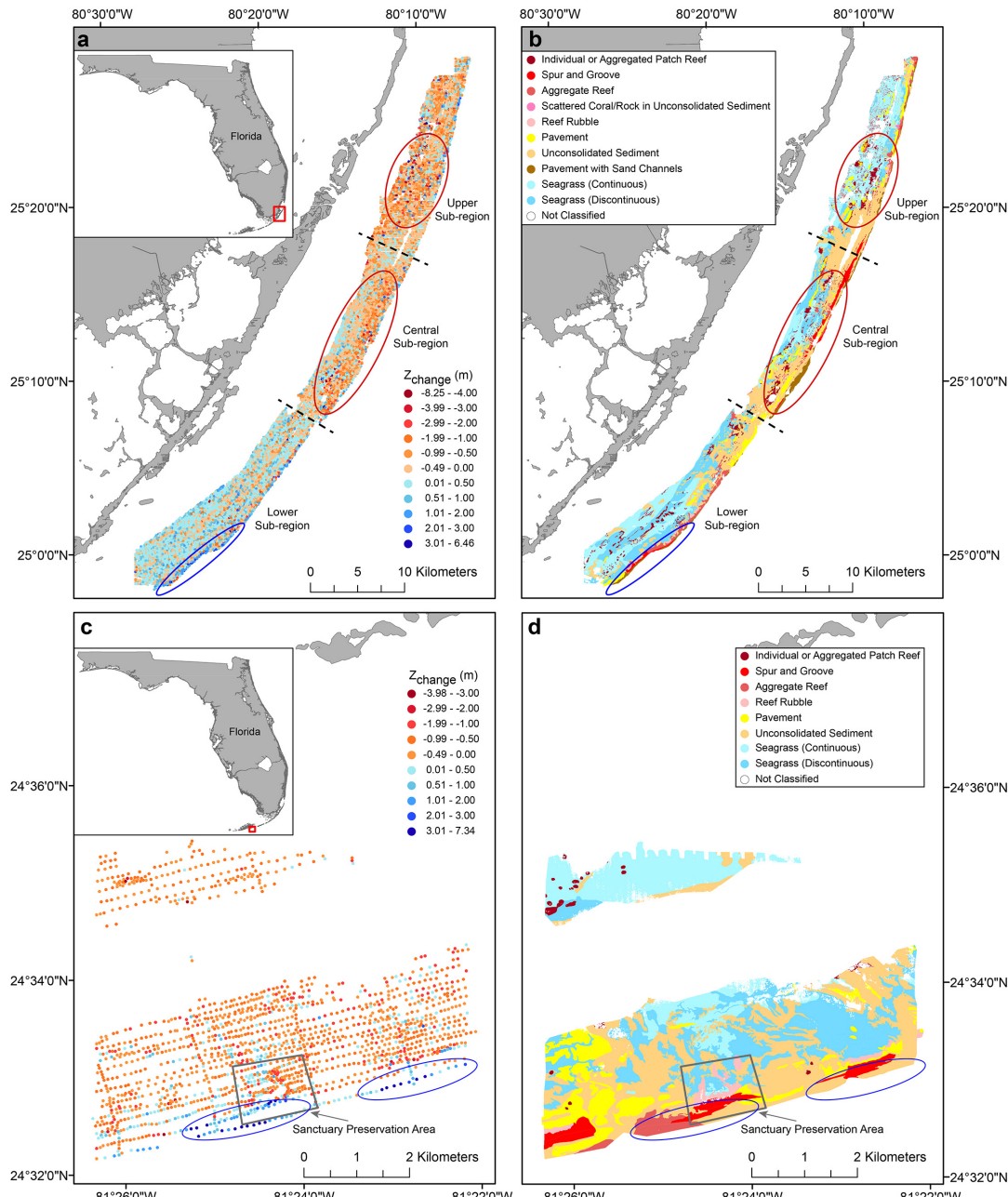

**Figure 5.** Elevation-change and habitat maps for the Upper Florida Keys (a and b) and Lower Florida Keys (c and d) over 68 and 66 years, respectively. Elevation values are differences in meters between modern and historical elevation data ($Z_{change}$ = modern elevation − historical elevation, see Tables S1 and S2). Negative numbers indicate elevation loss, positive numbers indicate elevation gain. Red and blue circles indicate areas of high erosion and accretion rates, respectively. Dashed lines = sub-regions. Habitat maps were modified from existing geo-databases (Table 3).

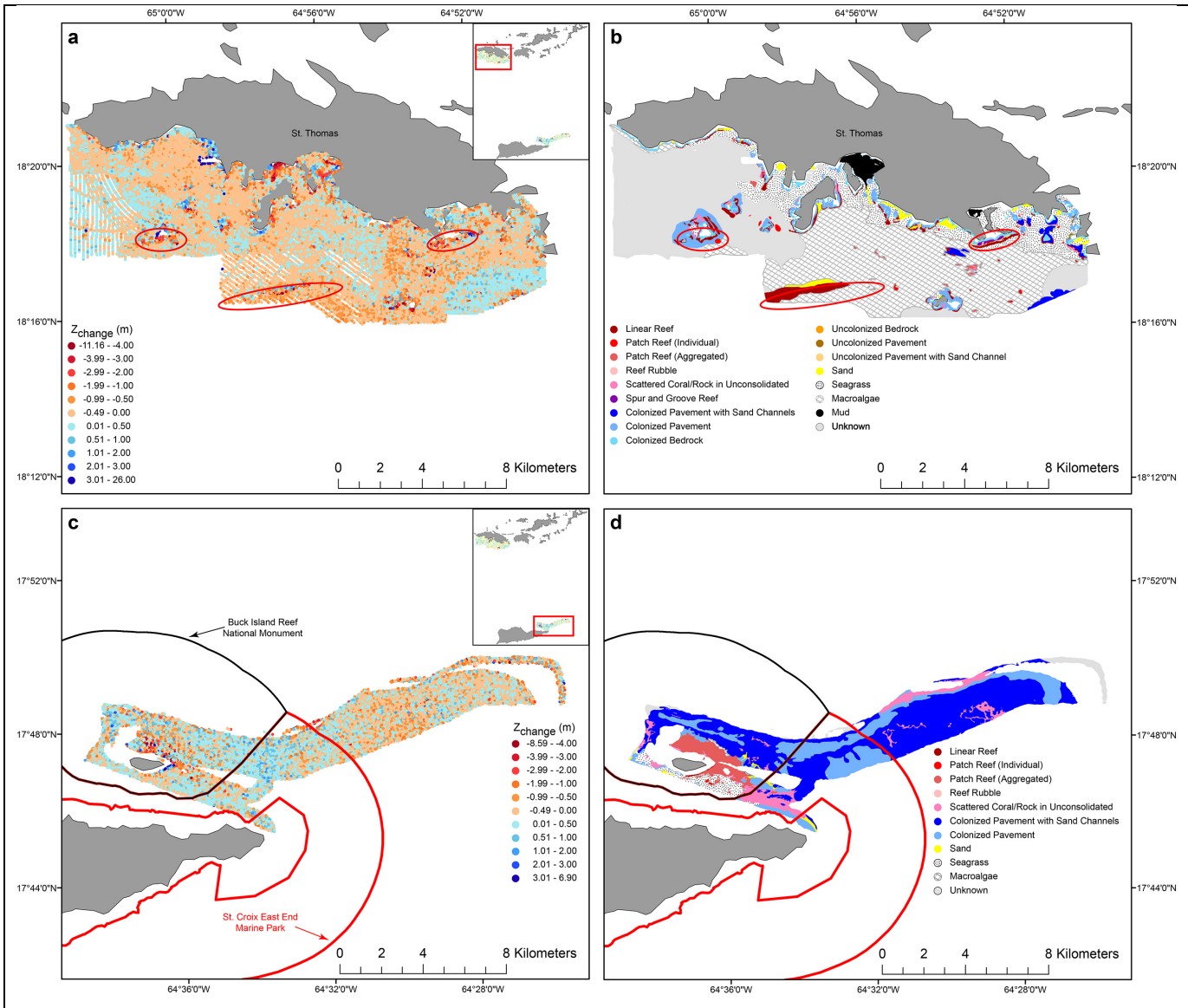

**Figure 6.** Elevation-change and habitat maps for the U.S. Virgin Islands. St. Thomas elevation-change over 48 years (a) and habitat (b). Buck Island elevation-change over 33 years (c) and habitat (d). Elevation values are differences in meters between modern and historical elevation data ($Z_{change}$ = modern elevation – historical elevation, see Tables S3 and S4). Negative numbers indicate elevation loss, positive numbers indicate elevation gain. Red circles indicate areas of high erosion rates. Habitat maps were modified from existing geo-databases (Table 3).

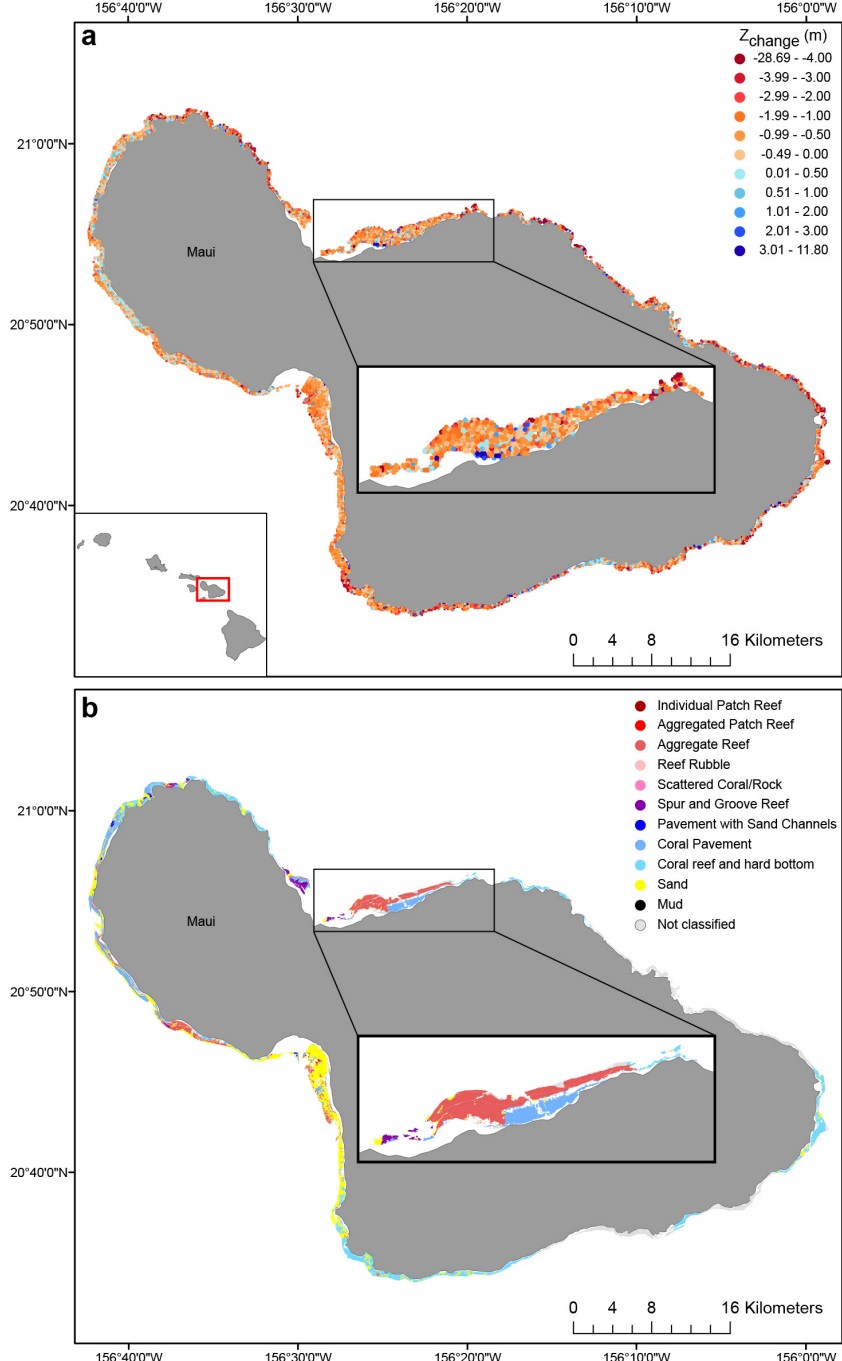

**Figure 7.** Elevation-change and habitat maps for Maui, Hawaii. Elevation-change over 38 years (a). Habitat (b). Elevation values are differences in meters between modern and historical elevation data ($Z_{change}$ = modern elevation – historical elevation, see Table S5). Inset shows magnified detail of the north central reef tract. Negative numbers indicate elevation loss, positive numbers indicate elevation gain. Habitat maps were modified from existing geo-databases (Table 3).

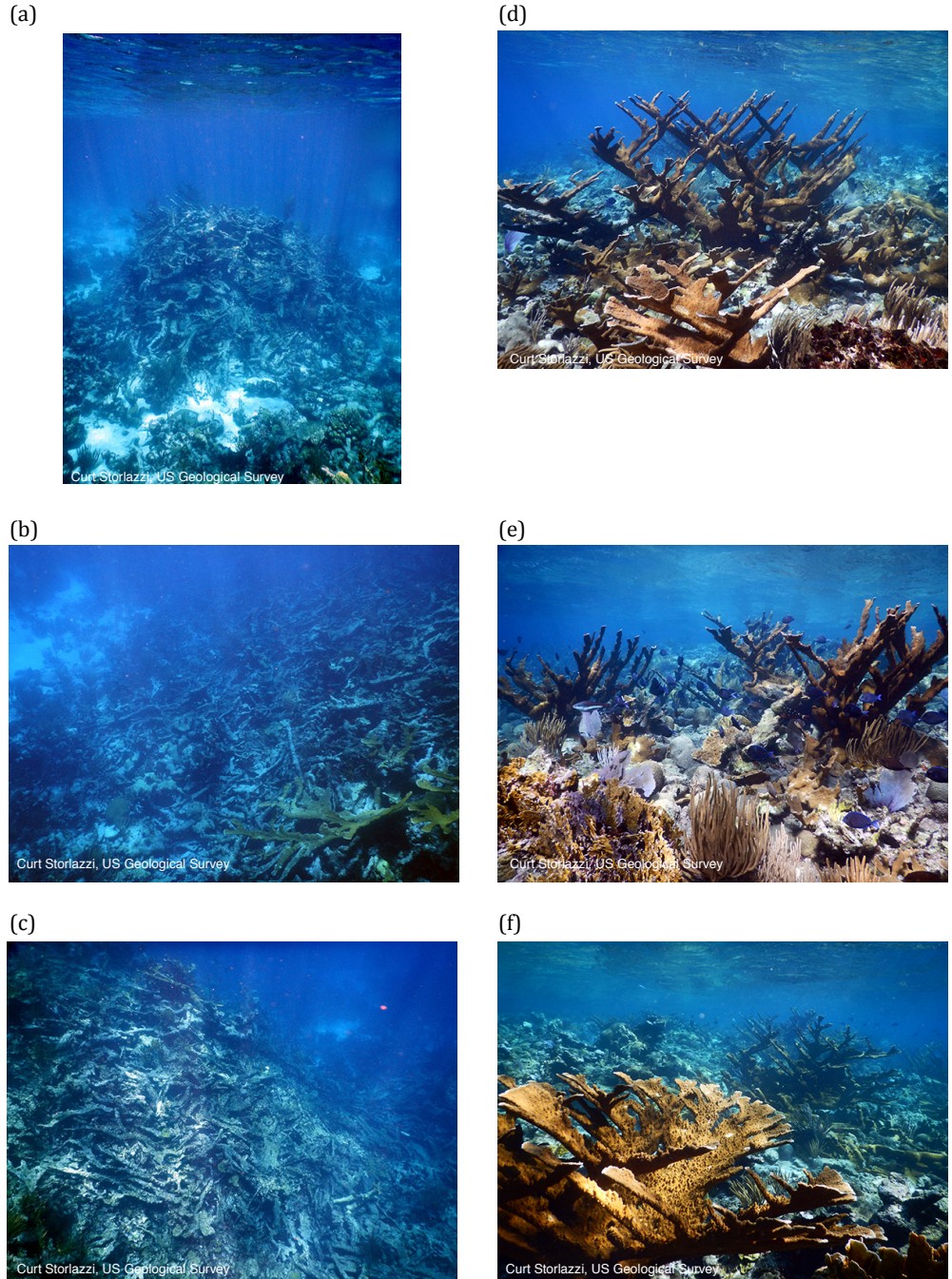

**Figure 8**. *Acropora palmata* coral rubble (a), (b), and (c) on the seafloor along the northeastern coast of Buck Island. Live *Acropora palmata* coral colonies (d), (e), (f) on the seafloor along the southeastern coast of Buck Island. Photo credit: C. Storlazzi, U. S. Geological Survey.

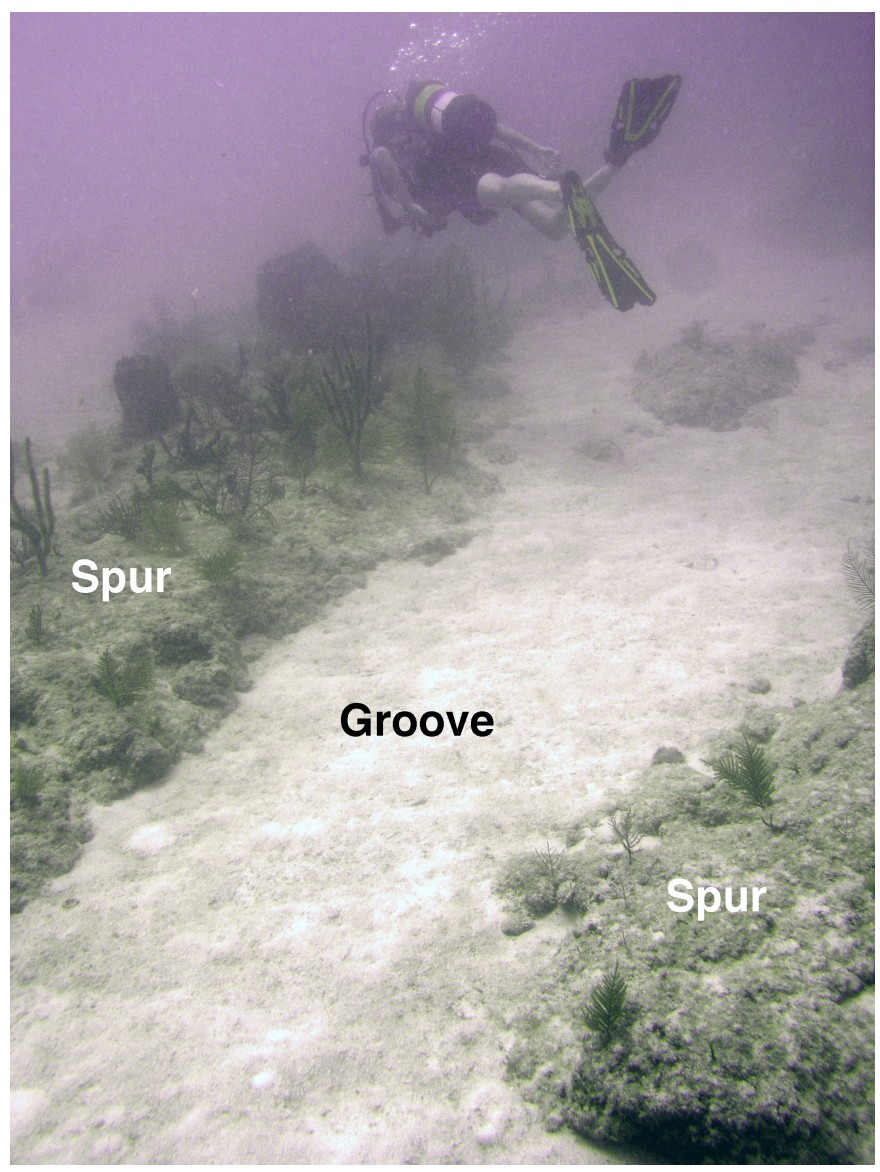

**Figure 9**. Spur-and-groove formation in-filled with sediments in along the Florida Keys reef tract. Photo credit: D. Zawada, U. S. Geological Survey.

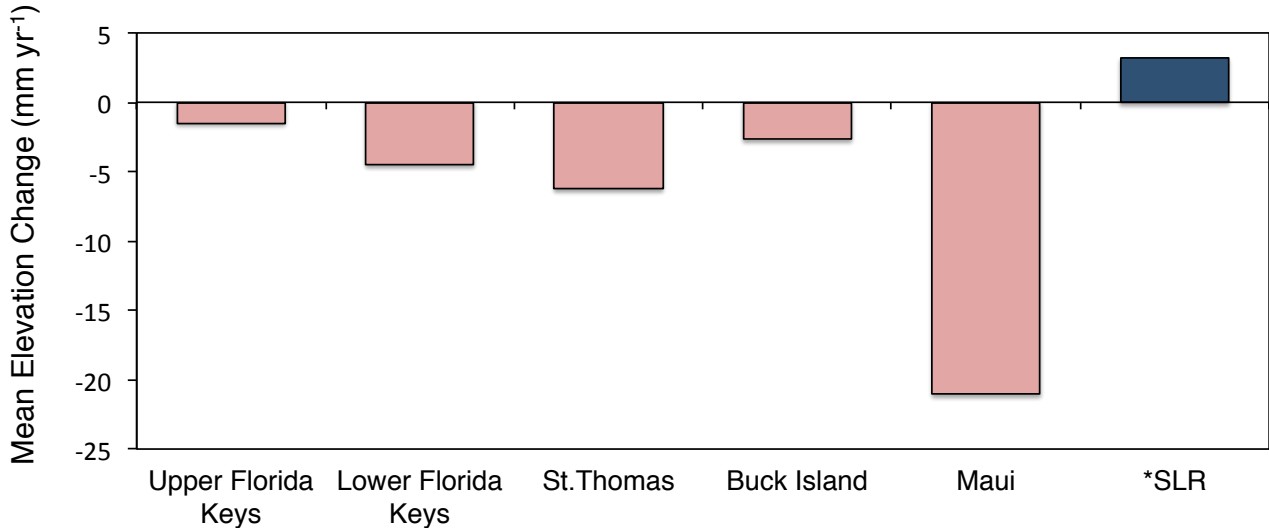

**Figure 10.** Mean annual elevation change calculated from total mean elevation change in meters (Table 8) divided by the time range of historical to modern bathymetric data for each study site. *SLR = global mean sea level rise of 3.2 mm yr$^{-1}$.

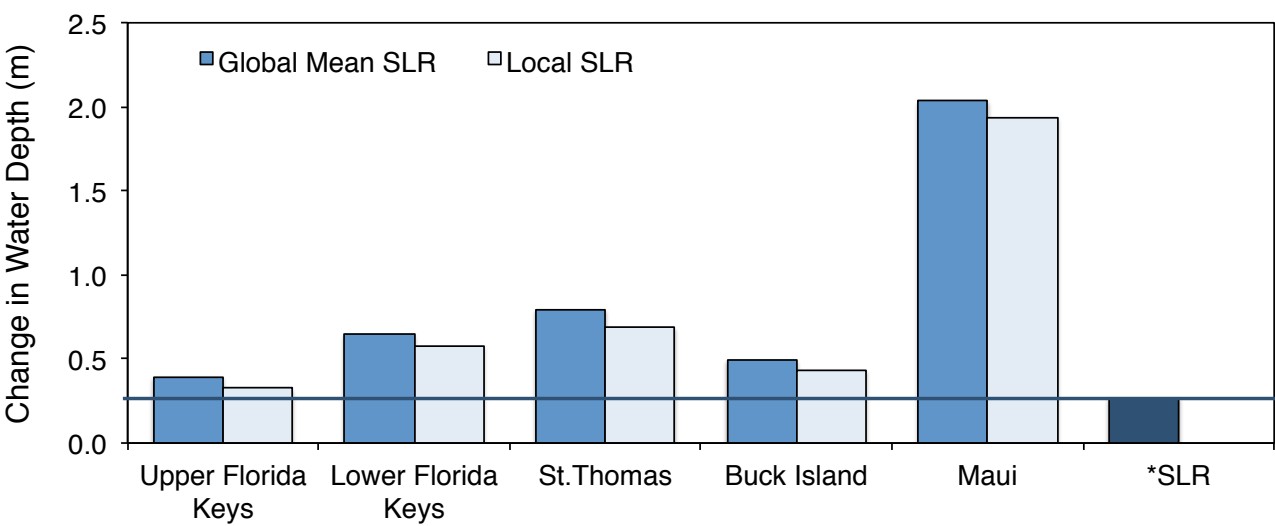

5    **Figure 11.** Projected water depth for the year 2100. Projected change in water depth by 2100 from combined impact of mean rates of seafloor elevation loss and rising sea level based on local (a) NOAA sea-level rise trends downloaded on August 4, 2016 (http://tidesandcurrents.noaa.gov/products.html) and global sea level rise trends (b), see Table S6. *SLR = water depth due only to global mean sea level rise of 3.2 mm yr$^{-1}$.