# Peer review of "Divergence of seafloor elevation and sea level rise in coral reef ecosystems"

_Biogeosciences, 2016_

## Referee Comment (RC1) · L. Montaggionni (Referee) · 20 Oct 2016

The aim of this contribution is to evaluate the effects of disturbances (resulting in erosion) that have affected coral reef systems over the last several decades and to project future evolution. Five reef systems selected from the Atlantic and the Pacific have served as test sites. The methods used herein to assess vertical reef growth and volume loss are highly sophisticated and a number of methodological subtleties are not realized by the reef geologist who I am. As a consequence, I really do not know what to think about the accuracy and value of the data. A colleague aware of this kind of innovative approach would be in a better poisition than me to discuss the value of the findings. However, I am a bit dubious about the accurary of the results (reef elevation and volume gains versus losses) based on the comparison between historical ( ?) and modern soundings records, despite the multitude of triturations and corrections made

to the raw data.

Therefore, my comments will be limited to some issues of detail.

Page 1, Line 1 : All dictionaries define Âń sea floor Âż as Âń the bottom of a sea or ocean Âż. Accordingly, using this term to describe the Pre-Holocene bed rock that underlies a coral reef body and forms its foundation is inappropriate.

Page 3, Line 17 : I question the use of Âń the number of people living close to the reef sites Âż as a parameter for anthropogenic impacts. Is the number of inhabitants the reflection of the local human activity (fishing, . . ..) ?

Page 12, Line 16, about Âń the coral-dominated habitats Âż : It would be useful to have some information about the living coral cover. This will inform the debate on the real state of health of each studied reef site.

Page 14, Line 8 : about the chronic erosion processes. These are natural processes affecting reef systems. Reef growth reveals to be the subtle balance between constructional and destructional processes. They occur continuously on both pristine and degraded systems.

Page 14, Lines 20 – 21 : It is clear that reefs that are located closed to urban areas are suffering significant deterioration. It would be intersting to compare these results with a reef system located in a remote and not inhabited area.

Page 14, Lines 26-27-28 : about the assertion Âń coral reefs in all three regions will be unable to keep up. . . . . . . . (Church et al., 2013) Âż. This is an overinterpretation of the data presented herein. Using mean rates of reef accretion established at the scale of the Atlantic and Pacific to infer future responses of reefs to the rise in sea level is not receivable. A number of previous studies worldwide indicated that vertical reef accretion varies from site to site in a given region. There, some reefs will be able to maintain pace with sea level, while others will be unable to compensate for sea level rise.

Page 15, Line 29 and Page 20, Line 23 : Please correct the reference : Neumann and Macintyre, 1985

---

## Referee Comment (RC2) · Anonymous Referee #2 · 2 Nov 2016

General comments:

Both the scientific quality of the paper and its presentation quality are generally insufficient.

The scope of the paper, as it is formulated pages 2 and 3, i.e. "measuring changes in seafloor elevation to assess and predict the impact of reef degradation on the vulnerability of coastal communities to sea-related hazards" is confusing. The title of the paper itself is also unclear. The authors announce that they address the coral reef issue and then they provide results on various habitats, including non-coralline habitats and even deep water offshore habitats (e.g., page 13, lines 3-5). Authors are unclear on what they measure, and on my view they fail in generating robust data (as both the data and the methods used lack accuracy. Thereafter, the presentation of the results and

their interpretation are confusing, as various types of processes are invoked to explain changes, with no specific process being robustly studied (e.g. page 13, lines 30-34). For example, the first paragraph of the Discussion Section clearly illustrates the wide (and unprecise) area covered by the paper (see page 14, lines 5-15).

Both the concepts ("change in seafloor elevation and volume" – in fact, it seems that the authors address "changes in shallow waters depth") and the method used (method "traditionally used to monitor seafloor changes", "use of historical bathymetric data from the 1930's to 1980's and LiDAR DEMs from 1990's to 2000's) –which are presented firstly in the introduction of the paper (pages 2-3) and then in the Methods Section (page 4) – are questionable and not accurate.

Concerning data and methods - How can bathymetric data from the 1930's to 1980's (constituting a single coherent period reflecting low anthropogenic impact?) be considered as a starting point (or reference) to "measure changes in seafloor elevation" and then be compared with data from the 1990's to the 2000's (= period reflecting high anthropogenic impact?). This raises several key questions. Firstly, how can the "magnitude of erosion" (page 3, line 9) be measured using such an approach that poses serious questions relating to the scientific quality of both the datasets and the method used. In other words, how can historical bathymetric data be compared with LiDAR data? The former (bathymetric data from the 1930's and next decades) do not have the required resolution for comparative measurements to be undertaken with LiDAR data. The low resolution of historical bathymetric data may generate significant errors in the results generated. Incidentally and curiously, no clear and complete information is provided by the authors on the resolution of the various datasets used in this study at the various study sites. Authors first indicate a 1 to 4 ̆am horizontal spatial resolution (this is a low resolution that do not allow the calculation of changes in the reef level) and then indicate a 11-12 ̆acm vertical resolution (which is questionable given the data used).

A second methodological problem is raised by the way anthropogenic impacts are considered in this study. (1) How can the "anthropogenic impact" only be measured by population numbers? In the present case (changes in water depth), it mostly depends on coastal and maritime human practices (sustainable/not sustainable). Major human activities, such as dredging in the substratum (should it be coralline or not) and extracting aggregate in particular, which may have occurred over the study period at some study sites and may have changed water depth, are not considered at all by the authors, which introduces a serious bias in the "elevation changes measured". Some parts of the paper, such as "However, greatest mean elevation losses occurred in coral-dominated habitats and near the central coastline where harbour and shipping channels existÂăÂż (page 13, lines 22-23) clearly indicate that not taking into account these human activities is problematic when assessing changes in shallow water depth.

Generally, this paper mainly appears as a "technical" paper that describes the GIS procedure applied to calculate changes in elevation, without addressing in an adequate way the conceptual, and the data and methods aspects raised. It seems that authors do not have the required background to address the complex scientific question that they have chosen to address. The technical procedure described on pages 4-6 is incomprehensible to me. Despite the fact that I failed in understanding this procedure, my feeling is that the method is not robust due to poor conceptual, and data and method, bases.

In different sections of the paper (e.g., page 13, line 1), the results obtained are correlated to generalities, e.g. on coral reef degradation, which is questionable. Results should be correlated to local data on reef health, including observed changes in living coral coverage, but not to worldwide observations.

The interpretation of the results generated is not satisfactory: for example, the authors mention hurricanes as key controls of changes in depth. This raises the question of "what is measured, either long-term changes related to climate change and sea level rise, or changes due to low-frequency high-magnitude events"? Once again, this makes the paper confusing.

[Figure]

Specific comments:

Page 2, lines 28 to 30 are incomprehensible: "measures of total system change in seafloor elevation and volume are required to accurately assess and predict the impact of reef degradation on the vulnerability of coastal communities to hazards caused by storms, waves, sea level rise and erosion".

Page2, lines 31-32: "we quantify the combined effect of all constructive and destructive processes on modern coral reef ecosystems by measuring regional-scale changes in seafloor elevation" is incomprehensible.

Page 3: "we adapted an elevation-change analysis method that has traditionally been used to monitor seafloor changes"

Page 7, line 25 "sediment thickness of the Holocene reef deposit": what do the authors talk about? Vertical sedimentation? Vertical Holocene reef building? The results exposed page 8, lines 11 to 14 for the Lower Florida Keys case study are incomprehensible to me. I do not understand how the authors "used a moder reef age of 6000 years and a constant erosion rate" to "compute the time required to completely erode the remaining Holocene reef down to the Pleistocene layer".

Page 8, lines 29-30: I am surprised to read that vertical errors would be comprised between 9.6 and 11.8 cm respectively, given what I know on LiDAR data and the horizontal error (1 to 4 m) applying to this study. More generally, I do not understand how vertical error estimation was conducted.

Page 12, lines 13 to 27– We understand that most study are not dominated by coral reefs, which means that this paper does not in fact address the pretended issue of reef response to changing environmental conditions. This suggests that the choice of study sites is not totally coherent with the objectives of the paper.

Page 12, lines 25 to 27: the conclusions drawn by the authors from the study of Buck Island correlates volume loss to sediment export. Both the results (volume loss) and

the interpretation of the results (sediment export) are unclear to the reader.

Page 14, lines 20-25: how can the authors convert "changes in elevation" into a "number of years of Holocene reef accretion"? This is not robust as coral reefs grow and erode over a given period, as a result of the complex imbricated processes driving both reef construction (i.e. construction) and sediment production (i.e. erosion allowing carbonate production).

Bottom of page 15-top of page 16: I do not understand how the results obtained by the authors can be compared to the results of previous studies conducted by C. Perry to attribute observed changes to specific drivers/processes.

Page 15, lines 19-21: the estimation that "the total reef volume could completely erode down to Pleistocene-bedrock-surface in approximately 1250 years" is not well-founded.

Page 15, line 33: "... reef systems... lack human impacts" is not correct in terms of style.

Page 15, lines 23-35: key references on reef islands future are not cited by the authors. See in particular the recent studies by Kench et al.

Page 16, lines 3-5: an assumption like "Modern carbonate production rates are an order of magnitude lower than Holocene averages (Perry et al., 2013), and are estimated to decrease by as much as 60% by mid-century (Langdon and Atkinson, 2005)" is far too general.

Tables 2 and 3 – The substrate categories included in these table are not presented and justified in the study. We additionally have not idea of the depth at which these habitats are situated.

The maps provided page 30 indicate a complex spatial distribution of gains and losses, which is not described in the paper. They also show that shallow habitats were not totally covered, suggesting that gains may have occurred in non-covered areas that may compensate observed losses in study areas. This is all the more to be considered

that the results obtained are contrasting (e.g. between the Central Sub-Region and the Lower Sub-region of the Florida Keys). Concerning the Florida Keys, curiously nothing is said in this paper about the dominant modes of planform change and about Keys' landward migration. This suggests that the general context that allows interpreting correctly the results is not presented and considered when analysing the results.

The results obtained in Saint Thomas, as shown by Map a page 31 mainly exhibit stability to limited elevation loss, if we consider grey and yellow areas. When we see this map, we are not convinced that elevation losses prevail, especially if we consider the error range. The same observation can be made when considering map c page 31 showing the situation of Buck Island (blue and yellow area are extensive).

———————————————————

---

## Author Comment (AC1) · 9 Nov 2016

We greatly appreciate Reviewer 1 providing constructive comments on the broader conceptual aspects of our paper related to his expertise, and for acknowledging that he is not a technical expert on the methods we applied in this study. We understand that the reviewer has doubts regarding the accuracy of results generated using historical and modern elevation data. However, in the absence of specific comments regarding our use of these data, we are uncertain as to how to address any specific concerns for the reviewer. We, therefore, provide a discussion to help clarify the methods by which we evaluated the data, calculated error, and additional analyses we performed to validate it's use.

Our study examined changes in seafloor elevation within 5 regions characterized by

extensive coral reef ecosystems that included reef flats as well as reef crest and slope habitats, adjacent (non-coral dominated) habitats such as seagrass beds, sand bottom, and hard bottom communities, and (in some cases) deeper water habitats. Our Maui study site included the coastal region surrounding the entire island to a depth of 20 m and examined changes in elevation from 1961/65 to 1999. Our St. Thomas study site included the entire southern coastline of the island including habitat out to 5 km offshore and examined changes in elevation from 1966/73 to 2014. The Buck Island study included an extensive area east of the island and to a depth of 37 m, and examined changes in elevation from 1981/82 to 2014. Our Florida Keys study site included outer reef track and surrounding habitats of the Upper Florida Keys and Lower Florida Keys, and examined changes in elevation from 1934/35 to 2002.

While the methods used to collect, process, validate and analyze the data used in this study are complicated, the general concept of the method is relatively simple. We have created a flow diagram that depicts the core processing steps (see Figure 1), and we will include this figure in our revised manuscript. We measured the differences between historical and modern seafloor elevation using data sets from the time periods indicated for each study site. Using this method provides a measure of the net change in seafloor elevation due to all of the constructive processes that cause accretion (or increases in seafloor elevation) and the destructive processes that cause erosion (decreases in seafloor elevation), and, therefore, provides a mechanism for assessing the combined impact of natural and anthropogenic processes that affect seafloor change. Although the general concept is straightforward, a number of very rigorous analyses were performed to test the validity of comparing historical and modern data sets; very conservative methods were used to calculate error associated with the methods; and the effect of that error on our results was quantified and reported. The key findings of this study indicate that all habitat types (including non-coral dominated habitats that are typically supported by carbonate sediment production on coral reefs) within these ecosystems have experienced a net loss in seafloor elevation causing a general decrease in mean seafloor elevation at the regional-scale.

The type of elevation change analysis we performed (using comparisons between historical and modern elevation data sets) has traditionally been used by coastal engineers to monitor seafloor changes such as sediment accretion and erosion, and migration of shipping channels, sand bars, mud banks and other seafloor features in coastal environments, and the methods are well documented. We provided a few references for examples of this type of work on page 3, line 4 (e.g. Byrnes et al., 2002; Taylor and Purkis, 2012; Byrnes et al., 2013). We present results from the first application of this method to coral reef ecosystems, and feel that this method is most appropriately used over the large spatial scales of this study.

Following published methods for use of historical elevation data sets, we performed visual inspection of all of the historical (Hsheet) data, relative to 2016 aerial and satellite imagery with a resolution of 1 m, to measure differences between stable coastal and geographic features (Lukas, 2014). For example, in the Upper Florida Keys we were able to locate a historic bridge that was surveyed in the 1930s and still exists today. We overlaid the georectified 1934 Hsheet on 2016 georectified Worldview imagery and used measuring tools in ArcMap to examine the offsets in this structure. The maximum offset of bridge boundaries between historical and modern locations was 4.8 m. See Figure 2, in this discussion, for that example.

This offset is similar to the sum of errors reported in our horizontal error analysis in Section 2.5.2, page 10, lines 16 - 28. Incidentally, we chose to sum the estimated sources of horizontal error (rather than to calculate a root mean square error, RMSE) because no horizontal error information was provided in the metadata for our historical data, and summing of sources of potential error (derived from the literature) provided a more conservative estimate of error. In other study sites, we were able to identify stable, rocky coastlines and/or man made features (such as shoreline berms) for these types of comparisons. In offshore areas, we examined areas in the historical data where the boundaries of large patch reefs were outlined by soundings when depths were too shallow to pass over the reef and deviations were made from linear transect lines.

There were no significant misalignments between historical and modern boundaries for reefs that experienced minimal erosion. We performed these visual inspections for all historical (Hsheet) data from all study sites and found similar alignment agreement.

Additionally, where it was possible for us to visit study sites (e.g., areas of the Florida Keys and Western Maui), we visually inspected the seafloor for erosion features that were consistent with trends identified by our analysis. For example, our data from the Florida Keys shows some offshore areas along relict spur and groove formations where seafloor elevation is increasing. Visual inspection of these areas showed infilling of spur and groove with sediments (see Figure 3 in this discussion).

We then performed an experimental exercise to determine the potential impact of horizontal error of up to 10 m (2 x the sum of our estimated horizontal uncertainties) on volume change calculations (see 'Section 2.5.2, page 11, lines 1 - 20) to account for estimated systematic error plus any additional, random error of up to an additional 5 m. We note that 10 m of horizontal error is consistent with horizontal error typically reported for data plotted at a 1:20,000 scale (Anders and Byrnes, 1991; Fletcher et al., 2003; Morton et al., 2004). These results indicate that horizontal error of up to 10 m and the resulting offsets in sounding points affects our volume calculations by 10% to 21% (depending on density of data points) and does not change the outcome or conclusion of our study. These results are consistent with reports that, over large areas (such as in our study), random errors largely cancel-out relative to change calculations derived from two surfaces (Byrnes et al. 2002).

We also performed an analysis of submerged substrate such as pavement or bedrock that may have some areas that showed no accretion or erosion to independently evaluate the potential vertical error associated with comparing historical and contemporary sounding methods. We did not include these results in the original manuscript because no subaqueous surface areas, including bedrock, are exempt from elevation change, and this analysis does not represent a true control. For example, our data indicate areas where pavement has been exposed due to erosion of overlying sediments, and

areas where pavement has been buried by deposition of sediments, growth of corals, etc. However, we include the analysis and results in this discussion as supporting evidence that validates comparison of these data sets. For the pavement analysis, we examined habitats labeled 'pavement' in the UFK and LFK, 'uncolonized pavement' in St. Thomas, 'colonized pavement' in Buck Island, and 'coral pavement' in Maui (see Table 2 in manuscript). We filtered the elevation-change data points for these habitats to include only data with elevation-changes between historical to modern points of less than +/- 0.5 m (1.65 x RMSETotal). We then calculated the average difference between these points and the corresponding standard deviations. See Table 1, in this discussion.

More than 50% of elevation-change points in the UFK, STT and BI showed elevation changes within our 1.65 x RMSETotal of 0.5 m, and standard deviations for these location were all within our RMSETotal of 0.29 m. 17% and 23% of points were within +/- 0.5 m for the LFK and Maui, respectively, with standard deviations below RMSE-Total of 0.29 m. Average differences between historical and modern elevation data in these locations were very small (ranging from 3 to 6 cm) indicating that vertical resolution and accuracy of historical and modern elevation data are, in fact, comparable. However, standard deviations were large, confirming that all areas of pavement are not stable and, therefore, should not be considered a true control. Further examination of histograms for the pavement analysis data support the general conclusions for each of our study sites (see Figures 4a – e, in this discussion).

Histograms for the UFK, LFK, STT and Maui are all skewed toward negative values, consistent with regional-scale trends of seafloor elevation loss and export of sediments at these sites. While Buck Island (St. Croix, d in the figure) shows a more even distribution of points among elevation losses and gains consistent with the lower regional-scale mean elevation losses observed at this study site. Notably, of the five study sites, the 'colonized pavement' we examined for BI was the only pavement habitat to show a very small increase in elevation that could be due either to coral growth on the pavement or burial of pavement by sediments. The latter supports our suggestion that less sediment is exported from this system. While we feel that these results are consistent with our general conclusions and further validate our results, we do not feel that they adequately serve the purpose of a control.

With respect to historical data, the U.S. Office of Coast Survey imparted strict standards on collection of data from the 1800's to 1950's that, in fact, allowed for less error than later in the 1900's when electronic means of collecting sounding data replaced lead lining and poling (Shalowitz, 1964). For data collected by the Coast Survey from the late 1800's to 1950's, maximum allowable differences at crossing data points was 0.06 m for depths less than 4.6 m, and 0.46 m for depths between 22 and 29 m (Byrnes et al., 2002). Lead line and poling resolution reported for the historical data sets was 0.15 m (similar to that of LiDAR), while fathometer resolution was 0.3 m. Furthermore, sounding poles were used in depths less than 3 m (where many shallow patch reefs are located). Many Hsheet descriptive reports indicate that the water was clear enough during surveys to see the bottom (which improves accuracy of the measurement). The most common error likely to occur during use of lead lines or sounding poles was over-estimation of water depth due to angling of the line or pole as currents move the boat past the point of measurement. Overestimation of historical water depth would erroneously decrease elevation losses calculated using our methods. Therefore, it is more likely that our erosion estimates are underestimated rather than overestimated due to lead line and poling techniques. Furthermore, historical data sets from the same time frames that used similar collection methods are routinely used by the U.S. Army Corps of Engineers for coastal engineering projects, seafloor evolution and sediment budget studies, and to examine migration of coastal seafloor features over much smaller and deeper geographic regions than our study and have been proven for this purpose.

The maximum error was reported in each historical Hsheet description, and was quantitatively determined at the time the surveys were conducted by repeat surveys and cross tracklines. The maximum reported vertical difference between all of the repeat

and cross-track line survey points (including deep water points) that were collected by the original surveyors for the purpose of determining historical data quality of the data sets we used in this study was 0.46 m (the standard set by the U.S. Office of Coast Survey); however, this is not a measurement of accuracy. We, therefore, reanalyzed all of the historical repeat and cross-track line survey data from the original historical data sets to perform a more rigorous systematic vertical error analysis (as opposed to simply using the reported maximum difference) of the historical sounding measurements that was not originally performed in the historical data QA/QC (see page 9, lines 2 – 29 in manuscript). We calculated the mean difference in depth between all pairs of repeat measurements and used these data to generate an RMSE for historical sounding data. Results from this analysis are termed RMSESounding in Table 4 (manuscript), and range from 2 to 37 cm. We also calculated the error associated with transformation of each data set to a common vertical datum using the latest release of VDatum software (version 3.6) from the NOAA National Geodetic Survey. The maximum cumulative uncertainty for VDatum regions of South Florida and the U.S Virgin Islands is 9.6 cm and 11.8 cm respectively, and the cumulative uncertainty values calculated for our data sets were 8.1 cm and 11.4 cm. No vertical transformations were performed on the Maui data because both historical and modern data sets were already in the common tidal datum of MLLW.

We then performed an analysis of vertical error using U.S. Federal Government Standards (see 'Vertical error analysis' section 2.5.1). Our vertical error analysis included error terms for:

1) modern LiDAR data sets (RMSELiDAR). LiDAR uncertainty was determined by independent validation of airborne LiDAR measurements with in-water acoustic sounding measurements performed at the time that the LiDAR data was collected and reported in the metadata for these data sets. 2) historical data sets (RMSESounding) as determined from our analysis of repeat measurements that were performed by the original surveyors at the time of data collection, and 3) uncertainty from transformation of data

to a common vertical datum as calculated using VDatum (RMSEVDatum) for each individual data set.

These uncertainty values specific to each data set (Table 4 in manuscript) were included in our calculations of RMSE (see page 8, equation 1 for RMSETotal).

Our average RMSETotal (Table 4 in manuscript) for all study sites was only 0.29 m. However, to take an even more conservative approach, we multiplied our RMSETotal by a factor of 1.65 to encompass 90% of the variance in our data and generate a more conservative RMSE of 0.48 m that we rounded up to 0.5 m; and we used this value to set minimum and maximum bounds in our volume calculations (Table 1 in manuscript). The minimum volume change values that we report in Table 1 were calculated by only including elevation changes that exceeded the range of -0.5 to +0.5 m to provide a very conservative estimate of volume change. These very conservative minimum volume change values also support our conclusions of net seafloor erosion at all study sites (Table 1 in manuscript). Additionally, the minimum elevation change unit in Figures 1 – 3 (in manuscript) is equivalent to our 1.65 x RMSETotal of 0.5 m to make is easier for the reader to visualize the data that was removed from our minimum bound volume calculations. Furthermore, our average RMSETotal of 0.29 m is half the reported error value for studies that used nautical chart data to create digital elevation models for examining the response of reefs to sea level rise (e.g., Leon et al. 2013 and Hamylton et al. 2014). We specifically did NOT use nautical charts for our analysis because they often represent smoothed, interpolated surfaces that are not as accurate as the sounding data from Hsheets.

The small (less than a meter) elevation changes reported in Table 1 (in manuscript) are mean values over the whole ecosystem or mean values for the habitat type (Tables 1 and 2 in manuscript). It must be noted that our mean elevation changes represent the net change encompassing all negative (elevation losses) and positive (elevation gains) values. We note that the standard deviations are large because individual measurements of elevation change were often very large (greater than 1 meter, see minimum

and maximum elevation loss and gain data by habitat in Table 2, and in Figures 1, 2 and 3). We analyzed 118,710 data points from 59 habitats across all study sites. Review of the 'Mean loss' column in Table 2 (in manuscript) shows that of the 59 individual habitat analyses we performed, only one habitat in St. Thomas (reef rubble), and one habitat in Buck Island (seagrass) showed mean losses less than our RMSETotal of 0.29. Only two habitats in the Upper Florida Keys (discontinuous seagrass and not-classified), five habitats in St. Thomas (seagrass, macroalgae, unknown, and uncolonized pavement), five habitats in Buck Island (colonized pavement with sand channels, scattered coral/rock in unconsolidated sediment, reef rubble, colonized pavement and seagrass), and two habitats in Maui (rubble and mud) showed mean losses less than 0.5 m, or 1.65 x RMSETotal that encompasses 90% of the variability. Therefore, mean losses in 97% of the habitats we analyzed were greater than our vertical RMSETotal of 0.29 m, and 77% of the habitats we analyzed showed mean losses greater than 1.65 x RM-SETotal or 0.5 m. In several cases, and particularly associated with coral-dominated substrate, mean losses exceeded 1 meter. Conversely, if one examines only those locations within habitats where elevation gains occurred ('Mean gain' column in Table 2), 92% of the 60 habitats show showed mean gains greater than our RMSETotal, and only 60% showed mean gains greater than our 1.65 x RMSETotal. Again, standard deviations are large for these data because they reflect the true nature of losses associated with a highly topographically complex system.

We feel that we have proven the validity of our results and use of historical and modern data sets for our analyses with our expanded error analysis and our use of a very conservative RMSE for data calculations. Our conclusions regarding loss of seafloor volume are based on actual measurements of elevation-change shown to be statistically significant in over 90% of the habitats we analyzed, and that account for all of the processes causing elevation loss in these regions. We have found no prior studies that fully account for all processes causing elevation change in coral reef ecosystems; and we recognize that total erosion at the regional scale has likely been underestimated in prior studies as a result. We have accounted for uncertainties, reported actual losses
that have already occurred, and made no assumptions regarding spatial and temporal scaling or distribution of elevation changes as are often made when modeling erosion from carbonate budgets. Our projections are based only on our measured rates for what has already occurred in these systems, and, therefore, likely underestimate future losses because they do not account for any future decreases in carbonate production, increases in carbonate dissolution and bioerosion, or increases in physical erosion due to increases in frequency and magnitude of storms. Similarly, our projections of increases in water depth are based on current, measured rates of seafloor elevation change and current rates of sea level rise. The water depth projections are also likely underestimated for the same reasons that seafloor volume loss may be underestimated, and because we have not accounted for any projected increases in rates of global sea level rise.

Responses to specific comments from Reviewer 1 (RC1).

RC1: Page 1, Line 1 : All dictionaries define Â′n sea floor ÂËŹz as Â′n the bottom of a sea or ocean ÂËŹz. Accordingly, using this term to describe the Pre-Holocene bed rock that underlies a coral reef body and forms its foundation is inappropriate.

The page and line reference for this comment refers to the manuscript title, which makes no reference to Pre-Holocene (or Pleistocene) bedrock. In Section 2.4 'Lower Florida Keys – volume to Pleistocene bedrock', we discuss the thickness of the Holocene reef layer lying above Pleistocene bedrock, but do not use the term 'seafloor' in this section. In the Discussion, Page 15, Lines 15 - 25, we discuss the results of our Holocene reef-volume-change analysis and provide an estimate of how long it would take for the remaining Holocene reef to erode to Pleistocene bedrock, assuming a constant rate of erosion, but do not use the term 'seafloor'. We make no explicit use of 'seafloor' in reference to Pre-Holocene bedrock. We are a bit confused as to where we have used the term inappropriately. Please clarify so we can correct or explain.

RC1: Page 3, Line 17 : I question the use of Â′n the number of people living close

to the reef sites ÂËŹz as a parameter for anthropogenic impacts. Is the number of inhabitants the reflection of the local human activity (fishing, : : :.) ?

We agree that population size alone does not reflect the total impact of human activity, which is why we stated that population size serves as the "simplest first-order parameter" for estimating anthropogenic impacts. Later on Lines 18-20, we further acknowledge that "Full analysis of anthropogenic impact factors (e.g., urban extent, developed land, terrestrial water run-off, water quality, etc.) is beyond the scope of this paper." The point we are making is that among our 5 study sites, the ones in proximity to higher human population densities exhibited larger net loss of seafloor elevation.

RC1: Page 12, Line 16, about Â'n the coral-dominated habitats ÂËŹz : It would be useful to have some information about the living coral cover. This will inform the debate on the real state of health of each studied reef site.

Thanks for this suggestion. We will provide information on percent live coral cover for those areas where it is available in our revised manuscript.

Page 14, Line 8 : about the chronic erosion processes. These are natural processes affecting reef systems. Reef growth reveals to be the subtle balance between constructional and destructional processes. They occur continuously on both pristine and degraded systems.

We fully agree and make no statement to the contrary. In the cited paragraph, our aim was to make the point that our estimates of seafloor elevation change reflect the net result of all constructional and erosional forces affecting these regions.

RC1: Page 14, Lines 20 – 21 : It is clear that reefs that are located closed to urban areas are suffering significant deterioration. It would be intersting to compare these results with a reef system located in a remote and not inhabited area.

We agree. With the exception of Buck Island (which is uninhabited, but not remote), we were unable to locate sufficient historical and contemporary bathymetric data to perform our analysis in more remote, uninhabited locations. We would be very interested in such a comparison as well.

RC1: Page 14, Lines 26-27-28 : about the assertion Â′n coral reefs in all three regions will be unable to keep up: : :: : :: : : (Church et al., 2013) ÂËŹz. This is an overinterpretation of the data presented herein. Using mean rates of reef accretion established at the scale of the Atlantic and Pacific to infer future responses of reefs to the rise in sea level is not receivable. A number of previous studies worldwide indicated that vertical reef accretion varies from site to site in a given region. There, some reefs will be able to maintain pace with sea level, while others will be unable to compensate for sea level rise.

This is a fair point. We will re-write that passage to indicate that the coral reef ecosystems at our study sites have lost pace with sea level rise, but local variability in coral growth rates may enable other reefs in these regions to keep pace with sea level rise. We did not use mean rates of reef accretion for the Atlantic and Pacific to infer future responses of reefs. Rather, we used these rates to infer/estimate the amount of reef accretion (in years) that has already been lost in the past few decades at our study sites.

RC1: Page 15, Line 29 and Page 20, Line 23 : Please correct the reference : Neumann and Macintyre, 1985

Thank you for pointing out this mistake. We will correct it in our revised manuscript.

References Anders, F.J. and M.R. Byrnes.: Accuracy of Shoreline Change Rates as Determined from Maps and Aerial Photographs. Shore and Beach. January. p. 17-26, 1991. Byrnes, M. R., Baker, J. L., and Li, F.: Quantifying potential measurement errors and uncertainties associated with bathymetric change analysis, Vicksburg, MS, ERDC/CHL CHETN-IV-50, 17 pp., 2002.
 Shalowitz, A. L.: Interpretation and Use of Nautical Charts, in: Shore and Sea Boundaries, U. S. Government Printing Office, Washington, DC, 269-355, 1964. Fletcher, C., J. Rooney, M. Barbee, S.C. Lim, and B.

Richmond.: Mapping Shoreline Change Using Digital Orthophotogrammetry on Maui, Hawaii. J. Coastal Res. SI-38, p. 106-124, 2003. Lukas, M. Cartographic Reconstruction of Historical Environmental Change.: Cartographic Perspectives. n 78. DOI: 10.14714/CP78.1218, 2014. Morton, Robert A., Miller, Tara L., and Moore, Laura J.: National assessment of shoreline change: Part 1: Historical shoreline changes and associated coastal land loss along the U.S. Gulf of Mexico: U.S. Geological Survey Open-file Report 2004-1043, 45p., 2004.

[Figure]

[Figure]

Figure 1. Flow diagram of the core data processing steps used to compute seafloor elevation and volume changes.

**Fig. 1.** Figure 1. Flow diagram of the core data processing steps used to compute seafloor elevation and volume changes.

[Figure]

[Figure]

Figure 2. Example of visual inspection of historical Hsheets relative to modern aerial
imagery as a primary 'check' for horizontal alignment. (a) overlay merging the (b) 2016
aerial imagery with (c) 1934 Hsheet. (d) ArcMap measurement lines.

**Fig. 2.** Figure 2. Example of visual inspection of historical Hsheets relative to modern aerial
imagery as a primary 'check' for horizontal alignment. (a) overlay merging the (b) 2016 aerial
imagery with (c) 1

[Figure]

Figure 3. Infilling of spur and groove formation on the outer reef tract of the Upper
Florida Keys.

**Fig. 3.** Figure 3. Infilling of spur and groove formation on the outer reef tract of the Upper
Florida Keys.

[Figure]

Figure 4. Histogram analysis of pavement elevation-change data.

**Fig. 4.** Figure 4. Histogram analysis of pavement elevation-change data.

| Study site | Total # habitat data points | # Points within +/- 0.5 m elevation change | Average elevation difference (m) | Stdev. |
|---|---|---|---|---|
| UFK - pavement | 1901 | 958 | -0.06 | 0.27 |
| LFK - pavement | 198 | 34 | -0.03 | 0.1 |
| STT – uncolonized pavement | 33 | 28 | -0.04 | 0.2 |
| BI – colonized pavement | 3286 | 2543 | 0.04 | 0.26 |
| Maui – coral pavement | 4412 | 1013 | 0.05 | 0.14 |

Table 1. Results of pavement elevation data analysis. Average elevation difference between modern and historical elevation data (modern – historical data) and standard deviation are reported for those data points within +/- 0.5 m (or 1.65 x RMSE$_{Total}$).

**Fig. 5.** Table 1. Results of pavement elevation data analysis.

---

## Referee Comment (RC3) · D. Hubbard (Referee) · 22 Nov 2016

Overall, I am impressed by this paper and think that it is an interesting attempt to elevate monitoring to something more than "counting corals". However, I am concerned that the likely variability in sources of substrate change were probably much more different from site to site than has been characterized. I could be wrong, but I suspect that bioerosion is less of a factor than is represented here... and is more likely declining at most Caribbean sites. While there is an effort to address site-to-site variability, I am not convinced that the relative roles of simple bioerosion, large-scale rugosity loss and export by storms have been adequately considered. I would like to see this paper appear in print, if only for the valuable data set. However, I am concerned that the explanations of the measured patterns is a bit oversimplified and relies too much on the mechanisms proposed. I, therefore, provide some over-arching thoughts below in the

hope that the authors can perhaps think a bit more about other possible explanations for the patterns they observed. Accordingly, I make a few general observations below that will hopefully be useful.

Comments:

Like Reviewer 1, I am not well versed in the GIS and data transformation methods utilized in this study. However, I am familiar with the vagaries of older hydrographic surveys. On the latter front, I am willing to accept their characterizations of (the direction of?) change in substrate level as the differences between sites are probably sufficient to overcome any stated errors. However, in my experience, the notes on smooth sheets leave us with a need to make defensible assumptions about a) the reliability of substrate characterization (and its stability) and b) how processes that potentially influence elevation change might differ from site to site. I have limited my comments to the latter, based on areas in the manuscript where I have experience in either the specific habitats or the processes that might contribute to the patterns described.

Before I start, I do have one comment on style. I am not qualified to comment on the statistics of the methods or the assumptions made in the GIS transformations and map algebra. Nevertheless, a more reader-friendly explanation on that front would make the paper more accessible to a broader audience. The paper in its present form is a wealth of information on methods for those inclined to apply them to other sites. However, those people are probably going to be less well informed on the evolution of carbonate substrates. Conversely, those with intimate understanding of carbonate cycling are going to be unable to tie their knowledge to the details of the methodology used here. I am in that latter group and would suggest that the minutiae of the transformations and GIS tools could be better placed in the Supplemental Materials.

The following ae my general thoughts based on elements of carbonate cycling that could lead to conclusions other than those drawn here. While I am willing to accept the numerical changes in substrate elevation, I am somewhat less comfortable with

assumptions about the degree to which they are related to bioerosion and the ensuing removal of sediment.

Biorosion versus structural reorganization – In the discussions, there is an apparent conflation of bioerosion and spatial heterogeneity. The paper by Alvarez-Fillip et al. (1990) that is cited to document the role of increased bioerosion focused on the loss in architectural complexity (aka rugosity) and its causes – not bioerosion. In the paper, they attributed the initial reduction in reef rugosity to the loss of acroporids and the second decline in rugosity to a loss of massive species following bleaching. It seems reasonable to assume that an increase in susceptible substrate could increase bioerosion. However, Alvarez-Fillip et al. focused on the loss of rugosity which, in the case of A. palmata, is more easily explained by physical toppling/breakage and incorporation of fragments into a broad, cemented pavement. The interval of measured elevation changes included the loss of A. palmata. It, therefore, seems likely that this could have played a greater role than the removal of bioeroded sediment in the changes described in the manuscript. Alvarez et al also pointed out that the loss of Diadema logically reduced bioerosion despite the greater availability of "bioerodable substrate". Likewise, in many (most?) Caribbean and western Atlantic sites, parrotfish populations have been decimated, further reducing the potential for bioerosion by grazers. The remaining option is infaunal bioerosion by sponges, worms, etc. However, unless there is a very significant increase in organic availability, the likelihood of that being significant seems unlikely.

It is interesting that at one of their sites (Buck Island), Bill Gladfelter proposed two threats to reef building in a 1977 report to the Park Service: 1) the loss of carbonate production if WBD increased, and 2) the possibility that protection of parrotfish might significantly increase bioerosion to the point where it could overwhelm even productive reefs. This would suggest that increased bioerosion by grazing fish could lead to detrimental increase in bioerosion. In the latter scenario, increased grazing becomes a problem only in protected areas where grazing fish have increased (like the FKMS, one

of the described sites where increased bioerosion might be a reasonable culprit). Elsewhere in the Caribbean, parrotfish populations have been decimated. In combination with the loss of the major grazing urchin, a wholesale increase in bioerosion capacity seems unlikely. Lost calcification ability would decrease accretion, but does not seem like a driver of net erosion unless bioerosion increases – a pattern that has not been documented at all sites.

So, that leaves us with export. As the paper points out, good data on export are rare. On page 2, Moses (2009) is cited for measuring sediment export from reefs, but I could find no measurements in that paper. Kench and McLean provide an estimate of transport potential through hoa in Indian Ocean atolls. However, the results are based on theoretical calculations and there is no effort to tie sediment to specific sources (e.g., bioeroded sediment, beaches, lagoons) or sinks (loss to lagoons vs export from the platform). What is, therefore, critically important is a reliable estimate of export inasmuch as volume must be exported from the system to trigger system-wide elevation loss... bioerosion just converts carbonate from solid substrate to sediment. In the latter case, we must remember that sediment has a much lower bulk density than solid carbonate substrate. Thus, increased bioerosion without export would reduce the volume of solid substrate but would turn this into a sediment pile with something akin to twice the net volume. Thus, increased bioerosion without export would result in substrate elevation; not lowering. A scenario based solely on increased bioerosion seems inadequate to explain the measured patterns.

Unfortunately, there has only been a few careful measurements of sediment export in the context of a reef-wide budget. Perry and various co-authors use our ratio (Export $\sim$ 50% of total bioerosion) from the north coast of St. Croix to characterize this in every one of their budgets. It is naïve to think that all reefs in all oceans have the same energy regime (the driver of export) – or that changes in energy regime is offset by proportional shifts in bioerosion to maintain the 50% value that is used throughout. With increasing storminess, sediment export looms as the single largest unquantified

variable. Therefore, export can only get more significant in the budgeting attempted in this paper.

In section 3.2, the paper acknowledges the difference between bioerosion and changes in structural complexity. How good the conclusion will be is going to depend on how well one can distinguish between the two as potential drivers of elevation change. The conclusions presented here seem to suggest 1) an ability to reliably distinguish between the two mechanisms and 2) an overwhelming importance of simple bioerosion over combined changes in export and reduced structural complexity following the loss of biological constructors.

Anthropogenic drivers of change - Using population as a proxy for anthropogenic impact seems overly simplified. Numerous recent papers have shown that some of the greatest reef losses occur due to warming/acidification at great distances from any recognizable urban stressors. I can't find the specific papers, but there has been quite a bit of discussion on the NOAA listserve about papers that show just this. While I am not in the midst of the debate over local versus global drivers of change and their implications for management, this proxy seems a bit simplistic. On a more specific point, the manuscript discusses the idea of proximity to anthropogenic areas to explain the positive elevation change in the lower Keys. Couldn't this also be due to separation from the inimical cold bank water allowing for higher calcification rates? In this vein, limited core data from the Keys seem to suggest that the "demise" of the reef tract likely started 4-5,000 years ago as Florida Bay flooded, triggering inimical (cold) water export onto the reefs. In contrast, the reefs around Buck Island enjoyed continuous building throughout this period as there was no similar source of stress.

All of this would suggest that these two areas have had very different exposures to natural stresses; this would presumably make for very different susceptibilities in more recent times when increasing anthropogenic stress is set up as the main driver. It may also be noteworthy that the sediment thicknesses in these two areas are different and there is evidence that sediment retention around Buck Island (much higher wave

energy and susceptibility to both storm damage and sediment export) may tend to be less than is the case in the Keys. If the latter is true, then changes in substrate elevation might be sediment export in one place, bioerosion in another and wholesale loss of rugosity in all. I assume that the substrate type and sediment thickness was not consistently noted in older surveys. Given the points above, this could be an important driver of how quickly substrate elevation might change in one place versus another. The wholesale loss of architecturally complex acroporids and the subsequent reduction of these to pavement could be construed as "degradation of framework-building corals" as could bioerosion. Which was the main agent in each case?

---

## Author Comment (AC2) · 28 Nov 2016

We appreciate the detailed comments provided by Reviewer #2, and note that the reviewer indicated that he/she had a difficult time comprehending our methods and results. Many of the review comments indicate the reviewer may be unfamiliar with the technical aspects and terminology associated with the type of analyses we performed in our paper. We agree that there are areas that can be explained in more detail, and we also discovered a few typos that occurred during upload and conversion of our manuscript file that may have also caused some confusion. We greatly appreciate that this review allows us the opportunity to clarify and correct any misunderstandings or confusion, and to expand our explanations and discussions so that our presentation can better address a broader audience of readers. However, we, respectfully, disagree with the reviewer's general conclusion regarding the scientific quality of our paper. In-

dividual comments are addressed below (reviewer comments indicated by 'R2', author response indicated by 'AR').

R2: General comments: Both the scientific quality of the paper and its presentation quality are generally insuffi- cient. The scope of the paper, as it is formulated pages 2 and 3, i.e. "measuring changes in seafloor elevation to assess and predict the impact of reef degradation on the vulnera- bility of coastal communities to sea-related hazards" is confusing.

AR: The statement to which the reviewer refers is located on lines 29-31 of page 2 and reads:

"Therefore, measures of total system change in seafloor elevation and volume are required to accurately assess and predict the impact of reef degradation on the vulner- ability of coastal communities to hazards caused by storms, waves, sea level rise and erosion."

This is not a statement of the scope of our paper, but rather a statement regarding the need for the type of comprehensive elevation change analysis we performed. This sentence concludes a paragraph that summarizes the numerous types of studies that have been performed to look at individual accretion and erosion processes, states that no prior studies have accounted for the net result of all of these processes combined, and points out that vertical accretion and erosion is a function of total mass balance. The point we are making in the referenced sentence is that accurate predictions of coastal hazards depend on accurate measurement of changes in seafloor elevation (from a modeling standpoint). While many studies have examined individual processes that contribute to accretion and erosion in coral reef ecosystems, none (of which we know) have provided a measure of total system change due to combined accretion and erosion processes.

The scope of our paper is clearly stated in the abstract on page 1, lines 11-13 and lines 18-19 as:

"Here, we provide a comprehensive assessment of the combined effect of all of the processes affecting seafloor accretion and erosion by measuring changes in seafloor elevation and volume for 5 coral reef ecosystems in the Atlantic, Pacific and Caribbean over the last several decades."

"We show that regional-scale loss of seafloor elevation and volume has accelerated the rate of relative sea level rise in these regions."

The scope of our paper is then stated again, immediately after the sentence referenced by the reviewer, on page 2, lines 32-33 as:

"Here, we quantify the combined effect of all constructive and destructive processes on modern coral reef ecosystems by measuring regional-scale changes in seafloor elevation."

We will rewrite the sentence on lines 32-33 to read "The aim of our study is to quantify the combined effect of all constructive processes that cause accretion (or increases in seafloor elevation) and destructive processes that cause erosion (or decreases in seafloor elevation) on modern coral reef ecosystems by measuring regional-scale changes in seafloor elevation" to distinguish the scope of our study from the statement of need for this type of work.

R2: The title of the paper itself is also unclear.

AR: We chose this title because it summarizes the major finding of our work, namely that seafloor elevation is decreasing (rather than increasing) while sea level is rising in the coral reef ecosystems we studied (thus, we use the term divergence); and the combination of seafloor elevation loss and sea level rise has accelerated the relative increase in water depth at these locations. The title of the paper is derived from our concluding statement on page 16, lines19-21 that states:

"The divergence between rising sea level and declining seafloor elevation has already increased the risk to coastlines in these regions from long-term, persistent oceanographic pressures and periodic events such as storms."

We prefer to re-write the title to read: "Divergence of seafloor elevation and sea level rise in coral reef ecosystems" to more accurately limit our conclusions to the sites we studied based on comments from Reviewer 2 regarding page 14, lines 26-28.

R2: The authors announce that they address the coral reef issue and then they provide results on various habitats, including non-coralline habitats and even deep water offshore habitats (e.g., page 13, lines 3-5).

AR: Correction, we clearly state that our study sites encompass coral reef ecosystems, not only coral reefs (e.g. page 1 lines 12-13, page 2, line 32). We recognize that the definition of a coral reef and the term coral reef ecosystem have long been debated by coral reef ecologists and geologists. However, it is generally accepted that coral reef ecosystems include many non-coral dominated habitat/substrate types such as seagrass, hard bottom, sand bottom, macroalgal-dominated communities, as well as coral-dominated substrate including framework building reef structure. We chose to perform our analyses at the ecosystem scale because accretion in non-coral dominated habitats, as well as off shore habitats (and beaches which are excluded from our study) is supported by sand/sediment production from the breakdown of carbonate produced by corals and other calcifying reef ecosystem organisms. It is also well known that coral reefs have been degrading rapidly over the past few decades. However, little is known as to the effect of reef ecosystem degradation on accretion/erosion of non-coral dominated habitats within coral reef ecosystems. Our results suggest that the balance of erosion versus accretion has tipped enough that carbonate sediment production in these coral reef ecosystems is no longer sufficient to support accretion of adjacent habitats as indicated by the broad scale loss of seafloor elevation (erosion) that we observe across all habitat types. We will add a statement to our revised paper clarifying what we define as a coral reef ecosystem for the purpose of our study.

R2: Authors are unclear on what they measure, and on my view they fail in generating

robust data (as both the data and the methods used lack accuracy. Thereafter, the presentation of the results and their interpretation are confusing, as various types of processes are invoked to explain changes, with no specific process being robustly studied (e.g. page 13, lines 30-34).

AR: We understand that the methods used in our study to collect, process, validate and analyze the data are complicated. However, the general concept of the method is relatively simple, based on calculating the difference between elevation data points at the same location but from different time periods. We have created a flow diagram that depicts the core processing steps (see Figure 1 in our response to Reviewer 1), and we will include this figure in our revised manuscript to help non-experts better understand the process. We will also move Supplementary Tables 1, 2 and 3 (currently in the supplementary material section) to the main paper. These tables provide details regarding the data sources, data conversions and study periods for each study site to help further clarify our methods.

We measured the differences between historical and modern seafloor elevation using data sets from the time periods indicated for each study site (see Table S1 in the supplementary materials). This method allowed us to comprehensively measure the net change in seafloor elevation due to all of the constructive processes that cause accretion (or increases in seafloor elevation) and the destructive processes that cause erosion (or decreases in seafloor elevation), which was the scope of our study, the results of which represent a major finding on their own. It was NOT the scope of our study to robustly study the individual processes contributing to the changes we observed. However, we discuss the various accretion and erosion processes (and their rates as measured in previous studies) that are known to contribute to seafloor elevation change. We then place our results in context with published rates of accretion and erosion from these previous studies as is standard practice for rigorous discussion and interpretation of new results.

Although the general concept of our work is straightforward, a number of very rigorous analyses were performed to test the validity of comparing historical and modern data sets; very conservative methods were used to calculate error associated with the methods; and the effect of that error on our results was quantified and reported. We provide a detailed discussion of these analyses in our Response to Reviewer 1. We believe that our very rigorous analysis of the data and conservative computation and reporting of error has resulted in a robust data set and quantitatively significant results.

R2: For example, the first paragraph of the Discussion Section clearly illustrates the wide (and unprecise) area covered by the paper (see page 14, lines 5-15).

AR: We do not discuss the area covered by our study sites in this paragraph. However, our study sites are very clearly defined on the maps in figures 1-3 of the main paper, and the exact size of each study area as well as each habitat area within each study site is provided in Table 2 (in manuscript). The paragraph on page 14, lines 5-15 discusses the large number of processes that cause seafloor elevation and volume changes and the very general time frames over which they occur. We are, therefore, confused by this comment.

R2: Both the concepts ("change in seafloor elevation and volume" – in fact, it seems that the authors address "changes in shallow waters depth") and the method used (method "traditionally used to monitor seafloor changes", "use of historical bathymetric data from the 1930's to 1980's and LiDAR DEMs from 1990's to 2000's) –which are presented firstly in the introduction of the paper (pages 2-3) and then in the Methods Section (page 4) – are questionable and not accurate.

AR: We used methods practiced and tested by coastal engineers (including the U.S. Army Corps of Engineers) that have long been used to quantify changes in seafloor elevation over time (page 3, lines 2-5). We encourage the reviewer to read the references provided in this section to better understand the rigor with which these analyses are performed. The method is as accurate as the data sets, we provide a detailed error analysis of the data in section 2.5 of the manuscript, and account for that error in

our results. We have also provided further discussion in our response to Reviewer 1. With respect to the reviewer's concern regarding "changes in shallow water depth", we converted the sounding data (water depth data) to seafloor elevation using the NOAA National Ocean Service VDatum version 3.6 vertical transformation tool (see page 4, lines 15-18), which is standard practice for this field of study. This allowed us to compare historical elevation data with modern elevation data. Additionally, we used the locations of the historical soundings to extract modern elevation values from the seamless LiDAR digital elevation models which is more accurate than determining elevation change from two interpolated elevation surfaces (see page 5, lines 1-5).

R2: Concerning data and methods - How can bathymetric data from the 1930's to 1980's (constituting a single coherent period reflecting low anthropogenic impact?) be con- sidered as a starting point (or reference) to "measure changes in seafloor elevation" and then be compared with data from the 1990's to the 2000's (= period reflecting high anthropogenic impact?).

AR: Each of our five study sites was analyzed independently over a specific time period using the oldest reliable data that we could find for each site as well as the most recent bathymetric data available for that site. We did not measure change over a continuous historic range of 1930's to 80's and modern range of 1990's to 2000's. For example, the Upper Florida Keys historic data set was from 1934 and 35, and the modern data were from 2002. The dates for the individual data sets are available in Table S1 of the Supplementary Materials. The aim of the study was to look at change in seafloor elevation over several decades. Within the time frame of the study, population approximately doubled at each of the populated study sites (with the exception of uninhabited Buck Island). We clearly state that we use population as a first order approximation of relative anthropogenic impact. We make no claims of comprehensively analyzing all anthropogenic impact factors (see page 3, lines 16-21).

R2: This raises several key questions. Firstly, how can the "mag- nitude of erosion" (page 3, line 9) be measured using such an approach that poses serious questions

relating to the scientific quality of both the datasets and the method used. In other words, how can historical bathymetric data be compared with LiDAR data? The former (bathymetric data from the 1930's and next decades) do not have the required resolution for comparative measurements to be undertaken with LiDAR data. The low resolution of historical bathymetric data may generate significant errors in the results generated. Incidentally and curiously, no clear and complete information is provided by the authors on the resolution of the various datasets used in this study at the various study sites.

AR: Thank you for pointing out that we did, inadvertently, omit the vertical resolution information for our data sets. This information is available in the published metadata for each data set as referenced in Table S1 of the Supplementary Material. We will include that information in the revised manuscript and we discuss it below.

During these surveys several methods were used to collect soundings by USC&GS/NOS including lead line (0.15 m resolution), graduated sounding pole (0.15 m resolution) and fathometer (0.3 m resolution), and were noted in each NOAA hydrorgraphic data sheet (H-sheet) descriptive report. LiDAR data resolution is also reported in the metadata for each data set and ranged from 0.135 to 0.15 m. Therefore, the vertical resolution of lead line and sounding pole methods was similar to LiDAR methods. In general, for the 1930's surveys, sounding poles were often used in depths approximately less than 3 m and replaced with lead lines at greater depths. Descriptive reports from these data sets indicate that often the seafloor was visible due to high water clarity during pole and lead line data collection (which improves accuracy of the measurement). The most common error likely to occur during use of lead lines or sounding poles was overestimation of water depth due to angling of the line or pole as currents move the boat past the point of measurement. Overestimation of historical water depth would erroneously decrease elevation losses calculated using our methods. Therefore, it is more likely that our erosion estimates are underestimated rather than overestimated due to error associated with lead line and poling techniques. Additionally, we performed an analysis of differences between historical and modern elevation for areas of pavement that showed little change over the time periods (see response to Reviewer 1 for a detailed discussion). These results showed that average differences between modern and historical data sets for locations where little change has occurred range from 3 to 6 cm, providing evidence of the validity of comparing these data sets. Also note that we performed an independent, and more rigorous, error analysis of original repeat point measurements for the historical data sets, and used these error values (RMSESounding, Table 4) in our calculation (Equation 1, in manuscript) of total vertical error (RMSETotal, Table 4 in manuscript).

R2: Authors first indicate a 1 to 4Âa ÌĘm horizontal spatial reso- lution (this is a low resolution that do not allow the calculation of changes in the reef level) and then indicate a 11-12Âa ÌĘcm vertical resolution (which is questionable given the data used).

AR: The 1 to 4 m horizontal resolution applies to the LiDAR data sets only. We calculated horizontal error for the oldest historical data using published values for the methods (pages 10 and 11, section 2.5.2) of 4.8 m. To estimate the effect of horizontal error on our seafloor volume results, we performed a horizontal shift analysis. For this analysis, we doubled our calculated horizontal error to 10 m, then shifted the historical data set relative to the modern data set by 10 m in each of the four cardinal directions (N, S, E, W) and recalculated volume change for each scenario. These results indicate that horizontal error of up to 10 m and the resulting offsets in sounding points affects our volume calculations by 10% to 21% (depending on density of data points) and does not change the outcome or conclusion of our study. These results are consistent with reports that, over large areas (such as in our study), random errors largely cancel-out relative to change calculations derived from two surfaces (Byrnes et al. 2002). Additionally, this analysis provides further evidence that seafloor elevation loss is occurring at a very broad-scale across all habitat types.

With respect to vertical resolution, the reviewer's statement is incorrect. Nowhere in the paper do we claim vertical resolution of 11-12 cm. We believe the reviewer may be

making reference to the VDatum transformation error that we report for each data set that is only one parameter used to calculate total error (see page 8, lines 30-32 and Equation 1 in manuscript). We state the maximumum cumulative uncertainty for operational VDatum regions of South Florida and the Virgin Islands are 9.6 and 11.8 cm, respectively as reported by the NOAA National Ocean Service (page 8, line 31-32). We also report the VDatum error that was calculated specifically for each individual study site under the RMSEVDatum column in Table 4 of the manuscript that ranges from 8.1 to 11.4 cm (note no VDatum error was reported for Maui because no VDatum transformation was required). Total vertical error for each study site was calculated using equation 1, reported in Table 4 under the column heading RMSETotal, and ranged from 20 to 37 cm (average of 29 cm for all sites). As discussed on page 9, lines 30-31 and page 10, lines 1-6 (and in our response to Reviewer 1), we used a very conservative approach in our consideration of vertical error by multiplying our RMSETotal by a factor of 1.65 to encompass 90% of the variance in our data. This approach generated a more conservative RMSE of 0.48 m that we rounded up to 0.5 m; and we used this value to set minimum and maximum bounds in our volume calculations (Table 1 in manuscript). The minimum volume change values that we report in Table 1 were calculated by only including elevation changes that exceeded the range of -0.5 to +0.5 m to provide a very conservative estimate of volume change. These very conservative minimum volume change values also support our conclusions of net seafloor erosion at all study sites (Table 1 in manuscript).

R2: A second methodological problem is raised by the way anthropogenic impacts are considered in this study. (1) How can the "anthropogenic impact" only be measured by population numbers? In the present case (changes in water depth), it mostly depends on coastal and maritime human practices (sustainable/not sustainable). Major human activities, such as dredging in the substratum (should it be coralline or not) and extract- ing aggregate in particular, which may have occurred over the study period at some study sites and may have changed water depth, are not considered at all by the au- thors, which introduces a serious bias in the "elevation changes measured".

Some parts of the paper, such as "However, greatest mean elevation losses occurred in coral-dominated habitats and near the central coastline where harbour and shipping channels existÂa ÌĘÂz ÌĞ (page 13, lines 22-23) clearly indicate that not taking into account these human activities is problematic when assessing changes in shallow water depth.

AR: This claim is incorrect. In fact, on page 14, lines 8-18 (first paragraph of our Discussion), we specifically state that we DID include changes due to episodic events including, for example, dredging and infilling of channels and coastal harbors. This paragraph is quoted below:

"Our results include elevation and volume changes caused by chronic erosion processes that occur slowly over time frames of months to decades such as changes in carbonate production rates, bioerosion, chemical erosion from carbonate dissolution, degradation of large framework building coral colonies, and physical movement of reef sediments due to persistent oceanographic conditions such as waves and currents. Our results also include changes caused by episodic events that occur over very short time frames of minutes to days, and often cause large changes in elevation. Examples include dredging and infilling of channels and coastal harbors, deposition of terrigenous materials from landslides and run-off, slumping and relocation of seafloor materials at steeply sloping locations, storm erosion and deposits. We included large elevation-change 15 data in our calculations likely caused from these episodic events because such changes affect process modeling for hazards analysis and alter habitat distribution. We note that much reef degradation contributing to elevation change likely occurred after 1970 (Gardner et al., 2003; Bruno and Selig, 2007; 2014). Therefore, data sets containing pre-1970's data (Table S1) could be biased toward lower annual elevation and volume-change rates."

Additionally, we state both in the Abstract (page 1, lines 11-12) and in the Introduction on page 2, lines 32-22 that we provide a comprehensive assessment of the combined effect of all processes affecting seafloor accretion and erosion (constructive and destructive) on modern coral reef ecosystems. We provide a list of examples of all of the processes included in our comprehensive assessment on page 2, lines 24-28 that includes direct human alterations to the seafloor:

"However, no prior studies provide a comprehensive assessment of total seafloor elevation and volume change due to the combined effect of all of the processes affecting seafloor accretion and erosion (i.e., including physical erosion; redistribution, import or export of seafloor sediments; compaction; direct human alterations to the seafloor, carbonate production, bioerosion, chemical erosion)."

We include a figure in this response (see Figure 1 in this comment to Reviewer 2) that provides a very clear example of one of the human alterations included in our data set. Figure 1 (in this comment) is derived from Figure 2a in the manuscript and clearly shows an accretion area surrounding the airport in Charlotte Amalie, St. Thomas, USVI that resulted from infilling during construction of the airport. Again, the advantage of our seafloor elevation change method is that it does include all of the processes that cause accretion and erosion in a system.

R2: Generally, this paper mainly appears as a "technical" paper that describes the GIS pro- cedure applied to calculate changes in elevation, without addressing in an adequate way the conceptual, and the data and methods aspects raised. It seems that authors do not have the required background to address the complex scientific question that they have chosen to address. The technical procedure described on pages 4-6 is in- comprehensible to me. Despite the fact that I failed in understanding this procedure, my feeling is that the method is not robust due to poor conceptual, and data and method, bases.

AR: We feel that the reviewer has discounted the rigor and significance of our work due, perhaps, to a lack of expertise in the methods applied in our study. We hope that our responses have clarified our procedures and the conservative nature with which we have calculated and considered error in our results and interpretation. No previous

work (of which we know) has performed such a comprehensive analysis of the net result of all erosion and accretion processes affecting these coral reef ecosystems at the regional scale. As researchers in a science agency program whose mission includes assessing and predicting coastal hazards from natural and anthropogenic impacts to coastal ecosystems (including coral reefs), we are well versed in the concepts and complex scientific questions in our field of study that we have chosen to address.

Our findings showing that the magnitude of regional scale seafloor erosion that has occurred in these systems has increased relative sea level rise causing water depths not expected to occur until 2100 are, in fact, very significant. We recognize that total erosion at the regional scale has likely been underestimated because no prior studies fully account for all processes causing elevation change in coral reef ecosystems; and we understand that our results may cause controversial feelings. However, we feel that we have proven the validity of our results and use of historical and modern data sets for our analyses with our expanded error analysis and our use of a very conservative RMSE for data calculations. Our conclusions regarding loss of seafloor volume are based on actual measurements of elevation-change shown to be statistically significant in over 90% of the habitats we analyzed, and that account for all of the processes causing elevation loss in these regions. We have described our technical procedures in great detail because they are complex, this is the first application of these methods to coral reef regions, and we hope that other scientists are able to apply these methods in many other coral reef regions.

R2: In different sections of the paper (e.g., page 13, line 1), the results obtained are correlated to generalities, e.g. on coral reef degradation, which is questionable. Results should be correlated to local data on reef health, including observed changes in living coral coverage, but not to worldwide observations. The interpretation of the results generated is not satisfactory: for example, the au- thors mention hurricanes as key controls of changes in depth. This raises the question of "what is measured, either long-term changes related to climate change and sea level rise, or changes due to

low-frequency high-magnitude events"? Once again, this makes the paper confusing.

AR: We are somewhat confused by reviewer's comment and the reference to page 13, line 1 which only states the number of habitats in which mean elevation and volume losses occurred in the Upper and Lower Florida Keys. We do, in fact, cite local data on reef health and processes that support our observations. For example, on page 12, lines 32-22 and page 12, line 1, we state:

"Largest mean elevation losses occurred at shallow patch and aggregate reefs, coral-dominated and reef rubble habitats, consistent with documented declines in abundance of large framework-building corals over the past several decades (2014)." We note that part of this reference was missing due to a typo. The 2014 reference that should have been included here (and is in our reference list) is:

Jackson, J., Donovan, M., Cramer, K., and Lam, V. (Eds.): Status and trends of Caribbean coral reefs: 1970-2012, Global Coral Reef Monitoring Network, IUCN, Gland, Switzerland, 306 pp., 2014.

We will correct this in our revised version. The Jackson et al. study is a comprehensive analysis of coral cover throughout the Caribbean including data for change in coral cover from the 1970's to 2000's for both the Upper and Lower Florida Keys, for Buck Island from 1989-2011, and for St. Thomas from 1979-2010. We will include this reference in our discussion of the Buck Island and St. Thomas results. Additionally, as recommended by Reviewer 1, we will also discuss changes in live coral cover for each of our study sites in the text of our revised manuscript.

On page 13, lines 9-12, we note that a sub-region of the Upper Florida Keys showed a slight increase in elevation and that this location is near an area of the Middle Florida Keys that has been identified by Manzello et al. (2012) as a possible refuge from ocean acidification based on local data for that area. On page 13, lines 16 -19, we reference work by Lidz et al. (2007) and Shinn et al. (2003) that corroborates our observation of accretion on spur-and-groove habitat due to burial by sand and evidence for redistribution of reef materials by hurricanes that has caused erosion in some areas and deposition in other areas of the Florida reef tract. On page 15, lines 5-8, we compare our erosion rate in the Upper Florida Keys to chemical erosion rates determined for the Florida Keys by Muehllehner et al. (2016). All of these are examples of comparison of our results to local data on reef health and processes. We previously discussed that our methods and results account for both long-term changes over the time periods of our study as well as low-frequency, high-magnitude events (see page 14, lines 8-18).

We do include a discussion of our results in context with, what we believe, are seminal papers on comprehensive (large-scale) analyses of various accretion and erosion processes that are generally accepted by our scientific peers (e.g., Shinn et al., 1977; Buddemeier and Smith, 1988; Church et al., 2013; Perry et al., 2013; among many others, see Discussion Section 4). Given that we performed ecosystem-scale analyses in the Atlantic, Caribbean and Pacific regions, we felt it appropriate to place our results in context with broader observations across these regions.

R2: Specific comments: Page 2, lines 28 to 30 are incomprehensible: "measures of total system change in seafloor elevation and volume are required to accurately assess and predict the impact of reef degradation on the vulnerability of coastal communities to hazards caused by storms, waves, sea level rise and erosion".

AR: We will clarify this passage to explain that hydrodynamic and other numerical models used to assess and predict the impact of reef degradation on coastal hazard vulnerability require accurate seafloor elevation data as well as accurate seafloor elevation change data.

R2: Page2, lines 31-32: "we quantify the combined effect of all constructive and destructive processes on modern coral reef ecosystems by measuring regional-scale changes in seafloor elevation" is incomprehensible.

AR: All accretion and erosion in coastal systems causes changes in seafloor elevation. Accretion is a constructive process and erosion is a destructive process. We will rewrite

this sentence to clarify.

R2: Page 3: "we adapted an elevation-change analysis method that has traditionally been used to monitor seafloor changes"

AR: We are uncertain as to your question regarding this statement.

R2: Page 7, line 25 "sediment thickness of the Holocene reef deposit": what do the authors talk about? Vertical sedimentation? Vertical Holocene reef building? The results exposed page 8, lines 11 to 14 for the Lower Florida Keys case study are incomprehensible to me. I do not understand how the authors "used a moder reef age of 6000 years and a constant erosion rate" to "compute the time required to completely erode the remaining Holocene reef down to the Pleistocene layer".

AR: The Holocene epoch began approximately 12,000 to 11,500 years ago after the Pleistocene epoch. The terms 'recent' and 'modern' reef are often used to describe the reef deposit that accreted during the Holocene. We are referring to vertical sedimentation of the Holocene reef ecosystem in the Florida Keys for these sections. The geologic history of the Florida reef tract is very well known, and the coral reefs and sediments of the present ecosystem began accumulating approximately 6000 to 7000 years ago on top of Pleistocene bedrock as described in detail in Lidz et al. (2007) that we cite. Thus, we used 6000 years as the modern reef age for this calculation. The constant erosion rate to which we refer is the erosion rate that we calculated from our elevation change analysis of the Lower Florida Keys study site. We assumed no change in this rate over time, for this calculation. We will add a few sentences to Section 2.4 describing the geologic setting of the Florida Keys reef tract for clarification.

R2: Page 8, lines 29-30: I am surprised to read that vertical errors would be comprised between 9.6 and 11.8 cm respectively, given what I know on LiDAR data and the horizontal error (1 to 4 a Įm) applying to this study. More generally, I do not understand how vertical error estimation was conducted.

AR: The reviewer has misunderstood the terminology on lines 29-30. The vertical error to which the reviewer is referring is the maximum cumulative uncertainty caused by transforming data from one vertical datum to another using the VDatum software, and represents only one component of our vertical error analysis (see page 8, lines 29-30). Please see equation 1 on page 8, line 24. This type of equation is widely used for calculation of root mean square error (RMSE) in elevation change analyses that considers multiple sources of error.

Our total vertical error analysis included error terms for:

1) modern LiDAR data sets (RMSELiDAR). LiDAR uncertainty was determined by independent validation of airborne LiDAR measurements with in-water acoustic sounding measurements performed at the time that the LiDAR data was collected and reported in the metadata for these data sets.

2) historical data sets (RMSESounding) as determined from our analysis of repeat measurements that were performed by the original surveyors at the time of data collection, and

3) uncertainty due to transforming data to a common vertical datum as calculated using VDatum (RMSEVDatum) for each individual data set.

These uncertainty values specific to each data set are reported in Table 4 (in manuscript), and were included in our calculations of RMSE (see page 8, equation 1 for RMSETotal). Our average RMSETotal (Table 4 in manuscript) for all study sites was 0.29 m. We considered the RMSETotal values from each study site as proxies for the standard deviations. In a normal distribution, data within plus or minus one standard deviation of the mean encompasses approximately 68% of the variability; and data within plus or minus 2 standard deviations of the mean encompasses approximately 95% of the variability. We note a typo on page 9, line 28 in the notation of this statement that may have caused some confusion, and we will correct this in the revised manuscript. We used the Normal Inverse Cumulative Distribution Function to

compute that 90% of our elevation change values would occur within plus or minus 1.65 standard deviation of the mean. We chose to multiply our RMSETotal by a factor of 1.65 to encompass 90% of the variance in our data and generate a more conservative RMSE of 0.48 m that we rounded up to 0.5 m; and we used this value to set minimum and maximum bounds in our volume calculations (Table 1 in manuscript). We will rewrite this passage in the revised manuscript to include more detail and clarify these calculations.

R2: Page 12, lines 13 to 27– We understand that most study are not dominated by coral reefs, which means that this paper does not in fact address the pretended issue of reef response to changing environmental conditions. This suggests that the choice of study sites is not totally coherent with the objectives of the paper.

AR: Again, we remind the reviewer that our study sites encompass coral reef ecosystems that include, but are not limited to coral reefs as discussed previously in our comment to the reviewer.

R2: Page 12, lines 25 to 27: the conclusions drawn by the authors from the study of Buck Island correlates volume loss to sediment export. Both the results (volume loss) and the interpretation of the results (sediment export) are unclear to the reader.

AR: The point that we are making here is that elevation and volume loss occurred in all of the regions, but to a lesser extent at the Buck Island study site. When materials are lost from a region (i.e. exported), that loss causes a decrease in volume of materials within that region. Therefore, this suggests that less export of materials is occurring from the Buck Island study site. We will rewrite this sentence to clarify.

R2: Page 14, lines 20-25: how can the authors convert "changes in elevation" into a "num- ber of years of Holocene reef accretion"? This is not robust as coral reefs grow and erode over a given period, as a result of the complex imbricated processes driving both reef construction (i.e. construction) and sediment production (i.e. erosion allowing car- bonate production).

AR: The point of this exercise was to further demonstrate the significance of these losses. We will rewrite this paragraph to clarify and include more detail on this calculation. Accretion of a reef ecosystem over time occurs when the balance of accumulation of reef materials and sediments exceeds erosion and loss of the eroded material from the system. The fact that we observed mean seafloor elevation loss across the whole coral reef ecosystem scales that we studied indicates that more materials are being eroded and exported from these systems than are accumulating. Our annual mean seafloor elevation losses on page 14, line 20 are calculated by dividing the total mean elevation change in meters reported in Table 1, column 3 (in manuscript) by the number of years in each time period, for each study site:

UFK = -0.1 m / 68 years = -1.5 mm/year

LFK = -0.3 m / 66 years = -4.5 mm/year

STT = -0.3 m / 48 years = -6.3 mm/year

BI = -0.09 m / 33 years = -2.7 mm/year

Maui = -0.8 m / 38 years = -21.0 mm/year

We then divided our total mean elevation losses losses (Table 1, column 3 in manuscript) by published rates of average Holocene reef accretion rates for these regions (mm per year) to estimate how many years of reef accretion was lost due to erosion:

UFK = -0.1 m / 2.6 mm/yr = 38 years

LFK = -0.3 m / 2.6 mm/yr = 115 years

STT = -0.3 m / 2.6 mm/yr = 115 years

BI = -0.09 m / 2.6 mm/yr = 35 years

Maui = -0.8 m / 10 mm/yr = 80 years

R2: Bottom of page 15-top of page 16: I do not understand how the results obtained by the authors can be compared to the results of previous studies conducted by C. Perry to attribute observed changes to specific drivers/processes.

AR: Perry's 2015 paper is one of the most recent studies showing that while remote coral reefs that are largely isolated from human influence experience severe coral mortality from climate driven impacts like bleaching, most of these reefs recover very rapidly and continue to produce enough carbonate to keep up with present and future sea level rise. The C. Perry study was based on assessment of 28 reefs across the Chagos Archipelago reef system in the Indian Ocean. The point we are making here is that, although Maui is remote, it is not isolated from human influence; and our results showing large erosion rates suggest that these reefs systems have not recovered well from degradation and are not producing enough carbonate to keep up with rising sea level. We will expand this discussion to clarify.

R2: Page 15, lines 19-21: the estimation that "the total reef volume could completely erode down to Pleistocene-bedrock-surface in approximately 1250 years" is not well-founded.

AR: We have clarified our concept and procedure for this analysis in our response to the reviewer's previous comment regarding page 7, line 25. We will expand the discussion as previously indicated.

R2: Page 15, line 33: ". . . reef systems. . . lack human impacts" is not correct in terms of style.

AR: The terminology used in this sentence is from Perry's 2015 paper, and we are not certain as to what the reviewer is recommending here. Please clarify.

R2: Page 15, lines 23-35: key references on reef islands future are not cited by the authors. See in particular the recent studies by Kench et al.

AR: Good suggestion. We assume the reviewer is referring to the latest 2015 papers

listed below. We will include these in the discussion of our revised manuscript.

McLean, R., & Kench, P. (2015). Destruction or persistence of coral atoll islands in the face of 20th and 21st century sea-level rise? Wiley Interdisciplinary Reviews – Climate Change, 6 (5), 445-463. 10.1002/wcc.350

Kench, P. S., Thompson, D., Ford, M. R., Ogawa, H., & McLean, R. F. (2015). Coral islands defy sea-level rise over the past century: Records from a central Pacific atoll. Geology, 43 (6), 515-518. 10.1130/G36555.1

Kench, P. S., Owen, S. D., & Ford, M. R. (2014). Evidence for coral island formation during rising sea level in the central Pacific Ocean. Geophysical Research Letters, 41 (3), 820-827. 10.1002/2013GL059000

R2: Page 16, lines 3-5: an assumption like "Modern carbonate production rates are an or- der of magnitude lower than Holocene averages (Perry et al., 2013), and are estimated to decrease by as much as 60% by mid-century (Langdon and Atkinson, 2005)" is far too general.

AR: These statements are not assumptions, rather they are derived from the field and experimental results of the cited studies. We will add additional references to this section that support these statements. We do not use this data to quantify future responses of reefs, but rather to point out that carbonate production rates are projected to continue to decrease while bioerosion and chemical erosion are projected to increase in the future. These combined impacts are, in fact, likely to accelerate reef erosion rates.

R2: Tables 2 and 3 – The substrate categories included in these table are not presented and justified in the study. We additionally have not idea of the depth at which these habitats are situated.

AR: This statement is incorrect. The habitat/substrate maps are discussed and referenced on page 6, lines 18-27:

"We obtained benthic-habitat-map shapefiles (Florida Fish and Wildlife Conservation

Commission, 2015) for the Upper and Lower Florida Keys study sites from Florida Fish and Wildlife Conservation Commission (FWC). The Unified Florida Reef Tract (UFRT) map Version 1.2 is comprised of 5 class levels from 0 to 4. We used class level 2 for our study because the level of detail was consistent with benthic habitat data available at our other study sites. We obtained benthic-habitat-map shapefiles for the USVI and Maui from NOAA (Rohmann, 2001b, a; Battista and Christensen, 2007). We delineated USVI habitats using the 'type' descriptor in the shapefile's attribute table. We delineated Maui benthic habitats using the 'D_STRUCT' class in the attribute table. We retitled the habitat class named 'Rock/boulder' in the 'D_STRUCT' class that corresponded to the descriptor from the 'M_STRUCT' class named 'Coral Reef and Hard Bottom' to clarify that particular substrate type is a coral-dominated habitat. All classes were chosen to provide a common level of benthic habitat detail across study sites. Once the habitat classes were chosen, we exported them as individual shapefiles with ArcMap."

Both historical and LiDAR seafloor elevation data and latitude/longitude are provided for every single data point within these habitats for each study site in Tables S4 – S8 of the Supplementary materials. We can add mean water depths for each habitat type to Table 2.

R2: The maps provided page 30 indicate a complex spatial distribution of gains and losses, which is not described in the paper. They also show that shallow habitats were not totally covered, suggesting that gains may have occurred in non-covered areas that may compensate observed losses in study areas. This is all the more to be considered that the results obtained are contrasting (e.g. between the Central Sub-Region and the Lower Sub-region of the Florida Keys).

AR: We discuss the spatial distribution of gains and losses by habitat type in Section 3.2, and briefly discuss the large ranges of elevation change and the effect on standard deviations on page 11, lines 24 -28. We will expand the discussion of results in section 3.1 to more directly recognize the complexity of the spatial distribution of the data.

There are gaps in the data coverage for the Florida Keys and Buck Island data sets and we will note these in our revised manuscript. Our analysis was limited by the areal extent of available data sets. However, there is overwhelming evidence in the literature (as well as anecdotal evidence from field observations) that the trends we see in the areas covered by our data sets are consistent throughout the surrounding areas outside of the boundaries of our study sites. We feel that our study areas in these locations are at a large enough scale to be representative of the broader region. We do not argue that there are some locations within our study sites that show accretion. For example, offshore, downslope areas where sediment is infilling spur and groove formations; areas shoreward of patch reefs where sediment from degraded coral reefs has been redistributed to deeper water habitat behind the reef; and, in the case of the Lower Sub-region of the Upper Florida Keys, a small area of increased elevation primarily associated with seagrass beds (that we discuss on page 13, lines 10-12). However, the amount of total volume loss at these study sites is substantial. To put this in perspective, our results indicate that seafloor volume has decreased in the Upper Florida Keys study site by 14.6 (lower bound) to 37.9 (upper bound) million cubic meters. One million cubic meters is approximately the same volume as the Empire State Building. So, the amount of seafloor volume lost in this area is equivalent to approximately 14 to 37 Empire State Building's worth of material volume. There is no evidence from the numerous other geological, ecological, or geophysical studies throughout the Florida Keys for redistribution and deposition of this amount of seafloor material in the shallow habitats that lie between the outer reef tract and the shoreline of the Florida Keys.

R2: Concerning the Florida Keys, curiously nothing is said in this paper about the dominant modes of planform change and about Keys' landward migration. This suggests that the general context that allows interpreting correctly the results is not presented and considered when analysing the results.

As discussed above (re: Depth to Pleistocene Bedrock analysis), we will include a brief summary of Holocene reef formation and platform change in the Florida Keys to

help develop the context for the analysis described in section 2.4 (Lower Florida Keys – volume to Pleistocene Bedrock). The time period of our elevation change analysis in the Florida Keys focuses on changes over the past 64 to 68 years. We feel that a broader discussion of platform evolution and change that occurs over much longer time scales than the processes accounting for much of the change we observe (discussed in manuscript) is well beyond the scope of our paper. Additionally, all of our data were corrected for sea level rise that accounts for subsidence, etc.

R2: The results obtained in Saint Thomas, as shown by Map a page 31 mainly exhibit stability to limited elevation loss, if we consider grey and yellow areas. When we see this map, we are not convinced that elevation losses prevail, especially if we consider the error range. The same observation can be made when considering map c page 31 showing the situation of Buck Island (blue and yellow area are extensive).

AR: Unfortunately, we were required to change the color scheme of these figures to meet required color standards for the journal publication, and we agree that these figures don't do the data justice. We believe the reviewer is referring to the light yellow and light blue areas of elevation decrease and increase, respectively (the gray areas represent land). We will try to adjust the color schemes (within the color guidelines) in the revised manuscript so that the blue colors don't overwhelm the light yellows. I've included the figures from our unrevised submission (Figures 2 - 4 in this comment) that were color coded to better distinguish between elevation decreases and increases. However, we point to the actual data rather than the illustrations as proof that elevation losses exceed gains and of the significance of the results. It is important to note, that the light yellow and light blue boxes for the plus or minus 0.5 m elevation change data represent 1.65 x RMSETotal. Our RMSETotal was 0.29 m. Therefore, the light blue and yellow color areas include statistically significant data (greater than RMSETotal). We chose a plus or minus 0.5 m range for these figure categories because they represent the amount of data that we did NOT include in our minimum bound (conservative) volume calculations (see Table 1 in manuscript and page 6, lines 8-11 and page 9, lines

30-32 to page 10, lines 1-6 of vertical error analysis section). Even after we remove all of the data in light yellow and light blue (plus or minus 0.5 m) from our calculations, our results still show net volume loss/erosion of these study sites (see Table 1, column 9). The mean losses in 97% of the 59 habitats we analyzed for all study sites were greater than our RMSETotal of 0.29 m, and 77% of the habitats showed mean losses greater than 1.65 x RMSETotal of 0.5 m (see page bottom of page 11 and top of page 12, lines 30 – 6). One of 17 habitats in St. Thomas showed mean loss less than 0.29 m (reef rubble), and one of 11 habitats in Buck Island showed mean loss less than 0.29 m (seagrass), see Table 2, column 9 (Mean loss).
* * *
[Figure]

**Fig. 1.** Example of human alteration to seafloor included in data set. a) area of accretion near near airport, b) inset of accretion area, c) aerial imagery of airport

[Figure]

**Fig. 2.** Figure 1 from manuscript with original color scheme

[Figure]

**Fig. 3.** Figure 2 from manuscript with original color scheme

[Figure]

[Figure]

**Fig. 4.** Figure 3 from manuscript with original color scheme

---

## Author Comment (AC3) · 28 Nov 2016

Please note that a typographical error was made on page C7, lines 11-12 of the authors' response to Reviewer 1 comments. The sentence that states "Results from this analysis are termed RMSESounding in Table 4 (manuscript), and range from 2 to 37 cm." should be corrected to state "Results from this analysis are termed RMSESounding in Table 4 (manuscript), and range from 13 to 32 cm."

---

## Referee Comment (RC4) · Anonymous Referee #4 · 11 Dec 2016

The paper by Yates et al. analyses bathymetric data to quantify seafloor elevation changes in coral reef regions. As highlighted by reviewers 1 and 3, the dataset presented in the manuscript is especially impressive (number of sites considered, extent of the area considered). The results will be useful for coastal geomorphologists and managers concerned with the sustainability of coral reefs environments and the related ecosystem services. Despite the recommendation of reviewer 2, I think that the paper should be published after major revisions, to ensure that the amount of data analyzed in this work receives the attention it deserves.

Three previous reviews have extensively discussed the paper: overall, reviewers (1) have concerns regarding the ability of the method to retrieve seafloor elevation changes at the required accuracy; (2) made comments on the form of the paper; (3) and the interpretation of the results. The authors have already provided responses to several

comments of the reviewers, and intend to implement corrections to their paper, which I think are reasonable. I would suggest that these major revisions are implemented, considering the following points:

- All reviewers agree that the paper should separate more clearly what is the overall approach (comparing bathymetric data) from the technical details of the GIS procedure used to produce this data. The authors have prepared a figure as part of their response to reviewer 1 to address this comment. However, I think that the figure remains too technical (e.g., use of the TIN surface wording), and I would support producing the detailed GIS procedure in an annex to the paper. Overall, I agree with previous reviews, who suggested that the authors should consider that their results may have a large impact beyond specialists of coastal bathymetric surveys, so that ideally, they should try to separate the main messages from the technical implementation details.

- While the comparison of historical bathymetric sounding with contemporary LiDAR is quite widespread in coastal geomorphology (as reminded by the authors, see AC2 – pages C6-C7), there is always the suspicion that the two techniques induce errors, as highlighted by reviewers 1 and 2. Such errors can arise because the techniques have not the same purpose and therefore don't necessarily capture the same proxies (e.g., highest seafloor elevation features for navigation applications vs average seafloor elevation feature for bathymetry data in support to coastal hydrographic modelling). The techniques also have different accuracy/precision (as discussed already), or because of time-sampling issues (as commented by reviewer 2). Overall, I think that the authors make a fair assessment of these errors: in the response of the authors to this comment of reviewer 2 (AC2 pages C8), information regarding the vertical resolution of the techniques is provided, while the precision issues are given in Table 4 in the original manuscript. To complete this assessment, I would suggest to provide more information on the planimetric resolution, and the vertical accuracy of the techniques in the core of the article. This includes details regarding the definition of a common reference, which incorporates sea-level rise constructions in a way, which is not completely clear to me

based on the original manuscript, page 4 lines 10 and following. Nevertheless, I am confident this does not affect the results of the authors, as the RMSE in vertical datum adjustment is probably much larger than the RMSE due to uncertainties in relative sea-level changes for the sites of interest (table 4).

- Regarding the interpretation of the results: besides the aspects discussed with reviewer 1 and 2, I think that reviewer 3 provides a very clear line for improving the discussion section, and I hope that the authors will build on it in a future version of the article.

- Finally, I think that a "conclusion" section is needed.

I hope these comments are useful.

―――――――――――――――――――

---

## Referee Comment (RC5) · D. Hubbard (Referee) · 13 Dec 2016

As I initially said, I think this is a good study - for exactly the same reasons the readers intended. I am satisfied with the revision suggestions but strongly encourage the authors to make sure that the reader understands that the main goal is to focus on the magnitude of change with the caveat that attributing mechanisms is much less well constrained. In this vein, I suggest that the authors take an opportunity to perhaps comment on where we have big gaps in our knowledge that hamper such attributions and why at least a semi-quantitative understanding of the relative relationships between carbonate addition and removal is important to our understanding of the changing dynamic between accretion/erosion and SL rise. The non-linear relationship between water depth across the reef and wave energy passing by has been elegantly demonstrated by many careful instrumental studies. This study addresses ways to quantify

the first half of that relationship and, therefore, has considerable value when we move on from coral loss to longer-term physical and social impacts - whether in Miami or some small atoll in the Indo-Pacific region.

---

## Author Comment (AC4) · 13 Dec 2016

AR: Many thanks to the reviewer for a very thoughtful and constructive review of our manuscript. We believe that we will be able to greatly strengthen our manuscript and broaden its impact by re-writing the methods (as all reviewers have recommended) and by providing more detailed discussion and clarification in the areas addressed by Reviewer 3. We have addressed individual comments below (reviewer comments indicated by 'R3', author responses indicated by 'AR').

R3: Overall, I am impressed by this paper and think that it is an interesting attempt to ele- vate monitoring to something more than "counting corals". However, I am concerned that the likely variability in sources of substrate change were probably much more dif- ferent from site to site than has been characterized. I could be wrong, but I

suspect that bioerosion is less of a factor than is represented here. . . and is more likely declin- ing at most Caribbean sites. While there is an effort to address site-to-site variability, I am not convinced that the relative roles of simple bioerosion, large-scale rugosity loss and export by storms have been adequately considered. I would like to see this paper appear in print, if only for the valuable data set. However, I am con- cerned that the ex- planations of the measured patterns is a bit oversimplified and relies too much on the mechanisms proposed. I, therefore, provide some over-arching thoughts below in the hope that the authors can perhaps think a bit more about other possible explanations for the patterns they observed. Accordingly, I make a few general observations below that will hopefully be useful.

AR: We appreciate the reviewer recognizing the value of our work from the monitoring standpoint. Out primary intent for this paper was not to focus on a detailed analysis of the processes causing elevation change. Our intent was to focus on introducing this type of monitoring approach to the coral reef scientific community as the first application of seafloor elevation change methods to whole coral reef ecosystems, to demonstrate the value of this approach, and to report the substantial regional-scale net change in seafloor elevation that has occurred over the past several decades and has gone undocumented until now. The concept of our work is based on the premise that eleva- tion change analysis measures the net result of all of the constructive and destructive processes on whole reef systems including those processes for which rates are known (e.g., carbonate production, biological erosion, chemical erosion, etc.), as well as those processes that haven't been well characterized (for example physical erosion and ex- port). While the method does not attribute change to cause, it does provide a measure of the net result of all impacts (natural or anthropogenic) to seafloor structure in these ecosystems.

We agree that our explanations of the sources of substrate change are very general. Our results indicate that erosion has been largely underestimated in these regions. Our discussion of mechanisms of substrate change (page 14, line 30 through page 15, line

15) was an exercise to demonstrate the magnitude of net erosion that we measured by elevation change (that accounts for all processes) relative to individual processes for which process rate estimates were available including bioerosion among others. Additionally, it was a demonstration of how these regional scale elevation change analyses might be used to help attribute change to cause by way of identifying and quantifying changes that have been unaccounted for by known process rates. We do see how that exercise can be mistaken as an attempt at a comprehensive process analysis. We can clarify our intent for this comparison both in the section that discusses processes on pages 14 and 15, and modify our statement in the abstract regarding the results of that comparison.

In our brief discussion of bioerosion (on page 15 beginning on line 1) we purposefully selected a maximum bioerosion rate from the literature for that discussion to show that, even assuming maximum rates, bioerosion alone cannot account for the elevation changes we observed. We believe the reviewer perceived that we attributed much of the erosion we observed to bioerosion because we applied a maximum bioerosion rate to our estimates in the exercise discussed in the above paragraph. In fact, we agree with the reviewer that bioerosion is likely less of a factor than physical erosion and transport. We attribute much of the volume loss we observed to evidence for physical erosion and export in a discussion on page 12, lines 25-28. We will clarify with additional references, bioerosion rates, and text that bioerosion rates are likely much less and, in some areas, likely to decrease where bioeroder communities are decreasing.

We had hoped to convey to the readers the value of knowing net whole system change in trying to account for and attribute that change to individual processes as well as identifying missing gaps in erosion/accretion budgets. As an analogy, it's easier to put the puzzle together (and to figure out which pieces are missing) if you know how the whole picture looks. Additionally, we recognize that predicting effects of coral reef ecosystem degradation on hazards to coastal communities in these regions (sea level

rise, storms, tsunamis, etc.) has been largely limited due to a lack of whole system seafloor change analyses such as we performed for this study. Our results quantify substantial seafloor elevation loss in these regions and are (on their own) a significant finding that suggests risks from coastal hazards may be underestimated in these coral reef regions. These data sets provide a foundation for improving numerical models of present and future coastal hazards in these regions. We feel that a rigorous discussion of processes for each study site would be best addressed in individual manuscripts for each study site, and that a detailed process analysis of each study site is beyond the scope of our paper. We do, however, feel that our results will encourage and help improve future studies that develop comprehensive erosion budgets and account for process rates. We appreciate the very thoughtful and constructive comments from that reviewer that will help us add text to better explain the intent of our study, the need for these types of whole system change analyses, and to clarify that the point of our general assessment of processes was to demonstrate the magnitude of that change relative to change caused by a few individual processes for which rates are available, rather than a comprehensive process analysis. We will also expand our discussion to recognize the limitations of our examples, and add additional references and discussion to support that discussion.

R3: Comments:

Like Reviewer 1, I am not well versed in the GIS and data transformation methods utilized in this study. However, I am familiar with the vagaries of older hydrographic surveys. On the latter front, I am willing to accept their characterizations of (the direction of?) change in substrate level as the differences between sites are probably sufficient to overcome any stated errors. However, in my experience, the notes on smooth sheets leave us with a need to make defensible assumptions about a) the reliability of substrate characterization (and its stability) and b) how processes that potentially influence elevation change might differ from site to site. I have limited my comments to the latter, based on areas in the manuscript where I have experience in either the

specific habitats or the processes that might contribute to the patterns described.

Before I start, I do have one comment on style. I am not qualified to comment on the statistics of the methods or the assumptions made in the GIS transformations and map algebra. Nevertheless, a more reader-friendly explanation on that front would make the paper more accessible to a broader audience. The paper in its present form is a wealth of information on methods for those inclined to apply them to other sites. However, those people are probably going to be less well informed on the evolution of carbonate substrates. Conversely, those with intimate understanding of carbonate cycling are going to be unable to tie their knowledge to the details of the methodology used here. I am in that latter group and would suggest that the minutiae of the transformations and GIS tools could be better placed in the Supplemental Materials.

AR: Thank you for pointing this out. We have learned through the review process that we need to rewrite the methods section so that a broader audience can more easily follow along. As indicated in our response to Reviewer 1, we have developed a flow diagram that we will also include that summarizes the methodological steps. We will also move the detailed GIS methods to a supplementary section. Again, our aim was to introduce this approach to the coral reef community, and we want them to be able to use it. We provided detailed methods to help others pursue this type of work. We have come to realize through our reviews that the cross-pollination of expertise in elevation change analysis and coral reef studies is new enough to both areas of expertise that we must be more rigorous in defining terminology and explaining procedures.

R3: The following ae my general thoughts based on elements of carbonate cycling that could lead to conclusions other than those drawn here. While I am willing to accept the numerical changes in substrate elevation, I am somewhat less comfortable with assumptions about the degree to which they are related to bioerosion and the ensuing removal of sediment.

Biorosion versus structural reorganization – In the discussions, there is an apparent

conflation of bioerosion and spatial heterogeneity. The paper by Alvarez-Fillip et al. (1990) that is cited to document the role of increased bioerosion focused on the loss in architectural complexity (aka rugosity) and its causes – not bioerosion. In the paper, they attributed the initial reduction in reef rugosity to the loss of acroporids and the second decline in rugosity to a loss of massive species following bleaching. It seems reasonable to assume that an increase in susceptible substrate could increase bioerosion. However, Alvarez-Fillip et al. focused on the loss of rugosity which, in the case of A. palmata, is more easily explained by physical toppling/breakage and incorporation of fragments into a broad, cemented pavement.

AR: The Alvarez-Filip et al. (1990) reference should have been listed earlier in the sentence to which the reviewer is referring. We thank the reviewer for catching this mistake. On page 1, line 30 through page 2, lines 1-3, the sentence reads:

"Local and global, natural and human-induced stressors have caused the loss of reef-building organisms and reef structure, a decrease in biodiversity, a transition to algal-dominated communities (Pandolfi et al., 2003), and an increase of bioerosion (Alvarez-Filip et al., 2009), placing coral reefs around the world in a state of rapid decline (Madin and Madin, 2015)."

The Alvarez-Filip reference should have been placed after the statement "…have caused the loss of reef-building organisms and reef structure…". The reference that was intended to follow the statement "…and an increase in bioerosion…" should have been Enochs et al. (2015), see reference list in manuscript. The Enochs reference discusses experimental and modeling results showing that ocean acidification increases bioerosion of coral by the boring sponge Pione lampa. We will correct this mistake and clarify in this sentence that we are referring specifically to some species of boring and endolithic bioeroders. We will also include the following references as other examples:

Wisshak M, Schönberg CHL, Form A, Freiwald A. 2013. E ects of ocean acidi cation and global warming on reef bioerosion – lessons from a clionaid sponge. Aquat Biol.

19(2):111–127. http://dx.doi.org/10.3354/ab00527

Wisshak M, Schönberg CHL, Form A, Freiwald A. 2012. Ocean acidi cation accelerates reef bioerosion. PLoS ONE. 7:e45124. http://dx.doi.org/10.1371/journal.pone.0045124

Tribollet A, Godinot C, Atkinson M, Langdon C. 2009. E ects of elevated pCO2 on dissolution of coral carbonates by microbial euendoliths. Global Biogeochem Cycles. 23(3):1–7. http:// dx.doi.org/10.1029/2008GB003286

Reyes-Nivia C, Diaz-Pulido G, Kline D, Hoegh-Guldberg O, Dove S. 2013. Ocean acidification and warming scenarios increase microbioerosion of coral skeletons. Glob Change Biol. 19(6):1919–1929. http://dx.doi.org/10.1111/gcb.12158

DeCarlo T., Cohen, A., Barkley, H., Cobban, Q., Young, C., Shamberger, K., Brainard, R., and Golbuu, Y. 2015. Coral macrobioerosion is accelerated by ocean acidification and nutrient. Geology 43, 7-10, doi:10.1130/G36147.1

R3: The interval of measured elevation changes included the loss of A. palmata. It, therefore, seems likely that this could have played a greater role than the removal of bioeroded sediment in the changes described in the manuscript. Alvarez et al also pointed out that the loss of Diadema logically re- duced bioerosion despite the greater availability of "bioerodable substrate". Likewise, in many (most?) Caribbean and western Atlantic sites, parrotfish populations have been decimated, further reducing the potential for bioerosion by grazers. The remain- ing option is infaunal bioerosion by sponges, worms, etc. However, unless there is a very significant increase in organic availability, the likelihood of that being significant seems unlikely.

AR: We agree that loss of A. palmata and other framework building species likely contributed to elevation loss at these study sites, and we address this issue in section 3.2 beginning on page 12. To estimate the contribution of this process to total net volume loss, we calculated the percent of each study area classified as coral-dominated

substrate (note coral-dominated habitat classes were denoted in Tables 2 and 3 in manuscript with an asterisk). For example, the Upper Florida Keys coral-dominated habitat classes included scattered coral rock in unconsolidated sediment, aggregate reef, reef rubble, individual or aggregate patch reefs, and spur and groove habitat. The other study sites included similar lists of coral-dominated habitats. We included reef rubble as a coral-dominated substrate in our calculations because, for example, in the Florida Keys many areas of reef rubble contain large skeletal fragments of A. palmata and other species. We calculated that 91% of the Buck Island study area was covered by coral-dominated substrate types that accounted for more than 90% of the total net volume loss at this site. We stated that loss of framework building coral at this site may be the primary contributor to volume loss (page 12, 28-30), but we do not attribute loss of the framework building corals to bioerosion or any other process. We suspect that physical toppling of coral colonies from (e.g., from storm and wave impacts) likely contributes here as well, but do not have adequate data to quantify that (see additional discussion on this topic in comments later in this review response).

However, the areal extent of coral-dominated substrate in the Upper and Lower Florida Keys and St. Thomas was only 8% to 15% of the total study area. These coral-dominated habitats contributed up to only 26% of the total net volume loss that we measured for these study sites. Most of the volume loss in these areas was associated with sediment loss from non-coral-dominated habitats such as areas of unconsolidated sediments, seagrass, and uncolonized pavement (Table 3 in manuscript). The Maui study site was characterized by 57% coral-dominated substrate that accounted for 50% of the net volume loss. We do suggest that physical erosion and removal of sediment is a likely driver of much of the volume loss at these study sites rather than degradation of large framework building coral because the total areal extent of coral-dominated substrate cannot account for the loss, and most of the volume was lost from non-coral dominated substrate types.

R3: It is interesting that at one of their sites (Buck Island), Bill Gladfelter proposed two

threats to reef building in a 1977 report to the Park Service: 1) the loss of carbonate production if WBD increased, and 2) the possibility that protection of parrotfish might significantly increase bioerosion to the point where it could overwhelm even productive reefs. This would suggest that increased bioerosion by grazing fish could lead to detrimental increase in bioerosion. In the latter scenario, increased grazing becomes a problem only in protected areas where grazing fish have increased (like the FKMS, one of the described sites where increased bioerosion might be a reasonable culprit). Elsewhere in the Caribbean, parrotfish populations have been decimated. In combination with the loss of the major grazing urchin, a wholesale increase in bioerosion capacity seems unlikely. Lost calcification ability would decrease accretion, but does not seem like a driver of net erosion unless bioerosion increases – a pattern that has not been documented at all sites.

AR: We agree that losses and gains of grazing fish will directly impact rates of bioerosion by those species. We realize we need to clarify in our discussion that there are a number of different types of bioeroder species including grazing fish as well as sponges, bivalves, microendoliths, etc. We will note the potential impact of increases or decreases in grazing fish on rates of bioerosion; and we will clarify that recent work, cited earlier in our response, indicates that elevated pCO2 and ocean acidification (and especially OA combined with increased nutrients) accelerates bioerosion by endoliths and other boring bioeroders.

R3: So, that leaves us with export. As the paper points out, good data on export are rare. On page 2, Moses (2009) is cited for measuring sediment export from reefs, but I could find no measurements in that paper. Kench and McLean provide an estimate of trans- port potential through hoa in Indian Ocean atolls. However, the results are based on theoretical calculations and there is no effort to tie sediment to specific sources (e.g., bioeroded sediment, beaches, lagoons) or sinks (loss to lagoons vs export from the platform).

AR: The reviewer is mistaken regarding the Moses (2009) reference. Moses (2009)

is cited with respect to regional-scale chemical erosion measurements of carbonates (page 2, lines17-20). In a sentence after that (beginning on line 21), Morgan and Kench (2014) and Kench and McLean (2004) are cited in a sentence that reads "Very few studies have quantified sediment transport and export on reef systems". We note here that many of our references in the paper have typos in them from our End Note conversion that we did not catch prior to submission. For example, many were abbreviated (e.g. Morgan and Kench 2014 got shortened to Morgan 2014, and Kench and McLean 2004 got shortened to Kench 2004), and others reverted back to numbers (as was the case in Supplementary Table 1 in which the source references reverted back to numbers). All of these will be corrected in a revised paper, and we apologize for the confusion this may have caused.

Morgan and Kench (2014) is one of the few studies that estimated sediment flux and off-reef sediment export from direct point measurements using arrays of bi-directional sediment traps for an atoll reef (Vabbinfaru reef, North Male' Atoll) in the Maldives. They showed high off-reef export rates for both gravel and sand (annual export estimated at over 120 metric tons per year), as well as high sediment flux rates even during non-storm conditions.

Kench and McLean (2004) estimated transport potential from current meter and tidal gauges located in the HOAs, but also measured sediment fluxes in hoa's from sediment traps ranging from approximately 2kg to 268 kg per day. Based on daily rates of sediment flux, they estimate that from 44 to 223 metric tons per year of sediment may be transported by hoa's.

We recognize that there are other previous studies (e.g., work by Ogston, Presto, Storlazzi, Fletcher, Hubbard) on sediment transport on Caribbean and Hawaiian reefs that can be included, and will add these along with more detailed discussion of sediment transport and export to a revised manuscript. We note that the reviewer performed very detailed studies of sediment transport and export at St. Croix in the U.S. Virgin Islands including assessment of the physical and biological processes affecting sediment transport across reef zones (Hubbard et al. 1981), the impact of storms on sediment transport and export (Hubbard 1992), detailed carbonate budgets that account for export (Hubbard et al. 1990), and the effect of sedimentation on reef development (Hubbard 1986). Many of these studies suggest that physical processes (waves and storms) dominate with respect to transport and export of sediments from this reef system. Results from these (and other studies) provide evidence that very large amounts of sediment are transported within reef systems and exported from reef systems by physical transport of materials. These results support our suggestion that physical erosion and export of sediments could account for much of the volume loss we observed at our study sites (page 12, line 25-28).

R3: What is, therefore, critically important is a reliable estimate of export inas- much as volume must be exported from the system to trigger system-wide elevation loss. . . bioerosion just converts carbonate from solid substrate to sediment. In the latter case, we must remember that sediment has a much lower bulk density than solid carbonate substrate. Thus, increased bioerosion without export would reduce the volume of solid substrate but would turn this into a sediment pile with something akin to twice the net volume. Thus, increased bioerosion without export would result in substrate elevation; not lowering. A scenario based solely on increased bioerosion seems inadequate to explain the measured patterns.

AR: We agree with the reviewer, and make the case in our results and discussion that the volume loss we observe over the large scale of our study sites is an indication of export of materials from these systems. We do not suggest that any scenarios based solely on increased bioerosion could account for the volume losses we observed. We will reword the sentence on page 15 line 3-4 regarding bioerosion to clarify that export of that amount of bioeroded sediment per year could only account for as much as. . . .." as we suspect the reviewer may have misinterpreted our meaning in this sentence. The point we are trying to make here is that much more sediment is being physically eroded and exported from these systems than the amount that is annually generated by

bioerosion alone and exported. This is likely why we are seeing a broad-scale decrease in sea floor elevation over all habitat types rather than stability or accumulation. We will reword our discussion regarding physical erosion to more directly convey that message, and include additional references (as discussed above) to support our observations.

Note, we specifically do not convert our volume measurements to mass of carbonate because we recognize that the volume change we measure encompasses loss of sediments as well as framework building coral colonies of different porosities. We recognize the challenges in estimating the conversion of volume of framework building coral colonies (including the coral itself as well as the pore spaces in the colony framework) to volume of bioeroded sediment of potentially varying porosities. . .and these exercises are worthy of papers themselves.

R3: Unfortunately, there has only been a few careful measurements of sediment export in the context of a reef-wide budget. Perry and various co-authors use our ratio (Export âĹij 50% of total bioerosion) from the north coast of St. Croix to characterize this in every one of their budgets. It is naïve to think that all reefs in all oceans have the same energy regime (the driver of export) – or that changes in energy regime is offset by proportional shifts in bioerosion to maintain the 50% value that is used throughout. With increasing storminess, sediment export looms as the single largest unquantified variable. Therefore, export can only get more significant in the budgeting attempted in this paper.

AR: Again, we agree with the reviewer. Note in our discussion that we suggest that much of the volume loss we observe in these regions may be attributed to physical transport and export of sediments, and that export has been largely unaccounted for in previous studies (page 15, lines 10-12). We also note that the much greater seafloor elevation and volume losses observed over shorter time periods at the Maui study site could be caused by higher sediment export rates due to a combination of higher wave energy and physical erosion as well as a narrow shelf surrounding the island (page 14, lines 1-3). We feel that ecosystem-scale elevation change studies such as those

we performed will be instrumental in better quantifying large-scale sediment transport and export by way of accounting for total volume change in sediment budgets. These types of data sets will also assist with mapping elevation change patterns that suggest offshore movement of sediment such as the example we provide in our comment to Reviewer 1 showing movement and accumulation of sediment offshore and downslope along the Florida reef tract.

R3: In section 3.2, the paper acknowledges the difference between bioerosion and changes in structural complexity. How good the conclusion will be is going to depend on how well one can distinguish between the two as potential drivers of elevation change. The conclusions presented here seem to suggest 1) an ability to reliably distinguish between the two mechanisms and 2) an overwhelming importance of simple bioerosion over combined changes in export and reduced structural complexity following the loss of biological constructors.

AR: We are confused by this comment. We do not discuss bioerosion anywhere in section 3.2. We do discuss that greatest mean elevation losses in the Florida Keys were associated with shallow-coral dominated substrate and that this observation is consistent with observations of general flattening of reef topography and decreasing abundance of reef-building corals in previous studies. We make no claims as to the cause of the elevation losses in the coral-dominated habitats, nor do we claim to be able to distinguish between bioerosion and loss of structural complexity. Additionally, we claim that physical erosion and export is the likely driver of much of the volume loss that we observe. We do suggest that, at the Buck Island study site, much of the volume loss may be due to degradation of framework building corals because 91% of the study site was characterized by coral-dominated substrate. We do not attribute degradation of framework building corals to any particular cause in this discussion. We do, however, note that Buck Island showed the lowest net volume losses of all of our sites suggesting that much less sediment has been exported.

The main conclusions of our work focus on the large magnitude of elevation and volume

loss from these study sites that has been previously undocumented, how that has impacted local sea level rise, and the observations that most substrate types within these ecosystems are losing elevation and volume, not only coral-dominated substrate. As previously stated, we compare our measured rates of net erosion to rates for individual processes that cause erosion (including bioerosion, chemical dissolution and sediment export) in a paragraph in our discussion. We feel, from the reviewers comments, that the intent of that exercise has been misunderstood. We will reword this section to clarify our intent to demonstrate the magnitude of erosion we observed relative to known rates for individual erosion processes.

R3: Anthropogenic drivers of change - Using population as a proxy for anthropogenic im- pact seems overly simplified. Numerous recent papers have shown that some of the greatest reef losses occur due to warming/acidification at great distances from any rec- ognizable urban stressors. I can't find the specific papers, but there has been quite a bit of discussion on the NOAA listserve about papers that show just this. While I am not in the midst of the debate over local versus global drivers of change and their implications for management, this proxy seems a bit simplistic.

AR: We define relative anthropogenic impact within each study site based on the simplest first order parameter of population and note that the historic data sets for each site were from time periods characterized by approximately half or less the population than the date of the modern data set. We believe the reviewers would agree that local anthropogenic impacts have increased over the historic to modern time periods of each data set for each of our study sites where population has doubled. We agree that this is a very simplified proxy, and state that full analysis of anthropogenic impact factors is beyond the scope of the paper (see page 3, lines14-21). We could, reluctantly, remove reference to anthropogenic impact and simply state that population has doubled at each of the study sites (except for Buck Island) over the historic to modern time periods. But suspect that others would then criticize that we don't recognize that there has been anthropogenic impact. Our preference is to leave this as-is.

We do not attempt to attribute reef ecosystem degradation to either local anthropogenic or global climate impacts, and we don't disagree that reef losses due to global stressors have occurred in geographically isolated, low population areas (we can add some of these references). We do state that projections indicate that impacts from these stressors are likely to increase in the future. We cite Perry's (2015) work showing that reef systems in the Indian Ocean that are geographically isolated and lack human influence show heavy impact from bleaching events, but are able to recover very rapidly (see discussion on page 16, lines 1-4) and show very high accretion rates. We note that Maui is geographically isolated but shows very large erosion rates suggesting that it has not been able to recover from reef loss, possibly due to the fact that it is not isolated from a large human population. We also point out more than one case where lower losses of elevation and volume coincide with areas further removed from large population centers or associated with natural refuge zones (page 16, lines 4-6).

R3: On a more specific point, the manuscript discusses the idea of proximity to anthropogenic areas to explain the positive elevation change in the lower Keys. Couldn't this also be due to separation from the inimical cold bank water allowing for higher calcification rates? In this vein, limited core data from the Keys seem to suggest that the "demise" of the reef tract likely started 4-5,000 years ago as Florida Bay flooded, triggering inimical (cold) water export onto the reefs. In contrast, the reefs around Buck Island enjoyed continuous building throughout this period as there was no similar source of stress.

AR: The reviewer misunderstood this discussion located on page 13, lines 8 – 12:

"Mean total elevation loss was lowest at the UFK study site. However, mean elevation losses decreased from upper (-0.4 m) to central (-0.3 m) sub-regions of the UFK, and mean elevation increased slightly in the lower sub-region (0.1 m) primarily associated with seagrass habitat (Fig. 1a, b). Notably, the lower sub-region is further away from high-density population areas north of the study site and near an area of the middle Florida Keys identified as a possible refuge from ocean acidification due to seagrass

productivity (Manzello et al., 2012)."

The area that showed the elevation increase was in the lower sub-region of the Upper Florida Keys (not in the Lower Florida Keys), see Figure 1a in the manuscript. The mean elevation change in the Lower Florida Keys was -0.3 m (see Table 1 in manuscript), and was similar to the elevation loss we observed in the central sub-region of the Upper Florida Keys. These data would suggest that the inimical waters could be less likely to have had an impact on this sub-region of the Upper Florida Keys.

R3: All of this would suggest that these two areas have had very different exposures to natural stresses; this would presumably make for very different susceptibilities in more recent times when increasing anthropogenic stress is set up as the main driver. It may also be noteworthy that the sediment thicknesses in these two areas are different and there is evidence that sediment retention around Buck Island (much higher wave energy and susceptibility to both storm damage and sediment export) may tend to be less than is the case in the Keys. If the latter is true, then changes in substrate elevation might be sediment export in one place, bioerosion in another and wholesale loss of rugosity in all.

AR: Interestingly, our volume calculations showed the lowest 'maximum net volume' loss (Table 1, column 3 in manuscript) and 'mean elevation' loss (Table 1, column 10 in manuscript) at Buck Island, suggesting that sediment/material retention was greater at Buck Island over the study period than in the Florida Keys. We suggest that this could be due to loss of rugosity (large framework building coral) but less export of materials at Buck Island. Recent photographic surveys (2015) along the eastern coast of Buck Island (unpublished, C, Storlazzi, USGS) show large stands of Acropora palmata coral colonies that have died and toppled over, but remain largely in tact and in place on the seafloor along the northeastern side of the island from approximately the eastern-most point to at least the mid point of the island (see Figure 1a, b, and c in this response), consistent with our observation of greater elevation loss in that area (see Figure 2c in manuscript). The southeastern side of the island from the eastern-most

point to at least the barrier break was characterized by much more live A. palmata coral (see Figure 1d, e, and f in this response), also consistent with our observation of increased elevation in that area (see Figure 2c in manuscript). These photos illustrate how elevation and volume can be lost without a large amount of export. The live coral colonies clearly show relatively high elevation and colony volume that consists of both coral branches as well as very large open (pore) spaces between the branches. As large colonies topple, the coral branches break and the colony compacts causing a loss in elevation and volume as the open spaces between the branches are minimized. Large coral fragments are much harder to transport than sediments, typically require high-energy storm events for movement, and are, thus, not as easily exported from the system. This type of physical coral degradation is very different from degradation due to bioerosion by grazers whereby the coral is more directly reduced to sand that can be more easily transported and exported. Additionally, coral rubble fields generated from toppling corals can create a rubble pavement on top of sand making it more difficult to transport. Our observations of much lower elevation and volume loss at Buck Island relative to our other study sites combined with our observations that 91% of this study site was covered by coral-dominated substrate that accounted for over 90% of the volume loss is very consistent with this type of physical coral degradation process. The reviewer notes that parrotfish populations have decreased in the Caribbean, so it is also possible that less sand sized material is being generated for easy export at Caribbean sites than at sites with higher rates of bioerosion; and that may also help account for the lower export rates at this study site. We can include this discussion and the photos in Figure 1 (in this review) in the revised manuscript in support of our suggestion of physical degradation as a key process here; but, again, we do not have adequate data to fully quantify the contribution of physical degradation.

R3: I assume that the substrate type and sediment thickness was not consistently noted in older surveys. Given the points above, this could be an important driver of how quickly substrate elevation might change in one place versus another. The wholesale loss of architecturally complex acroporids and the subsequent reduction of these to

pavement could be construed as "degradation of framework-building corals" as could bioerosion. Which was the main agent in each case?

AR: Unfortunately, substrate type and sediment thickness were not consistently noted in the older surveys.
* * *
[Figure]

**Fig. 1.** Toppled (a -c) and live (d - f) Acropora palmata coral colonies along the coastline of Buck Island, USVI

---

## Author Comment (AC5) · 13 Dec 2016

AR: We appreciate the effort from Reviewer 4 in considering all available reviewer comments and for synthesizing the most salient points. We found this very helpful in further organizing an approach to a revised manuscript. Individual comments are below (R4 indicates reviewer comment, AR indicates author response).

R4: The paper by Yates et al. analyses bathymetric data to quantify seafloor elevation changes in coral reef regions. As highlighted by reviewers 1 and 3, the dataset presented in the manuscript is especially impressive (number of sites considered, extent of the area considered). The results will be useful for coastal geomorphologists and managers concerned with the sustainability of coral reefs environments and the related ecosystem services. Despite the recommendation of reviewer 2, I think that the paper

should be published after major revisions, to ensure that the amount of data analyzed in this work receives the attention it deserves.

Three previous reviews have extensively discussed the paper: overall, reviewers (1) have concerns regarding the ability of the method to retrieve seafloor elevation changes at the required accuracy; (2) made comments on the form of the paper; (3) and the interpretation of the results. The authors have already provided responses to several comments of the reviewers, and intend to implement corrections to their paper, which I think are reasonable. I would suggest that these major revisions are implemented, considering the following points:

- All reviewers agree that the paper should separate more clearly what is the overall approach (comparing bathymetric data) from the technical details of the GIS procedure used to produce this data. The authors have prepared a figure as part of their response to reviewer 1 to address this comment. However, I think that the figure remains too technical (e.g., use of the TIN surface wording), and I would support producing the detailed GIS procedure in an annex to the paper. Overall, I agree with previous reviews, who suggested that the authors should consider that their results may have a large impact beyond specialists of coastal bathymetric surveys, so that ideally, they should try to separate the main messages from the technical implementation details.

AR: We agree that separating the technical implementation details (GIS procedures) from a more clear discussion of the approach is an excellent idea, will greatly improve the paper, and will make it more accessible to a broader audience. We will rewrite the methods taking into account the suggestions by all of the reviewers, and will include the more detailed (GIS) steps in a methods supplementary section. We have included an example of a modified flow diagram showing our approach to describe the process in more general terms (See Figure 1 in this review response).

R4: - While the comparison of historical bathymetric sounding with contemporary Li-DAR is quite widespread in coastal geomorphology (as reminded by the authors, see

AC2 – pages C6-C7), there is always the suspicion that the two techniques induce errors, as highlighted by reviewers 1 and 2. Such errors can arise because the techniques have not the same purpose and therefore don't necessarily capture the same proxies (e.g., highest seafloor elevation features for navigation applications vs average seafloor ele- vation feature for bathymetry data in support to coastal hydrographic modelling). The techniques also have different accuracy/precision (as discussed already), or because of time-sampling issues (as commented by reviewer 2). Overall, I think that the authors make a fair assessment of these errors: in the response of the authors to this com- ment of reviewer 2 (AC2 pages C8), information regarding the vertical resolution of the techniques is provided, while the precision issues are given in Table 4 in the original manuscript. To complete this assessment, I would suggest to provide more information on the planimetric resolution, and the vertical accuracy of the techniques in the core of the article. This includes details regarding the definition of a common reference, which incorporates sea-level rise constructions in a way, which is not completely clear to me based on the original manuscript, page 4 lines 10 and following. Nevertheless, I am confident this does not affect the results of the authors, as the RMSE in vertical da- tum adjustment is probably much larger than the RMSE due to uncertainties in relative sea-level changes for the sites of interest (table 4).

AR: We will use our responses to the reviewers regarding the vertical and horizontal error assessments and data resolution to expand and improve our discussion of these issues in the main manuscript. We will also include our pavement analysis and results as further evidence for the validity of comparing these data sets (with a strong caution that pavement cannot be appropriately used as a control, but can provide supporting information for proper error analyses). We will provide information and definitions (in general terms) of horizontal and vertical references (datums) for these data sets and the need to adjust data so that compared data sets are aligned to the same reference points. The data and sources we used for our sea level rise corrections were included in Table S2 of the supplementary section, and we will move this information to the main paper to help clarify our sea level rise adjustments. In general, long-term sea level rise

data recorded by NOAA sea level trend stations in mm per year were used to calculate the total sea level rise at each study site by multiplying the mean rate of annual sea level rise over the study time period by the number of years between historical and modern data sets. These correction values ranged from approximately 7 cm to 16 cm and were added to the historical sounding value. NOAA reports 95% confidence intervals for these data ranging from approximately +/- 0.15 to +/- 0.81 mm/yr. The potential error from these corrections was insignificant relative to other sources of error, and we, therefore, excluded it from our RMSE calculations.

R4: - Regarding the interpretation of the results: besides the aspects discussed with re- viewer 1 and 2, I think that reviewer 3 provides a very clear line for improving the discussion section, and I hope that the authors will build on it in a future version of the article.

AR: We agree that comments from Reviewer 3 were very helpful for improving our discussion section, and we have provided discussion on how we will use those comments to improve our paper in our Response to Reviewer 3.

R4: - Finally, I think that a "conclusion" section is needed.

AR: We agree that a conclusion section will improve the paper, and will include that in a revised manuscript.

R4: I hope these comments are useful.
* * *
[Figure]

Negative values = losses, positive values = gains, net volume change = volume loss + volume gain

**Fig. 1.** Modified methods flow diagram for incorporation into revised manuscript

---

## Author Comment (AC6) · 16 Dec 2016

Again, we thank the reviewer for thoughtful comments that will help strengthen our revised manuscript. We will make sure that it is clear that the goals of our study were to focus on the magnitude of elevation change and to demonstrate the value of these types of elevation analyses, and not to provide a comprehensive process analysis. We will include a discussion of knowledge gaps with respect to attribution to mechanisms in that discussion. We hope that the results of our study and further use of these methods by the broader community will facilitate more in-depth process studies and projections of long term physical and societal impacts to coral reef ecosystems and the coastal communities who depend on them.

---

## Author Response (AR1)

**Yates et al., bg-2016-407**

**Point-by-point reply to reviewers' comments (reviewer comments in bold):**

Note that much of the text developed in our published, detailed responses to the reviewers was incorporated into our revised manuscript. However, we have removed much of that detailed discussion from this response to avoid repetition where possible, and to focus on the specific locations in the manuscript where improvements have been made. We feel that all of the reviewers' comments have helped us greatly improve the paper. We have added all of the references recommended by the reviewers as well as those indicated in our published responses to the reviewers, and discussion of those references in context with our results. We have rewritten the methods and substantial pieces of the discussion as recommended by reviewers, and we have moved the technical data analysis procedures to a supplementary methods section. We have also added 6 figures that further explain our methods and support our results. As recommended, we have moved many of the Tables from the Supplemental section to the main paper to support our description of the methods. We have also included a conclusions section. Point-by-point replies to all reviewers' comments and specific page and line number references to changes are below.

**Reviewer 1**

**Page 1, Line 1 : All dictionaries define ´n sea floor Â˙z as ´n the bottom of a sea or ocean Â˙z. Accordingly, using this term to describe the Pre-Holocene bed rock that underlies a coral reef body and forms its foundation is inappropriate.**

The page and line reference for this comment refers to the manuscript title, which makes no reference to Pre-Holocene (or Pleistocene) bedrock. In Section 2.4 'Lower Florida Keys – volume to Pleistocene bedrock', we discuss the thickness of the Holocene reef layer lying above Pleistocene bedrock. The Pleistocene bedrock lies buried below the surface of the seafloor except for where it is exposed in some places. We have added a more detailed discussion of the formation of Pleistocene bedrock and its relation to the Holocene Reef formation of the Florida Keys on page 25, lines 5-24.

**Page 3, Line 17 : I question the use of ´n the number of people living close to the reef sites Â˙z as a parameter for anthropogenic impacts. Is the number of inhabitants the reflection of the local human activity (fishing, : : :.) ?**

We have reworded this section to only indicate that population has doubled over the time period of our study at each populated study site on page 5, lines19-22. We have also added information on trends in live coral cover and other indicator species as evidence for anthropogenic impacts on page 4, line 20 – page 5, line 5.

**Page 12, Line 16, about ´n the coral-dominated habitats Â˙z : It would be useful to have some information about the living coral cover. This will inform the debate on the real state of health of each studied reef site.**

We have provided a summary of changes in live coral cover, diadema sea urchin densities, parrotfish biomass, and macralgae cover in a paragraph on page 4, line 20 – page 5, line 5. We have also included table 1, that summarizes results of this discussion.

**Page 14, Line 8 : about the chronic erosion processes. These are natural processes affecting reef systems. Reef growth reveals to be the subtle balance between constructional and destructional processes. They occur continuously on both pristine and degraded systems.**

We fully agree and make no statement to the contrary.  In the cited paragraph, our aim was to make the point that our estimates of seafloor elevation change reflect the net result of all constructional and erosional forces affecting these regions. We have included more detail in our discussion on pages 19-20 regarding both the advantages and limitations of the methods we used in our study, and we have also included a list of key limitations in the conclusions section on page 27, lines 1-7.

**Page 14, Lines 20 – 21 : It is clear that reefs that are located closed to urban areas are suffering significant deterioration. It would be intersting to compare these results with a reef system located in a remote and not inhabited area.**

We agree. With the exception of Buck Island (which is uninhabited, but not remote), we were unable to locate sufficient historical and contemporary bathymetric data to perform our analysis in more remote, uninhabited locations. We would be very interested in such a comparison as well.

**Page 14, Lines 26-27-28 : about the assertion ´n coral reefs in all three regions will be unable to keep up: : :: : :: : : (Church et al., 2013) Â˙z. This is an overinterpretation of the data presented herein. Using mean rates of reef accretion established at the scale of the Atlantic and Pacific to infer future responses of reefs to the rise in sea level is not receivable. A number of previous studies worldwide indicated that vertical reef accretion varies from site to site in a given region. There, some reefs will be able to maintain pace with sea level, while others will be unable to compensate for sea level rise.**

We gave rewritten this section, now on page 21, lines31-34 to indicate that our study sites, specifically, we be unable to keep pace with rising sea level. We have also included discussion of studies showing coral reef islands and ecosystems that are keeping up with rising sea level including work by Kench, McLean, Hopley and Kinsey on page 22, line 26 – page 23, line 15. We compare results from these studies with observations from our study sites.

**Page 15, Line 29 and Page 20, Line 23 : Please correct the reference : Neumann and Macintyre, 1985**

We found numerous reference errors in our original manuscript to due problems with End Note conversions. We believe we have now corrected all of these errors.

**Reviewer 2**

**General comments:**

**Both the scientific quality of the paper and its presentation quality are generally insuffi- cient.**

**The scope of the paper, as it is formulated pages 2 and 3, i.e. "measuring changes in seafloor elevation to assess and predict the impact of reef degradation on the vulnera- bility of coastal communities to sea-related hazards" is confusing.**

We have rewritten this section, now located on page 3, line 11 – page 4, line 2 to more clearly state the aim and scope of our study.

**The title of the paper itself is also unclear.**

We chose this title because it summarizes the major finding of our work, namely that seafloor elevation is decreasing (rather than increasing) while sea level is rising in the coral reef ecosystems we studied (thus, we use the term divergence); and the combination of seafloor elevation loss and sea level rise has accelerated the relative increase in water depth at these locations. The title of the paper is derived from our concluding statement on page 26, lines 19-21 of the revised manuscript.

"The divergence between rising sea level and declining seafloor elevation has already increased the risk to coastlines in these regions from long-term, persistent oceanographic pressures and periodic events such as storms."

We have rewritten the title as "Divergence of seafloor elevation and sea level rise in coral reef ecosystems" to more accurately limit our conclusions to the sites we studied based on comments from Reviewer 2 regarding page 14, lines 26-28 in the original manuscript.

**The authors announce that they address the coral reef issue and then they provide results on various habitats, including non-coralline habitats and even deep water offshore habitats (e.g., page 13, lines 3-5).**

We have added a paragraph on page 4, lines 4-10 that states what we include in the coral reef ecosystems that we analyzed, as well as justification for including non-coral dominated habitats within these coral reef ecosystems.

**Authors are unclear on what they measure, and on my view they fail in generating robust data (as both the data and the methods used lack accuracy. Thereafter, the presentation of the results and their interpretation are confusing, as various types of processes are invoked to explain changes, with no specific process being robustly studied (e.g. page 13, lines 30-34).**

We have made a number of changes in the paper to help clarify the methods we used in the paper, and the rigor with which we evaluated accuracy and error associated with our results:

1) We have created and included a flow diagram (Figure 1 in revised manuscript) that depicts the core processing steps in our methods.
2) We have rewritten the methods in more detail and more clearly for a broader audience, and we have moved the technical data processing steps to a supplementary methods section.
3) We included Section 2.2 (page 7, lines 15-31) on the methods we used for preliminary inspection of historical data sets during the selection process, including an example figure of that process (Figure 2).
4) We included more detailed discussion on developing the elevation-change surface models used in our study and an example figure (Figure 3) to further illustrate the process. We have also included photographs from visual inspection of our study sites as supporting evidence for our discussion (Figures 8 and 9).
5) We have included more detail on the very rigorous data and error evaluation analyses that we performed including a detailed discussion of data resolution and allowances for historical and modern data (page 6, line 10 – page 7, line 13).
6) We also included methods and results for pavement analyses we performed to determine validity of comparing historical and modern data sets (page 13, one 25 – page 14, line 11) prior to beginning elevation-change analyses, and a figure of results from this analysis (Figure 4).
7) We have also modified our discussion of error calculations to better clarify these methods page 11, line 15 – page 15, line 20.
8) We have moved supplementary Tables S1, S2 and S3 that describe details regarding the data sources, data conversions and study periods for each study site to the main paper, now Tables 2, 3 and 4 in revised manuscript, to help further clarify our methods.

Additionally, we have added a detailed discussion emphasizing that a key limitation of our elevation-change analyses is that they do not attribute change to cause and that a detailed analysis of the processes causing elevation-change in these systems was beyond the scope of this paper. We explained that we provided general comparisons of our elevation and volume-loss measurements to individual process rates from the literature to provide context for the magnitude of our results. These comparisons also demonstrate the potential value of knowing net whole system change when accounting for and identifying individual processes, for identifying missing gaps in erosion/accretion budgets, and serve as an example of how our results may be used in future studies to improve erosion and accretion

budgets in coral reef ecosystems. This discussion is now located on page 19, line 24 –page 21, line 2 in our revised manuscript.

**For example, the first paragraph of the Discussion Section clearly illustrates the wide (and unprecise) area covered by the paper (see page 14, lines 5-15).**

We do not discuss the area covered by our study sites in this paragraph in the original manuscript. However, our study sites are very clearly defined on the maps in figures 1-3 of the original paper (now 6-7 in the revised paper) and the exact size of each study area as well as each habitat area within each study site is provided in Table 10 (in revised manuscript). The paragraph on page 14, lines 5-15 discusses the large number of processes that cause seafloor elevation and volume changes and the very general time frames over which they occur. We are, therefore, confused by this comment.

**Both the concepts ("change in seafloor elevation and volume" – in fact, it seems that the authors address "changes in shallow waters depth") and the method used (method "traditionally used to monitor seafloor changes", "use of historical bathymetric data from the 1930's to 1980's and LiDAR DEMs from 1990's to 2000's) –which are presented firstly in the introduction of the paper (pages 2-3) and then in the Methods Section (page 4) – are questionable and not accurate.**

We have now included a new section in the revised manuscript title "Section 2.1 Selection and description of data sets beginning on page 5, line 12. We provide a very detailed discussion of the cautions that must be applied when selecting appropriate historical and modern data sets for elevation-change analyses, as well as detailed information on the rigor and resolution with which these data sets were collected and QA/QCed.

**Concerning data and methods - How can bathymetric data from the 1930's to 1980's (constituting a single coherent period reflecting low anthropogenic impact?) be con- sidered as a starting point (or reference) to "measure changes in seafloor elevation" and then be compared with data from the 1990's to the 2000's (= period reflecting high anthropogenic impact?).**

Each of our five study sites was analyzed independently over a specific time period using the oldest reliable data that we could find for each site as well as the most recent bathymetric data available for that site. We have clarified this in the revised manuscript on page 5, lines 13-22. We did not measure change over a continuous historic range of 1930's to 80's and modern range of 1990's to 2000's. For example, the Upper Florida Keys historic data set was from 1934 and 35, and the modern data were from 2002. The dates for the individual data sets are now available in Table 3 of the revised manuscript. The aim of the study was to look at change in seafloor elevation over several decades. Within the time frame of the study, population approximately doubled at each of the populated study sites (with the exception of uninhabited Buck Island).

**R2: This raises several key questions. Firstly, how can the "mag- nitude of erosion" (page 3, line 9) be measured using such an approach that poses serious questions relating to the scientific quality of both the datasets and the method used. In other words, how can historical bathymetric data be compared with LiDAR data? The former (bathymetric data from the 1930's and next decades) do not have the required resolution for comparative measurements to be undertaken with LiDAR data. The low resolution of historical bathymetric data may generate significant errors in the results generated. Incidentally and curiously, no clear and complete information is provided by the authors on the resolution of the various datasets used in this study at the various study sites.**

We have now included detailed information on the resolution of the historical and modern data on page 6, line 11 through page 7, line 13. We have also included methods and results for pavement analyses that we used to show the validity of comparing historical elevation data to modern LiDAR elevation data, Section 2.12, page 13, line 25 – page 14, line 11. We have also included point density information for the historical data sets on page 8, lines26-28, and note the potential impact of point density on error in these data sets. We prove through both vertical and horizontal error analyses the potential impact of these errors on our results in sections 2.11, 2.12, and 2.13 on pages 11-15, and confirm the statistical significance of our elevation and volume calculations.

**Authors first indicate a 1 to 4Âaˇm horizontal spatial reso- lution (this is a low resolution that do not allow the calculation of changes in the reef level) and then indicate a 11-12Âaˇcm vertical resolution (which is questionable given the data used).**

The 1 to 4 m horizontal resolution applies to the LiDAR data sets only. We calculated horizontal error for the oldest historical data using published values for the methods (section 2.13, beginning on page 14) of 4.8 m. To estimate the effect of horizontal error on our seafloor volume results, we performed a horizontal shift analysis. For this analysis, we doubled our calculated horizontal error to 10 m, then shifted the historical data set relative to the modern data set by 10 m in each of the four cardinal directions (N, S, E, W) and recalculated volume change for each scenario. These results indicate that horizontal error of up to 10 m and the resulting offsets in sounding points affects our volume calculations by 10% to 21% (depending on density of data points) and does not change the outcome or conclusion of our study. These results are consistent with reports that, over large areas (such as in our study), random errors largely cancel-out relative to change calculations derived from two surfaces (Byrnes et al. 2002). Additionally, this analysis provides further evidence that seafloor elevation loss is occurring at a very broad-scale across all habitat types.

With respect to vertical resolution, the reviewer's statement is incorrect. Nowhere in the paper do we claim vertical resolution of 11-12 cm. We believe the reviewer may be making reference to the VDatum transformation error that we report for each data set that is only one parameter used to calculate total error (see page 8, lines 30-32 and

Equation 1 in original manuscript, now in Section 2.11 beginning on page 11 of revised manuscript). We state the maximum cumulative uncertainty for operational VDatum regions of South Florida and the Virgin Islands are 9.6 and 11.8 cm, respectively as reported by the NOAA National Ocean Service. We also report the VDatum error that was calculated specifically for each individual study site under the RMSEVDatum column in Table 5 of the revised manuscript that ranges from 8.1 to 11.4 cm (note no VDatum error was reported for Maui because no VDatum transformation was required). Total vertical error for each study site was calculated using equation 3, reported in Table 5 under the column heading RMSETotal, and ranged from 20 to 37 cm (average of 29 cm for all sites). We used a very conservative approach in our consideration of vertical error by multiplying our $RMSE_{Total}$ by a factor of 1.65 to encompass 90% of the variance in our data. This approach generated a more conservative RMSE of 0.48 m that we rounded up to 0.5 m; and we used this value to set minimum and maximum bounds in our volume calculations (Table 8 in revised manuscript). The minimum volume change values that we report in Table 8 were calculated by only including elevation changes that exceeded the range of -0.5 to +0.5 m to provide a very conservative estimate of volume change. These very conservative minimum volume change values also support our conclusions of net seafloor erosion at all study sites (Table 8 in manuscript).

**A second methodological problem is raised by the way anthropogenic impacts are considered in this study. (1) How can the "anthropogenic impact" only be measured by population numbers? In the present case (changes in water depth), it mostly depends on coastal and maritime human practices (sustainable/not sustainable). Major human activities, such as dredging in the substratum (should it be coralline or not) and extract- ing aggregate in particular, which may have occurred over the study period at some study sites and may have changed water depth, are not considered at all by the au- thors, which introduces a serious bias in the "elevation changes measured". Some parts of the paper, such as "However, greatest mean elevation losses occurred in coral-dominated habitats and near the central coastline where harbour and shipping channels existÂaˇÂz˙(page 13, lines 22-23) clearly indicate that not taking into account these human activities is problematic when assessing changes in shallow water depth.**

This claim is incorrect. In fact, on page 14, lines 8-18 (first paragraph of our Discussion in the original manuscript), we specifically state that we DID include changes due to episodic events including, for example, dredging and infilling of channels and coastal harbors. This statement remains in our revised manuscript beginning on page 19, line 24. These changes were included in our analyses because they directly change seafloor structure and alter habitat distribution.

**Generally, this paper mainly appears as a "technical" paper that describes the GIS pro- cedure applied to calculate changes in elevation, without addressing in an adequate way the conceptual, and the data and methods aspects raised. It seems that authors do not have the required background to address the complex scientific question that they have chosen to address. The technical procedure described on pages 4-6 is in-**

**comprehensible to me. Despite the fact that I failed in understanding this procedure, my feeling is that the method is not robust due to poor conceptual, and data and method, bases.**

We have rewritten the methods section to more clearly explain the concepts and procedures of our methods and analyses. We have added substantial discussion on the limitations of these types of analyses, and the value of considering elevation-change results alongside process studies to better understand and account for causes of physical change in seafloor structure. Our findings showing that the magnitude of regional scale seafloor erosion that has occurred in these systems has increased relative sea level rise causing water depths not expected to occur until 2100 are, in fact, very significant. We recognize that total erosion at the regional scale has likely been underestimated because no prior studies fully account for all processes causing elevation change in coral reef ecosystems; and we understand that our results may cause controversial feelings. However, we feel that we have proven the validity of our results and use of historical and modern data sets for our analyses with our expanded error analysis and our use of a very conservative RMSE for data calculations. Our conclusions regarding loss of seafloor volume are based on actual measurements of elevation-change shown to be statistically significant in over 90% of the habitats we analyzed, and that account for all of the processes causing elevation loss in these regions. We have moved the technical procedures to a supplemental section to improve readability of the main paper.

**In different sections of the paper (e.g., page 13, line 1), the results obtained are cor- related to generalities, e.g. on coral reef degradation, which is questionable. Results should be correlated to local data on reef health, including observed changes in living coral coverage, but not to worldwide observations. The interpretation of the results generated is not satisfactory: for example, the au- thors mention hurricanes as key controls of changes in depth. This raises the question of "what is measured, either long-term changes related to climate change and sea level rise, or changes due to low-frequency high-magnitude events"? Once again, this makes the paper confusing.**

We have clarified in our discussion that our intent was to provide general comparisons of our elevation and volume-loss measurements to individual process rates from the literature to provide context for the magnitude of our results. These comparisons also demonstrate the potential value of knowing net whole system change when accounting for and identifying individual processes, for identifying missing gaps in erosion/accretion budgets, and serve as an example of how our results may be used in future studies to improve erosion and accretion budgets in coral reef ecosystems. This discussion is now included on page 20, lines 30-34 and beginning of page 21. We have also included reference to and discussion of more site-specific process studies for comparison to our studies, again to demonstrate the magnitude of our results and potential value of combining elevation-change analyses with process studies. See page 22, line 1 – page 24, line 14. Again, we emphasize that detailed study of the processes causing elevation change in these study sites was beyond the scope of the paper. We do, however, compare our results to observations of change and processes specific to these study sites. For example,

On page 18, lines 6-28, we note that a sub-region of the Upper Florida Keys showed a slight increase in elevation and that this location is near an area of the Middle Florida Keys that has been identified by Manzello et al. (2012) as a possible refuge from ocean acidification based on local data for that area. We reference work by Lidz et al. (2007) and Shinn et al. (2003) that corroborates our observation of accretion on spur-and-groove habitat due to burial by sand and evidence for redistribution of reef materials by hurricanes that has caused erosion in some areas and deposition in other areas of the Florida reef tract. On page 22, lines 7-10, we compare our erosion rate in the Upper Florida Keys to chemical erosion rates determined for the Florida Keys by Muehllehner et al. (2016). All of these are examples of comparison of our results to local data on reef health and processes. We discussed that our methods and results account for both long-term changes over the time periods of our study as well as low-frequency, high-magnitude events in the first paragraph of our discussion.

**Page 2, lines 28 to 30 are incomprehensible: "measures of total system change in seafloor elevation and volume are required to accurately assess and predict the impact of reef degradation on the vulnerability of coastal communities to hazards caused by storms, waves, sea level rise and erosion".**

We have clarifed this passage to explain that hydrodynamic and other numerical models used to assess and predict the impact of reef degradation on coastal hazard vulnerability are limited by lack of comprehensive assessments of change in seafloor structure of coral reef ecosystems, on page 2, line 33 to page 3, line 2.

**Page2, lines 31-32: "we quantify the combined effect of all constructive and destructive processes on modern coral reef ecosystems by measuring regional-scale changes in seafloor elevation" is incomprehensible.**

We have rewritten this sentence to clarify, "Our study is based on the premise that elevation-change analysis measures the net result of all of the constructive processes that cause accretion (or increases in seafloor elevation) and the destructive processes that cause erosion (or decreases in seafloor elevation) on whole reef systems", on page 3, lines 13-16.

**Page 3: "we adapted an elevation-change analysis method that has traditionally been used to monitor seafloor changes"**

We are uncertain as to the question regarding this statement.

**Page 7, line 25 "sediment thickness of the Holocene reef deposit": what do the au- thors talk about? Vertical sedimentation? Vertical Holocene reef building? The results exposed page 8, lines 11 to 14 for the Lower Florida Keys case study are incompre- hensible to me. I do not understand how the authors "used a moder reef age of 6000 years and a constant erosion rate" to "compute the time required to completely erode the remaining Holocene reef down to the Pleistocene layer".**

We have added a section describing the geologic setting of the Holocene reef deposit and Pleistocene bedrock of the Florida Keys, and we have expanded discussion of the methods to better clarify the concept and procedures we used for this analysis. See page 25, line5 to page 26, line 4.

**Page 8, lines 29-30: I am surprised to read that vertical errors would be comprised between 9.6 and 11.8 cm respectively, given what I know on LiDAR data and the hor- izontal error (1 to 4Âa ̆m) applying to this study. More generally, I do not understand how vertical error estimation was conducted.**

We have rewritten this section (now Section 2.11 beginning on page 11) to better define and clarify the error terms in our calculations, how they were derived, and calculation of total RMSE.

**Page 12, lines 13 to 27– We understand that most study are not dominated by coral reefs, which means that this paper does not in fact address the pretended issue of reef response to changing environmental conditions. This suggests that the choice of study sites is not totally coherent with the objectives of the paper.**

We have explained what we included in coral reef ecosystems within our study, as well as justification for including non-coral dominated habitats in our analysis on page 4, lines 4-10.

**Page 12, lines 25 to 27: the conclusions drawn by the authors from the study of Buck Island correlates volume loss to sediment export. Both the results (volume loss) and the interpretation of the results (sediment export) are unclear to the reader.**

We have expanded our discussion of the results from Buck Island and included figure 6 to support our discussion of volume loss, sediment export and how physical degradation of coral colonies can lead to volume loss without export. See page 17, lines 10-34 to beginning of page 18.

**Page 14, lines 20-25: how can the authors convert "changes in elevation" into a "num- ber of years of Holocene reef accretion"? This is not robust as coral reefs grow and erode over a given period, as a result of the complex imbricated processes driving both reef construction (i.e. construction) and sediment production (i.e. erosion allowing car- bonate production).**

The point of this exercise was to further demonstrate the significance of these losses. We have rewritten this paragraph to clarify and included more detail on this calculation including equations, see page 21, lines 4-34.

**Bottom of page 15-top of page 16: I do not understand how the results obtained by the authors can be compared to the results of previous studies conducted by C. Perry to attribute observed changes to specific**

**drivers/processes.**

Perry's 2015 paper is one of the most recent studies showing that while remote coral reefs that are largely isolated from human influence experience severe coral mortality from climate driven impacts like bleaching, most of these reefs recover very rapidly and continue to produce enough carbonate to keep up with present and future sea level rise. The C. Perry study was based on assessment of 28 reefs across the Chagos Archipelago reef system in the Indian Ocean. The point we are making here is that, although Maui is remote, it is not isolated from human influence; and our results showing large erosion rates suggest that these reefs systems have not recovered well from degradation and are not producing enough carbonate to keep up with rising sea level. We have rewritten a sentence on page 24, lines 25-28 to clarify this concept.

**Page 15, lines 19-21: the estimation that "the total reef volume could completely erode down to Pleistocene-bedrock-surface in approximately 1250 years" is not well-founded.**

AR: We have clarified our concept and procedure for this analysis and expanded discussion of the geologic setting on page 25, line 5 to page 26, line 4.

**Page 15, line 33: ". . . reef systems. . . lack human impacts" is not correct in terms of style.**

The terminology used in this sentence is from Perry's 2015 paper, and we are not certain as to what the reviewer is recommending here.

**Page 15, lines 23-35: key references on reef islands future are not cited by the authors. See in particular the recent studies by Kench et al.**

We have now included a detailed discussion on the future of reef islands on page 22, line 26 to page 23, line 15, and how their results differ from our observations in Maui. We have included the following references in this discussion:

McLean, R., & Kench, P. (2015). Destruction or persistence of coral atoll islands in the face of 20th and 21st century sea-level rise? Wiley Interdisciplinary Reviews – Climate Change, *6* (5), 445-463. 10.1002/wcc.350

Kench, P. S., Thompson, D., Ford, M. R., Ogawa, H., & McLean, R. F. (2015). Coral islands defy sea-level rise over the past century: Records from a central Pacific atoll. *Geology, 43* (6), 515-518. 10.1130/G36555.1

Kench, P. S., Owen, S. D., & Ford, M. R. (2014). Evidence for coral island formation during rising sea level in the central Pacific Ocean. *Geophysical Research Letters, 41* (3), 820-827. 10.1002/2013GL059000

**Page 16, lines 3-5: an assumption like "Modern carbonate production rates are an or- der of magnitude lower than Holocene averages (Perry et al., 2013), and are estimated to decrease by as much as 60% by mid-century (Langdon and Atkinson, 2005)" is far too general.**

These statements are not assumptions, rather they are derived from the field and experimental results of the cited studies. We do not use this data to quantify future responses of reefs, but rather to point out that carbonate production rates are projected to continue to decrease while bioerosion and chemical erosion are projected to increase in the future. These combined impacts are, in fact, likely to accelerate reef erosion rates. We have added reference to other studies that showed some reef systems will be able to maintain pace with sea level rise, along with the statement that "vertical accretion rates vary from site to site within a given region, some localized areas may continue accreting, and other coral reef ecosystems may be able to maintain pace with sea level rise". See page 24, line 30 to page 25, line 3.

**Tables 2 and 3 – The substrate categories included in these table are not presented and justified in the study. We additionally have not idea of the depth at which these habitats are situated.**

This statement is incorrect. The habitat/substrate maps were discussed and referenced on page 6, lines 18-27 (now page 9, line 22 to page 10, line 6) and in the supplemental methods section. We have also moved tables 9 and 10 in the revised manuscript from the supplemental section. These tables list the habitats within each study site along with habitat area, and we have now included mean elevation (depth) data for each habitat in Table 9.

**The maps provided page 30 indicate a complex spatial distribution of gains and losses, which is not described in the paper. They also show that shallow habitats were not totally covered, suggesting that gains may have occurred in non-covered areas that may compensate observed losses in study areas. This is all the more to be considered that the results obtained are contrasting (e.g. between the Central Sub-Region and the Lower Sub-region of the Florida Keys).**

We have added a sentence on page 15, lines 26-25 specifically pointing to the complex spatial distribution gains and losses in our map figures (now 5, 6 and 7), to accompany our discussion of the range of elevation gains and losses we observed. We have included discussion of this distribution of data on page 20, lines 6-14 of our Discussion section.

There are gaps in the data coverage for the Florida Keys and Buck Island data sets and we have noted these in our revised manuscript on page 5, lines 16-19. We have also explained that our analysis was limited by the areal extent of available data sets. However, there is overwhelming evidence in the literature (as well as anecdotal evidence from field observations) that the trends we see in the areas covered by our data sets are consistent throughout the

surrounding areas outside of the boundaries of our study sites. We feel that our study areas in these locations are at a large enough scale to be representative of the broader region. We do not argue that there are some locations within our study sites that show accretion. For example, offshore, downslope areas where sediment is infilling spur and groove formations; areas shoreward of patch reefs where sediment from degraded coral reefs has been redistributed to deeper water habitat behind the reef; and, in the case of the Lower Sub-region of the Upper Florida Keys, a small area of increased elevation primarily associated with seagrass beds (that we discuss on page 18, lines 16-21 of the revised manuscript). However, the amount of total volume loss at these study sites is substantial. To put this in perspective, our results indicate that seafloor volume has decreased in the Upper Florida Keys study site by 14.6 (lower bound) to 37.9 (upper bound) million cubic meters. One million cubic meters is approximately the same volume as the Empire State Building. So, the amount of seafloor volume lost in this area is equivalent to approximately 14 to 37 Empire State Building's worth of material volume. There is no evidence from the numerous other geological, ecological, or geophysical studies throughout the Florida Keys for redistribution and deposition of this amount of seafloor material in the shallow habitats that lie between the outer reef tract and the shoreline of the Florida Keys.

**Concerning the Florida Keys, curiously nothing is said in this paper about the dominant modes of planform change and about Keys' landward migration. This suggests that the general context that allows interpreting correctly the results is not presented and considered when analysing the results.**

We have included a brief summary of Holocene reef formation and platform change in the Florida Keys to help develop the context for the analysis described in section 2.4 (Lower Florida Keys – volume to Pleistocene Bedrock), see page 25, lines 5-23 of revised manuscript. The time period of our elevation change analysis in the Florida Keys focuses on changes over the past 64 to 68 years. We feel that a broader discussion of platform evolution and change that occurs over much longer time scales than the processes accounting for much of the change we observe (discussed in manuscript) is well beyond the scope of our paper. Additionally, all of our data were corrected for sea level rise that accounts for subsidence, etc.

**The results obtained in Saint Thomas, as shown by Map a page 31 mainly exhibit stability to limited elevation loss, if we consider grey and yellow areas. When we see this map, we are not convinced that elevation losses prevail, especially if we consider the error range. The same observation can be made when considering map c page 31 showing the situation of Buck Island (blue and yellow area are extensive).**

We have adjusted the color schemes in all of our map figures (5, 6 and 7) to better illustrate gains and losses. The figures in the original manuscript were unevenly toned such that colors showing elevation gains overwhelmed colors showing elevation losses. We have also moved [now] Table 9 from the supplemental to the main manuscript. This table provides detailed elevation gain and loss data from each habitat type.

**Reviewer 3**

**Overall, I am impressed by this paper and think that it is an interesting attempt to ele- vate monitoring to something more than "counting corals". However, I am concerned that the likely variability in sources of substrate change were probably much more dif- ferent from site to site than has been characterized. I could be wrong, but I suspect that bioerosion is less of a factor than is represented here. . . and is more likely declin- ing at most Caribbean sites. While there is an effort to address site-to-site variability, I am not convinced that the relative roles of simple bioerosion, large-scale rugosity loss and export by storms have been adequately considered. I would like to see this paper appear in print, if only for the valuable data set. However, I am concerned that the ex- planations of the measured patterns is a bit oversimplified and relies too much on the mechanisms proposed. I, therefore, provide some over-arching thoughts below in the hope that the authors can perhaps think a bit more about other possible explanations for the patterns they observed. Accordingly, I make a few general observations below that will hopefully be useful.**

We have clarified that the primary intent for this paper was not to focus on a detailed analysis of the processes causing elevation change, rather to focus on introducing this type of monitoring approach to the coral reef scientific community as the first application of seafloor elevation change methods to whole coral reef ecosystems, to demonstrate the value of this approach, and to report the substantial regional-scale net change in seafloor elevation that has occurred over the past several decades and has gone undocumented until now. We have added an expanded explanation of the aim and scope of our study to the introduction on page 3, line 11 to beginning of page 4. We have also emphasized that our methods do not attribute change to cause in an expanded section of our discussion, page 19, line 24 to page 21, line 2.  In this section we have clarified that discussion of mechanisms of substrate change was an exercise to demonstrate the magnitude of net erosion that we measured by elevation change relative to individual processes for which process rate estimates were available including bioerosion among others. We also included a statement in this section clarifying that these comparisons demonstrate of how these regional scale elevation change analyses might be used to help attribute change to cause by way of identifying and quantifying changes that have been unaccounted for by known process rates.

**Comments:**
**Like Reviewer 1, I am not well versed in the GIS and data transformation methods utilized in this study. However, I am familiar with the vagaries of older hydrographic sur- veys. On the latter front, I am willing to accept their characterizations of (the direction of?) change in substrate level as the differences between sites are probably suffi- cient to overcome any stated errors. However, in my experience, the notes on smooth sheets leave us with a need to make defensible assumptions about a) the reliability of substrate characterization (and its stability) and b) how processes that potentially influence elevation change might differ from site to site. I have limited my comments to the latter, based on areas in the manuscript where I have experience in either the specific habitats or the processes that might contribute to the patterns described.**

**Before I start, I do have one comment on style. I am not qualified to comment on the statistics of the methods or the assumptions made in the GIS transformations and map algebra. Nevertheless, a more reader-friendly explanation on that front would make the paper more accessible to a broader audience. The paper in its present form is a wealth of information on methods for those inclined to apply them to other sites. However, those people are probably going to be less well informed on the evolution of carbonate substrates. Conversely, those with intimate understanding of carbonate cycling are going to be unable to tie their knowledge to the details of the methodology used here. I am in that latter group and would suggest that the minutiae of the transformations and GIS tools could be better placed in the Supplemental Materials.**

We have rewritten the methods section so that a broader audience can more easily follow along and have made significant other changes including:

1) We have created and included a flow diagram (Figure 1 in revised manuscript) that depicts the core processing steps in our methods.
2) We have moved the technical data processing steps to a supplementary methods section.
3) We included Section 2.2 (page 7, lines 15-31) on the methods we used for preliminary inspection of historical data sets during the selection process, including an example figure of that process (Figure 2).
4) We included more detailed discussion on developing the elevation-change surface models used in our study and an example figure (Figure 3) to further illustrate the process. We have also included photographs from visual inspection of our study sites as supporting evidence for our discussion (Figures 8 and 9).
5) We have included more detail on the very rigorous data and error evaluation analyses that we performed including a detailed discussion of data resolution and allowances for historical and modern data (page 6, line 10 – page 7, line 13).
6) We also included methods and results for pavement analyses we performed to determine validity of comparing historical and modern data sets (page 13, one 25 – page 14, line 11) prior to beginning elevation-change analyses, and a figure of results from this analysis (Figure 4).
7) We have also modified our discussion of error calculations to better clarify these methods page 11, line 15 – page 15, line 20.
8) We have moved supplementary Tables S1, S2 and S3 that describe details regarding the data sources, data conversions and study periods for each study site to the main paper, now Tables 2, 3 and 4 in revised manuscript, to help further clarify our methods.

**The following ae my general thoughts based on elements of carbonate cycling that could lead to conclusions other than those drawn here. While I am willing to accept the numerical changes in substrate elevation, I am somewhat less comfortable with assumptions about the degree to which they are related to bioerosion and the ensuing removal of sediment.**
**Biorosion versus structural reorganization – In the discussions, there is an apparent conflation of bioerosion and spatial heterogeneity. The paper by Alvarez-Fillip et al. (1990) that is cited to document the role of**

**increased bioerosion focused on the loss in architectural complexity (aka rugosity) and its causes – not bioerosion. In the paper, they attributed the initial reduction in reef rugosity to the loss of acroporids and the second decline in rugosity to a loss of massive species following bleaching. It seems reasonable to assume that an increase in susceptible substrate could increase bioero- sion. However, Alvarez-Fillip et al. focused on the loss of rugosity which, in the case of A. palmata, is more easily explained by physical toppling/breakage and incorporation of fragments into a broad, cemented pavement.**

The Alvarez-Filip et al. (1990) reference should have been listed earlier in the sentence to which the reviewer is referring. We thank the reviewer for catching this mistake. On page 1, line 30 through page 2, lines 1-3 of the original manuscript, the sentence reads:

"Local and global, natural and human-induced stressors have caused the loss of reef-building organisms and reef structure, a decrease in biodiversity, a transition to algal-dominated communities (Pandolfi et al., 2003), and an increase of bioerosion (Alvarez-Filip et al., 2009), placing coral reefs around the world in a state of rapid decline (Madin and Madin, 2015)."

The Alvarez-Filip reference should have been placed after the statement "…have caused the loss of reef-building organisms and reef structure…". The reference that was intended to follow the statement "…and an increase in bioerosion…" should have been Enochs et al. (2015), see reference list in manuscript. The Enochs reference discusses experimental and modeling results showing that ocean acidification increases bioerosion of coral by the boring sponge *Pione lampa*. We have corrected this mistake and clarified in this sentence that we are referring specifically to some species of boring and endolithic bioeroders. We have also include the following references as other examples (see page 1, line 25 to page 2, line 5):

Wisshak M, Schönberg CHL, Form A, Freiwald A. 2013. E ects of ocean acidi cation and global warming on reef bioerosion – lessons from a clionaid sponge. Aquat Biol. 19(2):111–127. http://dx.doi.org/10.3354/ab00527

Wisshak M, Schönberg CHL, Form A, Freiwald A. 2012. Ocean acidi cation accelerates reef bioerosion. PLoS ONE. 7:e45124. http://dx.doi.org/10.1371/journal.pone.0045124

Tribollet A, Godinot C, Atkinson M, Langdon C. 2009. E ects of elevated $pCO_2$ on dissolution of coral carbonates by microbial euendoliths. Global Biogeochem Cycles. 23(3):1–7. http:// dx.doi.org/10.1029/2008GB003286

Reyes-Nivia C, Diaz-Pulido G, Kline D, Hoegh-Guldberg O, Dove S. 2013. Ocean acidification and warming scenarios increase microbioerosion of coral skeletons. Glob Change Biol. 19(6):1919–1929. http://dx.doi.org/10.1111/gcb.12158

DeCarlo T., Cohen, A., Barkley, H., Cobban, Q., Young, C., Shamberger, K., Brainard, R., and Golbuu, Y. 2015. Coral macrobioerosion is accelerated by ocean acidification and nutrient. Geology 43, 7-10, doi:10.1130/G36147.1

**The interval of measured elevation changes included the loss of A. palmata. It, therefore, seems likely that this could have played a greater role than the removal of bioeroded sediment in the changes described in the manuscript. Alvarez et al also pointed out that the loss of Diadema logically re- duced bioerosion despite the greater availability of "bioerodable substrate". Likewise, in many (most?) Caribbean and western Atlantic sites, parrotfish populations have been decimated, further reducing the potential for bioerosion by grazers. The remain- ing option is infaunal bioerosion by sponges, worms, etc. However, unless there is a very significant increase in organic availability, the likelihood of that being significant seems unlikely.**

We agree that loss of A. palmata and other framework building species likely contributed to elevation loss at these study sites, and we address this issue in section 3.2 beginning on page 16, line18. We have also added more detailed discussion of the effect of coral toppling without bioerosion on elevation and volume changes (page 10, lines 10-27), as well as information on declines in grazing bioeroders for these study sites in Table 1 and discussion on page 4 beginning on line 20. We have expanded our discussion of the relative roles of bioerosion versus physically toppling of corals and how these processes might affect results (page 17, line 10 to beginning of page 18). We have also shown by comparison to studies that consider rates of bioerosion, that bioerosion alone cannot account for the elevation and volume losses that we observed, page 21, line 31 to page 22, line 25.   We note that much of the erosion we observed at our study sites is likely attributed to causes other export of bioeroded sediments, page 22, lines 24-25.

**It is interesting that at one of their sites (Buck Island), Bill Gladfelter proposed two threats to reef building in a 1977 report to the Park Service: 1) the loss of carbonate production if WBD increased, and 2) the possibility that protection of parrotfish might significantly increase bioerosion to the point where it could overwhelm even produc- tive reefs. This would suggest that increased bioerosion by grazing fish could lead to detrimental increase in bioerosion. In the latter scenario, increased grazing becomes a problem only in protected areas where grazing fish have increased (like the FKMS, one of the described sites where increased bioerosion might be a reasonable culprit). Else- where in the Caribbean, parrotfish populations have been decimated. In combination with the loss of the major grazing urchin, a wholesale increase in bioerosion capacity seems unlikely. Lost calcification ability would decrease accretion, but does not seem like a driver of net erosion unless bioerosion increases – a pattern that has not been documented at all sites.**

We agree that losses and gains of grazing fish will directly impact rates of bioerosion by those species. We have clarified in our discussion that there are a number of different types of bioeroder species including grazing fish as well as sponges, bivalves, microendoliths, etc. We have noted the potential impact of increases or decreases in grazing fish on rates of bioerosion; and we clarified that recent work indicates that elevated pCO2 and ocean

acidification accelerates bioerosion by endoliths and other boring bioeroders. See pages 17, line 27 to page 18, line 5. While we point to bioerosion as one of many contributors to erosion in these systems, we agree (and show by comparison in our discussion) that the contribution is small relative to the net changes we measured.

**So, that leaves us with export. As the paper points out, good data on export are rare. On page 2, Moses (2009) is cited for measuring sediment export from reefs, but I could find no measurements in that paper. Kench and McLean provide an estimate of trans- port potential through hoa in Indian Ocean atolls. However, the results are based on theoretical calculations and there is no effort to tie sediment to specific sources (e.g., bioeroded sediment, beaches, lagoons) or sinks (loss to lagoons vs export from the platform).**

The reviewer is mistaken regarding the Moses (2009) reference. Moses (2009) is cited with respect to regional-scale chemical erosion measurements of carbonates (page 2, lines 23-24). In a sentence after that (beginning on line 24), Morgan and Kench (2014) and Kench and McLean (2004) are cited in a sentence that reads "Very few studies have quantified sediment transport and export on reef systems". We note again here that many of our references in the paper have typos in them from our End Note conversion that we did not catch prior to submission. For example, many were abbreviated (e.g. Morgan and Kench 2014 got shortened to Morgan 2014, and Kench and McLean 2004 got shortened to Kench 2004), and others reverted back to numbers (as was the case in Supplementary Table 1 in which the source references reverted back to numbers). We have corrected all of these in the revised paper.

Morgan and Kench (2014) is one of the few studies that estimated sediment flux and off-reef sediment export from direct point measurements using arrays of bi-directional sediment traps for an atoll reef (Vabbinfaru reef, North Male' Atoll) in the Maldives. They showed high off-reef export rates for both gravel and sand (annual export estimated at over 120 metric tons per year), as well as high sediment flux rates even during non-storm conditions.

Kench and McLean (2004) estimated transport potential from current meter and tidal gauges located in the HOAs, but also measured sediment fluxes in hoa's from sediment traps ranging from approximately 2kg to 268 kg per day. Based on daily rates of sediment flux, they estimate that from 44 to 223 metric tons per year of sediment may be transported by hoa's.

We have included additional references to sediment export studies by Hubbard in this section (page 2, lines 24-25). We have also expanded our discussion of sediment export including more detailed discussion of results from previous studies (including Hubbard et al., 1981; Hubbard, 1986; Hubbard et al., 1990; Hubbard, 1992) as well as supporting evidence for our results from sediment resuspension and transport studies by Ogston, Presto, Storlazzi, and Fletcher), see page 23 line 16 to page 24, line 14). Results from these studies support our suggestion that physical erosion and export of sediments could account for much of the volume loss we observed at our study sites (page 12, line 25-28).

**What is, therefore, critically important is a reliable estimate of export inas- much as volume must be exported from the system to trigger system-wide elevation loss. . . bioerosion just converts carbonate from solid substrate to sediment. In the lat- ter case, we must remember that sediment has a much lower bulk density than solid carbonate substrate. Thus, increased bioerosion without export would reduce the vol- ume of solid substrate but would turn this into a sediment pile with something akin to twice the net volume. Thus, increased bioerosion without export would result in sub- strate elevation; not lowering. A scenario based solely on increased bioerosion seems inadequate to explain the measured patterns.**

We agree with the reviewer, and make the case in our results and discussion that the volume loss we observe over the large scale of our study sites is an indication of export of materials from these systems. We do not suggest that any scenarios based solely on increased bioerosion could account for the volume losses we observed. We reworded the sentence on page 22, line 23-25 to clarify that export of bioeroded sediment cannot fully account for the erosion we observed at our study sites. As indicated, we have greatly expanded our discussion on physical erosion and export of sediments as a likely cause for much of the erosion we have observed.

**Unfortunately, there has only been a few careful measurements of sediment export in the context of a reef-wide budget. Perry and various co-authors use our ratio (Export ~ 50% of total bioerosion) from the north coast of St. Croix to characterize this in every one of their budgets. It is naïve to think that all reefs in all oceans have the same energy regime (the driver of export) – or that changes in energy regime is offset by proportional shifts in bioerosion to maintain the 50% value that is used throughout. With increasing storminess, sediment export looms as the single largest unquantified variable. Therefore, export can only get more significant in the budgeting attempted in this paper.**

Again, we agree with the reviewer. Note in our discussion that we suggest that much of the volume loss we observe in these regions may be attributed to physical transport and export of sediments, and that export has been largely unaccounted for in previous studies (page 22, lines 23-26). We also note that the much greater seafloor elevation and volume losses observed over shorter time periods at the Maui study site could be caused by higher sediment export rates due to a combination of higher wave energy and physical erosion as well as a narrow shelf surrounding the island (page 19, lines 6-11). We have also added discussion of sediment flux studies that support this concept (page 23, lines 6-16).

**In section 3.2, the paper acknowledges the difference between bioerosion and changes in structural complexity. How good the conclusion will be is going to depend on how well one can distinguish between the two as potential drivers of elevation change. The conclusions presented here seem to suggest 1) an ability to reliably distinguish be- tween the two mechanisms and 2) an overwhelming importance of simple bioerosion over combined changes in export and reduced structural complexity following the loss of biological constructors.**

We have clarified and emphasized that our analyses do not attribute change to cause. We make no claims to be able to distinguish between bioerosion and changes in structural complexity. We have provided photographic evidence (Figure 8 in revised manuscript) to illustrate how loss of structural complexity can differ from the result of bioerosion, and as evidence that physical degradation (toppling of corals) has occurred at the BI study site, consistent with observations of elevation loss. We feel that our expanded discussion now emphasizes the role of changes in export and reduced structural complexity as likely causes for much of the erosion we observed at these study sites (page 22, line 28 to page 24, line 15).

**Anthropogenic drivers of change - Using population as a proxy for anthropogenic im- pact seems overly simplified. Numerous recent papers have shown that some of the greatest reef losses occur due to warming/acidification at great distances from any rec- ognizable urban stressors. I can't find the specific papers, but there has been quite a bit of discussion on the NOAA listserve about papers that show just this. While I am not in the midst of the debate over local versus global drivers of change and their im- plications for management, this proxy seems a bit simplistic.**

We have removed the statements that we use population as a proxy for anthropogenic impact, and instead we simply state that population has doubled and local anthropogenic impacts have increased at each study site (page 5, lines 19-22). We have moved the Table with population statistics from the supplemental to the main paper (Table 2), and we also include a table and discussion on declines in coral cover and other indicator species (Table 1, page 4, line 20 to page 5, line 5) as evidence for anthropogenic impacts.

We cited Perry's (2015) work showing that reef systems in the Indian Ocean that are geographically isolated and lack human influence show heavy impact from bleaching events, but are able to recover very rapidly (see discussion on page 24, lines 23-31) and show very high accretion rates. We note that Maui is geographically isolated but shows very large erosion rates suggesting that it has not been able to recover from reef loss, possibly due to the fact that it is not isolated from a large human population. We also point out more than one case where lower losses of elevation and volume coincide with areas further removed from large population centers or associated with natural refuge zones (same section).

**On a more specific point, the manuscript discusses the idea of proximity to anthropogenic areas to explain the positive elevation change in the lower Keys. Couldn't this also be due to separation from the inimical cold bank water allowing for higher calcification rates? In this vein, limited core data from the Keys seem to suggest that the "demise" of the reef tract likely started 4-5,000 years ago as Florida Bay flooded, triggering inimical (cold) water export onto the reefs. In contrast, the reefs around Buck Island enjoyed continuous building throughout this period as there was no similar source of stress.**

The reviewer misunderstood this discussion now located on page 18, lines 16 – 21:

"Mean total elevation loss was lowest at the UFK study site. However, mean elevation losses decreased from upper (-0.4 m) to central (-0.3 m) sub-regions of the UFK, and mean elevation increased slightly in the lower sub-region (0.1 m) primarily associated with seagrass habitat (Fig. 1a, b). Notably, the lower sub-region is further away from high-density population areas north of the study site and near an area of the middle Florida Keys identified as a possible refuge from ocean acidification due to seagrass productivity (Manzello et al., 2012)."

The area that showed the elevation increase was in the lower sub-region of the Upper Florida Keys (not in the Lower Florida Keys), see Figure 5a in the revised manuscript. The mean elevation change in the Lower Florida Keys was -0.3 m (see Table 8 in revised manuscript), and was similar to the elevation loss we observed in the central sub-region of the Upper Florida Keys. These data would suggest that the inimical waters could be less likely to have had an impact on this sub-region of the Upper Florida Keys.

**All of this would suggest that these two areas have had very different exposures to natural stresses; this would presumably make for very different susceptibilities in more recent times when increasing anthropogenic stress is set up as the main driver. It may also be noteworthy that the sediment thicknesses in these two areas are different and there is evidence that sediment retention around Buck Island (much higher wave energy and susceptibility to both storm damage and sediment export) may tend to be less than is the case in the Keys. If the latter is true, then changes in substrate elevation might be sediment export in one place, bioerosion in another and wholesale loss of rugosity in all.**

Interestingly, our volume calculations showed the lowest 'maximum net volume' loss (Table 8, column 10 in revised manuscript) and 'mean elevation' loss (Table 8, column 3 in revised manuscript) at Buck Island, suggesting that sediment/material retention was greater at Buck Island over the study period than in the Florida Keys. We suggest that this could be due to loss of rugosity (large framework building coral) but less export of materials at Buck Island and have expanded our discussion of that beginning on page 17, line 10. We also added photos from the Buck Island study site supporting our observations of rugosity loss and the potential implications for materials transport in this section.

**I assume that the substrate type and sediment thickness was not consistently noted in older surveys. Given the points above, this could be an important driver of how quickly substrate elevation might change in one place versus another. The wholesale loss of architecturally complex acroporids and the subsequent reduction of these to pavement could be construed as "degradation of framework-building corals" as could bioerosion. Which was the main agent in each case?**

Unfortunately, substrate type and sediment thickness were not consistently noted in the older surveys.

**Reviewer 3 – additional comment**

**As I initially said, I think this is a good study - for exactly the same reasons the readers intended. I am satisfied with the revision suggestions but strongly encourage the authors to make sure that the reader understands that the main goal is to focus on the magnitude of change with the caveat that attributing mechanisms is much less well constrained. In this vein, I suggest that the authors take an opportunity to perhaps comment on where we have big gaps in our knowledge that hamper such attributions and why at least a semi-quantitative understanding of the relative relationships between carbonate addition and removal is important to our understanding of the changing dynamic between accretion/erosion and SL rise. The non-linear relationship between water depth across the reef and wave energy passing by has been elegantly demonstrated by many careful instrumental studies. This study addresses ways to quantify the first half of that relationship and, therefore, has considerable value when we move on from coral loss to longer-term physical and social impacts - whether in Miami or some small atoll in the Indo-Pacific region.**

We expanded the first section of our discussion (page 19, line 24 to page 21, line 2 to comment on some of the key gaps and issues that make attribution of physical change to cause difficult in these systems, and the benefits from preforming comprehensive elevation-change analyses. We have also now included a Conclusions section (page 26, beginning on line 23) that the discusses the potential value of elevation change analyses for moving the field of science toward better establishing links among contemporary processes causing coral reef ecosystem degradation, long-term changes in the physical structure of the environment, and future implications for physical and societal impacts. We also include discussion of the key limitations of elevation change analyses, and summarize the key findings from our study.

**Reviewer 4**

**The paper by Yates et al. analyses bathymetric data to quantify seafloor elevation changes in coral reef regions. As highlighted by reviewers 1 and 3, the dataset pre- sented in the manuscript is especially impressive (number of sites considered, extent of the area considered). The results will be useful for coastal geomorphologists and managers concerned with the sustainability of coral reefs environments and the related ecosystem services. Despite the recommendation of reviewer 2, I think that the paper should be published after major revisions, to ensure that the amount of data analyzed in this work receives the attention it deserves.**

**Three previous reviews have extensively discussed the paper: overall, reviewers (1) have concerns regarding the ability of the method to retrieve seafloor elevation changes at the required accuracy; (2) made comments on the form of the paper; (3) and the interpretation of the results. The authors have already provided responses to several comments of the reviewers, and intend to implement corrections to their paper, which I**

**think are reasonable. I would suggest that these major revisions are implemented, considering the following points:**

**- All reviewers agree that the paper should separate more clearly what is the overall approach (comparing bathymetric data) from the technical details of the GIS procedure used to produce this data. The authors have prepared a figure as part of their response to reviewer 1 to address this comment. However, I think that the figure remains too technical (e.g., use of the TIN surface wording), and I would support producing the detailed GIS procedure in an annex to the paper. Overall, I agree with previous reviews, who suggested that the authors should consider that their results may have a large impact beyond specialists of coastal bathymetric surveys, so that ideally, they should try to separate the main messages from the technical implementation details.**

We have rewritten the methods section taking into account the suggestions by all of the reviewers, and we have included the more detailed (GIS) steps in a methods supplementary section. We included a modified flow diagram (Figure 1 in revised manuscript) showing our approach to describe the process in more general terms. See also responses to reviewers 1 and 3.

**While the comparison of historical bathymetric sounding with contemporary LiDAR is quite widespread in coastal geomorphology (as reminded by the authors, see AC2 – pages C6-C7), there is always the suspicion that the two techniques induce errors, as highlighted by reviewers 1 and 2. Such errors can arise because the techniques have not the same purpose and therefore don't necessarily capture the same proxies (e.g., highest seafloor elevation features for navigation applications vs average seafloor ele- vation feature for bathymetry data in support to coastal hydrographic modelling). The techniques also have different accuracy/precision (as discussed already), or because of time-sampling issues (as commented by reviewer 2). Overall, I think that the authors make a fair assessment of these errors: in the response of the authors to this com- ment of reviewer 2 (AC2 pages C8), information regarding the vertical resolution of the techniques is provided, while the precision issues are given in Table 4 in the original manuscript. To complete this assessment, I would suggest to provide more information on the planimetric resolution, and the vertical accuracy of the techniques in the core of the article. This includes details regarding the definition of a common reference, which incorporates sea-level rise constructions in a way, which is not completely clear to me based on the original manuscript, page 4 lines 10 and following. Nevertheless, I am confident this does not affect the results of the authors, as the RMSE in vertical da- tum adjustment is probably much larger than the RMSE due to uncertainties in relative sea-level changes for the sites of interest (table 4).**

We have included a statement of caution that not all historical and modern data sets are appropriate for elevation change analyses and indicated considerations that must be made on page 5, lines 24-29. We added discussion and quantification of the historical sounding point densities for each study (page 8, lines 26-30). We point out the

potential limitation of planimetric resolution on page 15, lines 14-17. We included a section on preliminary inspection of historical data sets for horizontal alignment prior to data analysis, and provided an example figure of that process (Figure 2). We also included our pavement analysis and results (Section 2.12, page 13, line 25 to page 14, line 11) as further evidence for the accuracy of these data and the validity of comparing these data sets (with a strong caution that pavement cannot be appropriately used as a control, but can provide supporting information for proper error analyses). We provided very basic definitions of horizontal and vertical references (datums) for these data sets and the need to adjust data so that compared data sets are aligned to the same reference points (Section 2.4, page 8, lines 13-20. We reworded the section on sea level rise correction (Section 2.3, page 8, beginning on line 1) to more clearly explain these corrections. We added information on the range of sea level correction values and showed that the potential error from these corrections was insignificant relative to other sources of error, and we, therefore, excluded it from our RMSE calculations (page 8, lines 8-11). We moved information on the data and sources we used for our sea level rise corrections from the supplemental section to the main paper to help clarify our sea level rise adjustments (Table 4 in revised manuscript).

**Regarding the interpretation of the results: besides the aspects discussed with re- viewer 1 and 2, I think that reviewer 3 provides a very clear line for improving the discussion section, and I hope that the authors will build on it in a future version of the article.**

We have greatly expanded our discussion section based on comments from reviewers 1, 2 and 3, see responses to those reviewers.

**Finally, I think that a "conclusion" section is needed.**

We have included a conclusions section that points out the benefits and limitations of comprehensive elevation change analyses, and summarizes the key findings of our study.

**I hope these comments are useful.**

[revised manuscript text omitted]

Kimberly Yates 2/13/2017 8:53 PM
Moved down [14]: (Min.) and maxim... [182]
Kimberly Yates 2/13/2017 8:53 PM
Kimberly Yates 2/13/2017 8:53 PM
Formatted ... [183]
Kimberly Yates 2/13/2017 8:53 PM
Formatted ... [185]
Kimberly Yates 2/13/2017 8:53 PM
Formatted ... [190]
Kimberly Yates 2/13/2017 8:53 PM
Formatted ... [192]
Kimberly Yates 2/13/2017 8:53 PM
Formatted ... [191]
Kimberly Yates 2/13/2017 8:53 PM
Formatted ... [193]
Kimberly Yates 2/13/2017 8:53 PM
Formatted ... [194]
Kimberly Yates 2/13/2017 8:53 PM
Formatted ... [196]
Kimberly Yates 2/13/2017 8:53 PM
Formatted ... [186]
Kimberly Yates 2/13/2017 8:53 PM
Formatted ... [187]
Kimberly Yates 2/13/2017 8:53 PM
Formatted ... [189]
Kimberly Yates 2/13/2017 8:53 PM
Formatted ... [195]
Kimberly Yates 2/13/2017 8:53 PM
Formatted ... [197]
Kimberly Yates 2/13/2017 8:53 PM
Formatted ... [188]
Kimberly Yates 2/13/2017 8:53 PM
Formatted ... [198]
Kimberly Yates 2/13/2017 8:53 PM
Formatted ... [199]
Kimberly Yates 2/13/2017 8:53 PM
Formatted ... [200]
Kimberly Yates 2/13/2017 8:53 PM
Formatted ... [201]
Kimberly Yates 2/13/2017 8:53 PM
Formatted ... [202]
Kimberly Yates 2/13/2017 8:53 PM
Formatted ... [203]
Kimberly Yates 2/13/2017 8:53 PM
Formatted ... [204]
Kimberly Yates 2/13/2017 8:53 PM
Formatted ... [205]
Kimberly Yates 2/13/2017 8:53 PM
Formatted ... [206]
Kimberly Yates 2/13/2017 8:53 PM
Formatted ... [207]
Kimberly Yates 2/13/2017 8:53 PM
Formatted ... [208]
Kimberly Yates 2/13/2017 8:53 PM
Formatted ... [209]
Kimberly Yates 2/13/2017 8:53 PM
Formatted ... [210]
Kimberly Yates 2/13/2017 8:53 PM
Formatted ... [211]
Kimberly Yates 2/13/2017 8:53 PM
Formatted ... [212]
Kimberly Yates 2/13/2017 8:53 PM
Formatted ... [213]
Kimberly Yates 2/13/2017 8:53 PM
Formatted ... [214]
Kimberly Yates 2/13/2017 8:53 PM
Formatted ... [215]
Kimberly Yates 2/13/2017 8:53 PM
Formatted ... [216]
Kimberly Yates 2/13/2017 8:53 PM
Formatted ... [217]
Kimberly Yates 2/13/2017 8:53 PM
Formatted ... [218]
Kimberly Yates 2/13/2017 8:53 PM

Kimberly Yates 2/13/2017 8:53 PM
Formatted ... [389]
Kimberly Yates 2/13/2017 8:53 PM
Kimberly Yates 2/13/2017 8:53 PM
Formatted ... [392]
Kimberly Yates 2/13/2017 8:53 PM
Formatted Table ... [390]
Kimberly Yates 2/13/2017 8:53 PM
Formatted ... [391]
Kimberly Yates 2/13/2017 8:53 PM
Formatted ... [393]
Kimberly Yates 2/13/2017 8:53 PM
Formatted ... [394]
Kimberly Yates 2/13/2017 8:53 PM
Formatted ... [395]
Kimberly Yates 2/13/2017 8:53 PM
Kimberly Yates 2/13/2017 8:53 PM
Formatted ... [396]
Kimberly Yates 2/13/2017 8:53 PM
Kimberly Yates 2/13/2017 8:53 PM
Formatted ... [397]
Kimberly Yates 2/13/2017 8:53 PM
Kimberly Yates 2/13/2017 8:53 PM
Formatted ... [398]
Kimberly Yates 2/13/2017 8:53 PM
Kimberly Yates 2/13/2017 8:53 PM
Formatted ... [399]
Kimberly Yates 2/13/2017 8:53 PM
Kimberly Yates 2/13/2017 8:53 PM
Formatted ... [400]
Kimberly Yates 2/13/2017 8:53 PM
Kimberly Yates 2/13/2017 8:53 PM
Formatted ... [401]
Kimberly Yates 2/13/2017 8:53 PM
Kimberly Yates 2/13/2017 8:53 PM
Formatted ... [402]
Kimberly Yates 2/13/2017 8:53 PM
Kimberly Yates 2/13/2017 8:53 PM
Formatted ... [403]
Kimberly Yates 2/13/2017 8:53 PM
Kimberly Yates 2/13/2017 8:53 PM
Formatted ... [404]
Kimberly Yates 2/13/2017 8:53 PM
Kimberly Yates 2/13/2017 8:53 PM
Formatted ... [405]
Kimberly Yates 2/13/2017 8:53 PM
Kimberly Yates 2/13/2017 8:53 PM
Formatted ... [406]
Kimberly Yates 2/13/2017 8:53 PM
Kimberly Yates 2/13/2017 8:53 PM
Formatted ... [407]
Kimberly Yates 2/13/2017 8:53 PM
Formatted ... [408]
Kimberly Yates 2/13/2017 8:53 PM
Kimberly Yates 2/13/2017 8:53 PM
Formatted ... [409]
Kimberly Yates 2/13/2017 8:53 PM
Kimberly Yates 2/13/2017 8:53 PM
Formatted ... [410]

**Table 9.** Elevation change by habitat type.

[revised manuscript text omitted]

due only to global mean sea level rise of 3.2 mm yr$^{-1}$.

Kimberly Yates 2/13/2017 8:53 PM

Unknown

Unknown

Kimberly Yates 2/13/2017 8:53 PM

Kimberly Yates 2/13/2017 8:53 PM

Kimberly Yates 2/13/2017 8:53 PM